# ScaLoRA: Optimally Scaled Low-Rank Adaptation for Efficient High-Rank Fine-Tuning

## Abstract

As large language models (LLMs) continue to scale in size, the computational overhead has become a major bottleneck for task-specific fine-tuning. While low-rank adaptation (LoRA) effectively curtails this cost by confining the weight updates to a low-dimensional subspace, such a restriction can hinder effectiveness and slow convergence. This contribution deals with these limitations by accumulating progressively a high-rank weight update from consecutive low-rank increments. Specifically, the per update optimal low-rank matrix is identified to minimize the loss function and closely approximate full fine-tuning. To endow efficient and seamless optimization without restarting, this optimal choice is formed by appropriately scaling the columns of the original low-rank matrix. Rigorous performance guarantees reveal that the optimal scaling can be found analytically. Extensive numerical tests with popular LLMs scaling up to 12 billion parameters demonstrate a consistent performance gain and fast convergence relative to state-of-the-art LoRA variants on diverse tasks including natural language understanding, commonsense reasoning, and mathematical problem solving.

## 1 Introduction

Large language models (LLMs) enjoy well-documented success in a broad spectrum of areas including conversational agents (Achiam et al., 2023), software development (Chen et al., 2021), text summarization (Zhang et al., 2024a), and education (Zhang et al., 2024b). Before deploying a pretrained LLM to a certain task, it is often necessary to fine-tune it on domain-specific data to enhance its expertise. With the rapid growth of LLM size in recent years however, conventional full fine-tuning approaches that revise all the model parameters, are increasingly prohibitive due to their substantial computational burden, especially critical for resource-limited applications. For instance, the recent Llama 4 Behemoth model consists of 2 trillion parameters in total, while even its smallest variant Llama 4 Scout contains 109 billion parameters. Even with half precision, full fine-tuning of the latter still necessitates over 1 TB GPU memory, and extended wall-clock time.

As a lightweight alternative, parameter-efficient fine-tuning (PEFT) has been introduced to lower the computational overhead (Houlsby et al., 2019). In contrast to full fine-tuning, PEFT methods refine merely a small subset of parameters (Houlsby et al., 2019; Sung et al., 2021; Li & Liang, 2021), thereby markedly reducing the memory footprint and runtime. Admist these, low-rank adaptation (LoRA) (Hu et al., 2022) has gained particular prominence for its simplicity and efficiency. LoRA presumes the fine-tuning weight update pertains to a low-dimensional manifold, and parameterize it as the outer product of two tall matrices. As a result, fine-tuning the large-scale LLM reduces to optimizing these small "adapter" matrices. Despite its effectiveness and popularity, recent studies have underscored that LoRA and its variants face challenges such as diminishing performance (Hu et al., 2022), and slower convergence (Meng et al., 2024) relative to full fine-tuning, which deteriorate further as the rank declines (Jiang et al., 2024; Huang et al., 2025). Consequently, one has to compromise notable model effectiveness to tradeoff the highly desired efficiency.

To overcome these challenges, this work commits to formulate a high-rank weight update by stacking the per-step low-rank increments. As opposed to vanilla LoRA operating in a fixed low-rank subspace, our key idea is to *dynamically identify the optimal low-rank adapters to update, that minimize the loss per iteration*. To ensure efficient optimization, this optimal choice is restricted to the family of matrices whose columns are scaled from the original low-rank adapters. The advocated ap-

proach is thus termed scaled low-rank adaptation (ScaLoRA). This column-wise scaling allows for efficient re-calculation of moment estimators in adaptive optimizers such as Adam(W), eliminating the need to reset optimizer and re-warm up learning rate. All in all, our contribution is three-fold:

- We prove a sufficient and necessary condition for the optimal low-rank adapters. This condition establishes that the optimal choice requires truncated singular value decomposition (SVD) of the weight gradient matrix, which leads to prohibitive overhead and requires restarting optimization.
- To cope with these two issues, we restrict the new adapters to certain transforms of the original ones. With column-wise scaling as the transform, tractable moment estimators and globally optimal adapters are provably identified in analytical form.
- Numerical tests are performed with DeBERTaV3-base, LLaMA-2-7B, LLaMA-3-8B, and Gemma-3-12B-pt on GLUE benchmark, commonsense reasoning datasets, and mathematical problems (MetaMathQA, GSM8K, and MATH), verifying our analytical claims and confirming ScaLoRA's superior performance as well as accelerated convergence.

**Related work.** Following LoRA (Hu et al., 2022), plenty of variants have been probed to further enhance its effectiveness. For instance, DoRA (yang Liu et al., 2024) decomposes the weight matrix into magnitude and direction components, where only the latter is updated via LoRA. QLoRA (Dettmers et al., 2023) quantizes the pre-trained weights to further reduce computational cost. FourierFT (Gao et al., 2024b) substitutes the low-rank matrices with spectral coefficients and recovers the weight update via inverse discrete Fourier transform. Flora (Hao et al., 2024) leverages random projections to encode and decode the weight gradients. FedPara (Hyeon-Woo et al., 2022) and LoKr (Yeh et al., 2024) integrate Hadamard and Kronecker products into the low-rank outer product. In addition to structural modifications, methods have been developed to refine the initialization of low-rank adapters (Meng et al., 2024; Li et al., 2024; Wang et al., 2024), and adjust the optimization iterations (Wang et al., 2025; Yen et al., 2025; Zhang et al., 2025). Another line of research (Lialin et al., 2024; Jiang et al., 2024; Huang et al., 2025) targets high-rank weight update induced by low-rank adapters. Our ScaLoRA falls in the latter category, and a more detailed comparison will be provided in the ensuing sections.

## 2 LOW-RANK ADAPTATION RECAP

This section briefly recaps LoRA (Hu et al., 2022), the challenges it faces, and existing remedies.

Consider a general weight matrix $\mathbf{W} \in \mathbb{R}^{m \times n}$ of a large model. LoRA decomposes $\mathbf{W} = \mathbf{W}^{\mathrm{pt}} + \mathbf{W}^{\mathrm{ft}}$, where $\mathbf{W}^{\mathrm{pt}}$ denotes the frozen pre-trained weight matrix, and $\mathbf{W}^{\mathrm{ft}}$ is the learnable fine-tuning update. Aiming at efficiency, LoRA assumes the latter lives on a low-dimensional manifold, and can be approximated via $\mathbf{W}^{\mathrm{ft}} := \mathbf{A}\mathbf{B}^\top$, where $\mathbf{A} \in \mathbb{R}^{m \times r}$ and $\mathbf{B} \in \mathbb{R}^{n \times r}$ are "adapter" matrices with $r \ll m, n$. For batched inputs $\mathbf{X} \in \mathbb{R}^{n \times k}$, LoRA's forward operation satisfies $\mathbf{W}\mathbf{X} = \mathbf{W}^{\mathrm{pt}}\mathbf{X} + \mathbf{A}(\mathbf{B}^\top\mathbf{X})$. LoRA reduces the number of trainable parameters to $(m + n)r \ll mn$, markedly lowering the associated memory footprint, and the computational burden of backpropagation.

Letting $\ell(\cdot)$ denote the loss function, LoRA seeks to optimize

$$\min_{\mathbf{A}, \mathbf{B}} \ell(\mathbf{W}^{\mathrm{pt}} + \mathbf{A}\mathbf{B}^\top)$$

With $t$ indexing iteration, define $\mathbf{W}_t := \mathbf{W}^{\mathrm{pt}} + \mathbf{A}_t\mathbf{B}_t^\top$. LoRA initializes $\mathbf{A}_0 \sim \mathcal{N}(0, \sigma^2)$ with a small variance $\sigma^2$, and $\mathbf{B}_0 = \mathbf{0}_{n \times r}$, so that $\mathbf{W}_0 = \mathbf{W}^{\mathrm{pt}}$ remains intact. The subsequent updates rely on adaptive optimizers such as AdamW (Loshchilov & Hutter, 2019). For illustration, consider instead the plain gradient descent (GD) update

$$\mathbf{A}_{t+1} = \mathbf{A}_t - \eta \nabla\ell(\mathbf{W}_t)\mathbf{B}_t, \quad \mathbf{B}_{t+1} = \mathbf{B}_t - \eta \nabla\ell(\mathbf{W}_t)^\top\mathbf{A}_t \tag{1}$$

where $\eta > 0$ is the learning rate, and the gradients $\nabla_{\mathbf{A}_t}\ell(\mathbf{W}_t) = \nabla\ell(\mathbf{W}_t)\mathbf{B}_t$ and $\nabla_{\mathbf{B}_t}\ell(\mathbf{W}_t) = \nabla\ell(\mathbf{W}_t)^\top\mathbf{A}_t$ follow from the chain rule. Then, the per-step weight increment satisfies

$$\Delta\mathbf{W}_t := \mathbf{W}_{t+1} - \mathbf{W}_t = \mathbf{A}_{t+1}\mathbf{B}_{t+1} - \mathbf{A}_t\mathbf{B}_t = -\eta\nabla\ell(\mathbf{W}_t)\mathbf{B}_t\mathbf{B}_t^\top - \eta\mathbf{A}_t\mathbf{A}_t^\top\nabla\ell(\mathbf{W}_t) + \mathcal{O}(\eta^2)$$

where the last term is negligible as $\eta$ is typically tiny (Wang et al., 2024; Hao et al., 2024; Wang et al., 2025; Yen et al., 2025). Summing over $T$ steps yields the cumulative update

$$\sum_{t=0}^{T-1} \Delta\mathbf{W}_t = \mathbf{W}_T - \mathbf{W}_0 = \mathbf{A}_T\mathbf{B}_T^\top - \mathbf{A}_0\mathbf{B}_0^\top = \mathbf{A}_T\mathbf{B}_T^\top. \tag{2}$$

This formulation confines LoRA's weight update to a low-dimensional subspace, which can degrade effectiveness and decelerate convergence when compared to full fine-tuning.

Recent studies show that the gap between LoRA and full fine-tuning can be mitigated by increasing the rank $r$ (Jiang et al., 2024; Huang et al., 2025). This motivates investigating high-rank updates with low-dimensional adapters. ReLoRA (Lialin et al., 2024) advocates learning a cascade of low-rank adapters and merging them sequentially into the pre-trained weights. However, learning each adapter requires restarting optimization, including random initialization, optimizer reset, and learning rate warm-up, which slows down convergence. MoRA (Jiang et al., 2024) replaces the two linear matrix multiplications $\mathbf{A}(\mathbf{B}^\top \mathbf{X})$ by nonlinear mappings $f_{\text{decompress}}(\mathbf{M} f_{\text{compress}}(\mathbf{X}))$ with learnable $\mathbf{M}$, while the two mappings demand careful handcrafted designs to ensure effective and stable fine-tuning. HiRA (Huang et al., 2025) parameterizes the weight update as the Hadamard product of low-rank matrix with pre-trained weight; i.e., $\mathbf{W}^{\text{ft}} := (\mathbf{A}\mathbf{B}^\top) \odot \mathbf{W}^{\text{pre}}$. Although this yields a high-rank update in Euclidean space, it remains confined to a smaller manifold of dimension $(m + n - r)r$, compared to full fine-tuning's $mn$-dimensional one. Moreover, HiRA demands explicit forward calculation and backpropagation through the $m \times n$ Hadamard product per iteration, which incurs $\mathcal{O}(mnr)$ complexity, and scales poorly to immense LLMs.

**Notation.** Bold lowercase letters (capitals) stand for vectors (matrices). $\mathbf{M}_{\mathcal{I}}$ represents the submatrix of $\mathbf{M}$ with columns indexed by set $\mathcal{I}$. Symbols $\odot$ and $\cdot^{\circ 2}$ stand for Hadamard (entry-wise) product and square. $\text{Row}(\cdot)$, $\text{Col}(\cdot)$, and $\text{Null}(\cdot)$ denote row, column and null spaces. $\text{rank}(\cdot)$ and $\text{tr}(\cdot)$ are the rank and trace of a matrix. $\text{diag}(\mathbf{v})$ is the diagonal matrix whose diagonal entries are from vector $\mathbf{v}$, while $\text{diag}(\mathbf{M})$ refers to the vector formed by the diagonals of matrix $\mathbf{M}$. $\cdot^\dagger$ denotes the Moore-Penrose pseudoinverse. For $\mathbf{M} \in \mathbb{R}^{m \times n}$, $\|\mathbf{M}\|_{\text{row}} \in \mathbb{R}^n$ defines the vector of row-wise norms; i.e., $[\|\mathbf{M}\|_{\text{row}}]_i = \|\mathbf{M}_{i,:}\|_2$. $\text{O}(r)$ refers to the orthogonal group of degree $r$; namely the set of all $r \times r$ orthogonal matrices. For readability, all proofs are deferred to Appendix A.

## 3 HIGH-RANK UPDATES VIA OPTIMAL SCALING

Unlike LoRA adhering to a fixed low-rank component $\mathbf{A}_t \mathbf{B}_t^\top$, the key idea of this work is to dynamically identify the "optimal" low-rank adapters per iteration that maximally descends the loss. By refining different low-dimensional subspaces over time, the cumulative increments effectively form a high-rank update, endowing LoRA with both improved effectiveness and faster convergence. Specifically, we will merge the current $\mathbf{A}_t \mathbf{B}_t^\top$ into $\mathbf{W}^{\text{pt}}$, and factor out an alternative low-rank matrix $\tilde{\mathbf{A}}_t \tilde{\mathbf{B}}_t^\top$ to optimize; that is,

$$\mathbf{W}_t = \mathbf{W}^{\text{pt}} + \mathbf{A}_t \mathbf{B}_t^\top = \underbrace{(\mathbf{W}^{\text{pt}} + \mathbf{A}_t \mathbf{B}_t^\top - \tilde{\mathbf{A}}_t \tilde{\mathbf{B}}_t^\top)}_{:=\tilde{\mathbf{W}}_t^{\text{pt}}, \text{ merge \& freeze}} + \underbrace{\tilde{\mathbf{A}}_t \tilde{\mathbf{B}}_t^\top}_{:=\tilde{\mathbf{W}}_t^{\text{ft}}, \text{ learnable}}. \tag{3}$$

The optimal choice of $\tilde{\mathbf{A}}_t \tilde{\mathbf{B}}_t^\top$ will be presented in the next subsection. Before that, we first illustrate how this change in the optimization direction influences the optimization dynamics to produce a high-rank update. With the alternative adapters $(\tilde{\mathbf{A}}_t, \tilde{\mathbf{B}}_t)$, the GD update (1) can be replaced by

$$\mathbf{A}_{t+1} = \tilde{\mathbf{A}}_t - \eta \nabla \ell(\mathbf{W}_t) \tilde{\mathbf{B}}_t, \quad \mathbf{B}_{t+1} = \tilde{\mathbf{B}}_t - \eta \nabla \ell(\mathbf{W}_t)^\top \tilde{\mathbf{A}}_t. \tag{4}$$

In doing so, the resultant update $\Delta \tilde{\mathbf{W}}_t$ to weight matrix $\mathbf{W}_t$, and the corresponding dynamics are

$$\Delta \tilde{\mathbf{W}}_t = \mathbf{A}_{t+1} \mathbf{B}_{t+1}^\top - \tilde{\mathbf{A}}_t \tilde{\mathbf{B}}_t^\top = -\eta \nabla \ell(\mathbf{W}_t) \tilde{\mathbf{B}}_t \tilde{\mathbf{B}}_t^\top - \eta \tilde{\mathbf{A}}_t \tilde{\mathbf{A}}_t^\top \nabla \ell(\mathbf{W}_t) + \mathcal{O}(\eta^2), \tag{5a}$$

$$\sum_{t=0}^{T-1} \Delta \tilde{\mathbf{W}}_t = \sum_{t=1}^{T} \mathbf{A}_t \mathbf{B}_t^\top - \sum_{t=0}^{T-1} \tilde{\mathbf{A}}_t \tilde{\mathbf{B}}_t^\top. \tag{5b}$$

By optimizing different low-rank matrices per iteration, the telescoping in (2) is avoided, thus allowing to accumulate the low-rank increments to render a high-rank update.

Although ReLoRA (Lialin et al., 2024) also employs a similar merging strategy, it performs this operation less frequently due to its optimization restarts, and simply reinitializes $\tilde{\mathbf{A}}_t \tilde{\mathbf{B}}_t^\top = 0$ without a principled selection. Next, we analyze the optimal selection and the associated challenges.

## 3.1 CHALLENGES IN ACCUMULATING LOW-RANK UPDATES

Though promising, this idea of accumulating low-rank updates faces two major challenges, namely prohibitive computation and inefficient restart, which are separately elaborated next.

We start by characterizing the optimal low-rank adapters and their computational complexity. Due to the nonlinearity of LLMs, the global optimum of the loss function is analytically infeasible. As a tractable alternative, a standard upper bound on the loss function will be presented, whose minimizer is available in closed form. The analysis relies on the following Lipschitz smoothness assumption.

**Assumption 1.** *The loss function $\ell$ has $L$-Lipschitz continuous gradients; i.e., $\|\nabla\ell(\mathbf{W}) - \nabla\ell(\mathbf{W}')\|_F \leq L\|\mathbf{W} - \mathbf{W}'\|_F, \ \forall \mathbf{W}, \mathbf{W}' \in \mathbb{R}^{m \times n}$.*

Assumption 1 is fairly mild and widely used in both machine learning (Goodfellow et al., 2016; Shalev-Shwartz & Ben-David, 2014), and optimization (Bertsekas, 2016; Kingma & Ba, 2015). It is default for analyzing first-order optimization approaches such as (stochastic) GD. Building upon this assumption, the loss function admits the quadratic upper bound as follows

$$\ell(\mathbf{W}_t + \Delta\mathbf{W}_t) \leq \ell(\mathbf{W}_t) + \langle\nabla\ell(\mathbf{W}_t), \Delta\mathbf{W}_t\rangle_F + \frac{L}{2}\|\Delta\mathbf{W}_t\|_F^2. \tag{6}$$

Minimizing the right-hand side of (6) incurs optimal update $\Delta\mathbf{W}_t^* = -\frac{1}{L}\nabla\ell(\mathbf{W}_t)$, which recovers GD of full fine-tuning. While the Lipschitz constant $L$ is hard to compute or even estimate especially for complicated LLMs, the effective step size $1/L$ is typically treated as a hyperparameter and tuned via grid search. Likewise, it holds for the alternative update (4) that

$$\ell(\mathbf{W}_t + \Delta\tilde{\mathbf{W}}_t) \leq \ell(\mathbf{W}_t) + \langle\nabla\ell(\mathbf{W}_t), \Delta\tilde{\mathbf{W}}_t\rangle_F + \frac{L}{2}\|\Delta\tilde{\mathbf{W}}_t\|_F^2 \overset{(a)}{=} \frac{L}{2}\|\Delta\mathbf{W}_t^* - \Delta\tilde{\mathbf{W}}_t\|_F^2 + \text{Const.}$$

where $(a)$ utilizes completing the square, and $\text{Const.}$ refers to constants not dependent on $\Delta\tilde{\mathbf{W}}_t$. This reformulation reveals that minimizing the loss upper bound is equivalent to aligning LoRA's weight increment with full fine-tuning. Plugging in (5a) and omitting high-order terms yield

$$\min_{\tilde{\mathbf{A}}_t, \tilde{\mathbf{B}}_t} \frac{L}{2}\left\|\frac{1}{L}\nabla\ell(\mathbf{W}_t) - \eta\nabla\ell(\mathbf{W}_t)\tilde{\mathbf{B}}_t\tilde{\mathbf{B}}_t^\top - \eta\tilde{\mathbf{A}}_t\tilde{\mathbf{A}}_t^\top\nabla\ell(\mathbf{W}_t)\right\|_F^2 \tag{7}$$

whose minimizer is offered in the following theorem.

**Theorem 1.** *Consider the SVD $\nabla\ell(\mathbf{W}_t) = \mathbf{U}_t\boldsymbol{\Sigma}_t\mathbf{V}_t^\top$. If $\text{rank}(\nabla\ell(\mathbf{W}_t)) \geq 2r, \ \forall t$ and Assumption 1 holds, then $(\tilde{\mathbf{A}}_t^*, \tilde{\mathbf{B}}_t^*)$ minimizes (7) if and only if*

$$\tilde{\mathbf{A}}_t^* = \frac{1}{\sqrt{L\eta}}[\mathbf{U}_t]_{\mathcal{A}_t}\mathbf{P}_t, \ \ \tilde{\mathbf{B}}_t^* = \frac{1}{\sqrt{L\eta}}[\mathbf{V}_t]_{\mathcal{B}_t}\mathbf{Q}_t \tag{8}$$

*where sets $\mathcal{A}_t \cup \mathcal{B}_t = \{1, \ldots, 2r\}$, $|\mathcal{A}_t| = |\mathcal{B}_t| = r$, and $\mathbf{P}_t, \mathbf{Q}_t \in \mathrm{O}(r)$.*

Theorem 1 establishes a sufficient and necessary condition for the optimal low-rank adapters. The optimal choice involves the truncated rank-$2r$ SVD of $\nabla\ell(\mathbf{W}_t)$, which prompts an iterative solver and incurs $\mathcal{O}(Smnr)$ time complexity, with $S$ denoting the number of iterations (Baglama & Reichel, 2005). Due to this prohibitively high complexity, it is generally infeasible to apply such a choice to (4) for each $t$. It is worthwhile mentioning that LoRA-GA (Wang et al., 2024) arises as a special case of Theorem 1, where a sufficient (yet not necessary) condition is derived at $t = 0$ and $\mathbf{P}_0 = \mathbf{Q}_0 = \mathbf{I}_r$ to initialize LoRA adapters. Moreover, the assumption $\text{rank}(\nabla\ell(\mathbf{W}_t)) \geq 2r, \ \forall t$ can be readily satisfied in practice; see numerical validations in Figure 2c of Section 4.

Aside from the prohibitive SVD computation, another challenge attributes to switching the optimization variables from $(\mathbf{A}_t, \mathbf{B}_t)$ to $(\tilde{\mathbf{A}}_t, \tilde{\mathbf{B}}_t)$. Specifically, LLM optimization relies on adaptive optimizers such as AdamW (Loshchilov & Hutter, 2019), which estimate the first and second moments of stochastic gradients via the exponential moving average of gradient samples; cf. Appendix A.4. When switching to the alternative $(\tilde{\mathbf{A}}_t, \tilde{\mathbf{B}}_t)$, their gradient moments need to be re-estimated from the optimization trajectory, incurring time and space complexities proportional to $t$. One straightforward remedy is to restart optimization (Lialin et al., 2024), which resets the moment estimators to accumulate them from scratch. However, as all gradient statistics are discarded, the optimization breaks off and the convergence slows down considerably.

To enable efficient and seamless optimization, we propose to restrict $\tilde{\mathbf{A}}_t$ and $\tilde{\mathbf{B}}_t$ to be structured transformations of $\mathbf{A}_t$ and $\mathbf{B}_t$. Upon appropriate design, the gradient moment estimators of the former can be equivariantly computed from those of the latter.

## 3.2 OPTIMAL SCALAR SCALING

We will first investigate a simple scalar scaling $\tilde{\mathbf{A}}_t = \alpha_t \mathbf{A}_t$, $\tilde{\mathbf{B}}_t = \beta_t \mathbf{B}_t$. Let $m_t(\cdot)$ and $v_t(\cdot)$ denote the first and second gradient moment estimators, which involve the general stochastic matrices $\mathbf{A}$, $\mathbf{B}$ and $\mathbf{W}$; see Appendix A.4 for details. The next lemma depicts the impact of scalar scaling on the gradient moment estimators.

**Lemma 2.** *For $\mathbf{W} = \mathbf{W}^{\mathrm{pt}} + \mathbf{A}\mathbf{B}^\top = \tilde{\mathbf{W}}^{\mathrm{pt}} + \tilde{\mathbf{A}}\tilde{\mathbf{B}}^\top$ with $\tilde{\mathbf{A}} = \alpha\mathbf{A}$ and $\tilde{\mathbf{B}} = \beta\mathbf{B}$, it holds that*

$$m_t(\nabla_{\tilde{\mathbf{A}}}\ell(\mathbf{W})) = \beta m_t(\nabla_{\mathbf{A}}\ell(\mathbf{W})), \; v_t(\nabla_{\tilde{\mathbf{A}}}\ell(\mathbf{W})) = \beta^2 v_t(\nabla_{\mathbf{A}}\ell(\mathbf{W})),$$

$$m_t(\nabla_{\tilde{\mathbf{B}}}\ell(\mathbf{W})) = \alpha m_t(\nabla_{\mathbf{B}}\ell(\mathbf{W})), \; v_t(\nabla_{\tilde{\mathbf{B}}}\ell(\mathbf{W})) = \alpha^2 v_t(\nabla_{\mathbf{B}}\ell(\mathbf{W})).$$

Lemma 2 suggests that the first and second moment estimators of $(\tilde{\mathbf{A}}, \tilde{\mathbf{B}})$ can be directly scaled from those of $(\mathbf{A}, \mathbf{B})$. Intuitively, given that the gradient of $\tilde{\mathbf{A}}$ is $\nabla\ell(\mathbf{W})\tilde{\mathbf{B}}$, it is hence scaled by $\beta$ proportionally when transforming $\tilde{\mathbf{B}} = \beta\mathbf{B}$; similar statements hold for $\mathbf{B}$'s gradient.

We now seek the optimal $(\tilde{\mathbf{A}}_t, \tilde{\mathbf{B}}_t)$ minimizing the loss upper bound. Under the aforementioned transform, the objective function (7) reduces to

$$\min_{\alpha_t, \beta_t} \frac{L}{2} \left\| \frac{1}{L}\nabla\ell(\mathbf{W}_t) - \eta\beta_t^2 \nabla\ell(\mathbf{W}_t)\mathbf{B}_t\mathbf{B}_t^\top - \eta\alpha_t^2 \mathbf{A}_t\mathbf{A}_t^\top\nabla\ell(\mathbf{W}_t) \right\|_{\mathrm{F}}^2. \tag{9}$$

To solve for the global minimizer of (9), the following technical assumption is adopted.

**Assumption 2.** $\|\mathbf{A}_t^\top\nabla\ell(\mathbf{W}_t)\|_{\mathrm{F}}$ *and* $\|\nabla\ell(\mathbf{W}_t)\mathbf{B}_t\|_{\mathrm{F}}$ *are not both 0, $\forall t$.*

Assumption 2 asserts that the gradients of $\mathbf{A}_t$ and $\mathbf{B}_t$ do not vanish simultaneously; otherwise there is no update, and the iteration can be skipped. With this assumption, the optimal scaling factors are derived as follows.

**Theorem 3.** *With Assumptions 1-2 in effect, the global minimizer of* (9) *is given by*

$$(\alpha_t^*, \beta_t^*) = \begin{cases} \left( \pm \frac{\|\mathbf{A}_t^\top\nabla\ell(\mathbf{W}_t)\|_{\mathrm{F}}}{\sqrt{L\eta}\|\mathbf{A}_t\mathbf{A}_t^\top\nabla\ell(\mathbf{W}_t)\|_{\mathrm{F}}}, 0 \right), & \text{if } C_t^A > 0 \text{ and } C_t^B \le 0, \text{ or } C_t = 0 \text{ and } \mathbf{A}_t \ne \mathbf{0} \\ \left( 0, \pm \frac{\|\nabla\ell(\mathbf{W}_t)\mathbf{B}_t\|_{\mathrm{F}}}{\sqrt{L\eta}\|\nabla\ell(\mathbf{W}_t)\mathbf{B}_t\mathbf{B}_t^\top\|_{\mathrm{F}}} \right), & \text{if } C_t^A \le 0 \text{ and } C_t^B > 0, \text{ or } C_t = 0 \text{ and } \mathbf{B}_t \ne \mathbf{0} \\ \left( \pm \sqrt{\frac{C_t^A}{L\eta C_t}}, \pm \sqrt{\frac{C_t^B}{L\eta C_t}} \right), & \text{if } C_t^A \ge 0, \; C_t^B \ge 0 \text{ and } C_t > 0 \end{cases}$$

*where we define*

$$C_t^A := \|\mathbf{A}_t^\top\nabla\ell(\mathbf{W}_t)\|_{\mathrm{F}}^2\|\nabla\ell(\mathbf{W}_t)\mathbf{B}_t\mathbf{B}_t^\top\|_{\mathrm{F}}^2 - \|\nabla\ell(\mathbf{W}_t)\mathbf{B}_t\|_{\mathrm{F}}^2\|\mathbf{A}_t^\top\nabla\ell(\mathbf{W}_t)\mathbf{B}_t\|_{\mathrm{F}}^2,$$

$$C_t^B := \|\nabla\ell(\mathbf{W}_t)\mathbf{B}_t\|_{\mathrm{F}}^2\|\mathbf{A}_t\mathbf{A}_t^\top\nabla\ell(\mathbf{W}_t)\|_{\mathrm{F}}^2 - \|\mathbf{A}_t^\top\nabla\ell(\mathbf{W}_t)\|_{\mathrm{F}}^2\|\mathbf{A}_t^\top\nabla\ell(\mathbf{W}_t)\mathbf{B}_t\|_{\mathrm{F}}^2,$$

$$C_t := \|\mathbf{A}_t\mathbf{A}_t^\top\nabla\ell(\mathbf{W}_t)\|_{\mathrm{F}}^2\|\nabla\ell(\mathbf{W}_t)\mathbf{B}_t\mathbf{B}_t^\top\|_{\mathrm{F}}^2 - \|\mathbf{A}_t^\top\nabla\ell(\mathbf{W}_t)\mathbf{B}_t\|_{\mathrm{F}}^4.$$

Note that the three cases in Theorem 3 may overlap, because the global optima can be non-unique. Moreover, all possible scenarios are covered by the three cases; cf. Appendix A.2.

## 3.3 OPTIMAL COLUMN-WISE SCALING

For improved fitting capacity, this section delves into a more complicated column-wise scaling with $\tilde{\mathbf{A}}_t = \mathbf{A}_t \operatorname{diag}(\boldsymbol{\alpha}_t)$ and $\tilde{\mathbf{B}}_t = \mathbf{B}_t \operatorname{diag}(\boldsymbol{\beta}_t)$, whose gradient moment estimators are provided next.

**Lemma 4.** *For $\mathbf{W} = \mathbf{W}^{\mathrm{pt}} + \mathbf{A}\mathbf{B}^\top = \tilde{\mathbf{W}}^{\mathrm{pt}} + \tilde{\mathbf{A}}\tilde{\mathbf{B}}^\top$ with $\tilde{\mathbf{A}} = \mathbf{A}\operatorname{diag}(\boldsymbol{\alpha})$ and $\tilde{\mathbf{B}} = \mathbf{B}\operatorname{diag}(\boldsymbol{\beta})$,*

$$m_t(\nabla_{\tilde{\mathbf{A}}}\ell(\mathbf{W})) = m_t(\nabla_{\mathbf{A}}\ell(\mathbf{W}))\operatorname{diag}(\boldsymbol{\beta}), \; v_t(\nabla_{\tilde{\mathbf{A}}}\ell(\mathbf{W})) = v_t(\nabla_{\mathbf{A}}\ell(\mathbf{W}))\operatorname{diag}^2(\boldsymbol{\beta}),$$

$$m_t(\nabla_{\tilde{\mathbf{B}}}\ell(\mathbf{W})) = m_t(\nabla_{\mathbf{B}}\ell(\mathbf{W}))\operatorname{diag}(\boldsymbol{\alpha}), \; v_t(\nabla_{\tilde{\mathbf{B}}}\ell(\mathbf{W})) = v_t(\nabla_{\mathbf{B}}\ell(\mathbf{W}))\operatorname{diag}^2(\boldsymbol{\alpha}).$$

Unlike column-wise scaling, moment estimators for transformations including row-wise scaling and left/right-multiplying a full matrix, are generally intractable.

With column-wise scaling on he other hand, the objective function (7) boils down to

$$\min_{\boldsymbol{\alpha}_t, \boldsymbol{\beta}_t} \frac{L}{2} \left\| \frac{1}{L}\nabla\ell(\mathbf{W}_t) - \eta\nabla\ell(\mathbf{W}_t)\mathbf{B}_t\operatorname{diag}^2(\boldsymbol{\beta}_t)\mathbf{B}_t^\top - \eta\mathbf{A}_t\operatorname{diag}^2(\boldsymbol{\alpha}_t)\mathbf{A}_t^\top\nabla\ell(\mathbf{W}_t) \right\|_{\mathrm{F}}^2. \tag{10}$$

Different from the scalar case in (9), Appendix A.3 shows that (10) has $\mathcal{O}(9^r)$ stationary points, among which the global optimum is generally hard to obtain in affordable time. Nevertheless, under certain conditions the optimum can be efficiently obtained through a $2r \times 2r$ linear system.

**Theorem 5.** *With the definitions*

$$\mathbf{S}_t^A := \begin{bmatrix} \mathbf{A}_t & \nabla\ell(\mathbf{W}_t)\mathbf{B}_t \end{bmatrix}, \ \mathbf{S}_t^B := \begin{bmatrix} \mathbf{B}_t & \mathbf{A}_t^\top \nabla\ell(\mathbf{W}_t) \end{bmatrix}, \boldsymbol{\lambda}_t := \begin{bmatrix} \|\mathbf{A}_t^\top \nabla\ell(\mathbf{W}_t)\|_{\text{row}}^2 \\ \|\mathbf{B}_t^\top \nabla\ell(\mathbf{W}_t)^\top\|_{\text{row}}^2 \end{bmatrix}$$

*and Assumptions 1-2 in effect, if the linear system of equations* $\left[(\mathbf{S}_t^{A\top}\mathbf{S}_t^A) \odot (\mathbf{S}_t^{B\top}\mathbf{S}_t^B)\right]\mathbf{v}_t = \boldsymbol{\lambda}_t$ *has a non-negative solution* $\mathbf{v}_t \in \mathbb{R}_+^{2r}$*, then the global minimizer of* (10) *is given by*

$$\begin{bmatrix} \boldsymbol{\alpha}_t^* \\ \boldsymbol{\beta}_t^* \end{bmatrix} = \pm\frac{1}{\sqrt{L\eta}}\mathbf{v}_t^{\circ\frac{1}{2}}. \tag{11}$$

Interestingly, our empirical observations suggest that around $80\%$ LoRA layers in an LLM satisfies the the non-negativity condition for $\mathbf{v}_t$ across iterations; see Figure 2d.

### 3.4 ScaLoRA FOR HIGH-RANK UPDATE AND FAST CONVERGENCE

Building upon these analytical insights, our scaled low-rank adaptation (ScaLoRA) method optimally scales the low-rank adapters per (few) iteration(s) to attain the desired high-rank update and fast convergence. In particular, ScaLoRA relies on a mixture of the aforementioned two scaling schemes. When the linear system in Theorem 5 yields a positive solution, (3) adopts the optimal column-wise scaling $\tilde{\mathbf{A}}_t = \mathbf{A}_t \operatorname{diag}(\boldsymbol{\alpha}_t^*)$, $\tilde{\mathbf{B}}_t = \mathbf{B}_t \operatorname{diag}(\boldsymbol{\beta}_t^*)$, with moment estimators updated as in Lemma 4; otherwise, the algorithm resorts to Theorem 3 for the optimal scalar scaling $\tilde{\mathbf{A}}_t = \alpha_t^*\mathbf{A}_t$, $\tilde{\mathbf{B}}_t = \beta_t^*\mathbf{B}_t$, and Lemma 2 to update moment estimators. Akin to full fine-tuning, the Lipschitz constant $L$ is viewed as a hyperparameter and we tune it using grid search. The step-by-step pseudocodes are provided in Appendix B.

Next, we analyze the computational cost of ScaLoRA, and compare it to SOTA approaches. To start, the gradients $\nabla\ell(\mathbf{W}_t)\mathbf{B}_t$ and $\nabla\ell(\mathbf{W}_t)^\top\mathbf{A}_t$ can be directly acquired from backpropagation, that incurs no extra overhead. As a consequence, the overall time complexity for ScaLoRA is $\mathcal{O}(mnr+(m+n+r)r^2)$, where the term $\mathcal{O}(mnr)$ comes from (3), and the rest can be deduced from Theorems 3 and 5. When $r \ll m, n$, the time complexity is dominated by the former. Moreover, as (3) can be performed in place, the space overhead is as small as $\mathcal{O}((m+n+r)r)$. In comparison, MoRA's overhead significantly depends on the design of $f_{\text{compress}}$ and $f_{\text{decompress}}$, which typically exceeds LoRA's simple bilinear structure. While HiRA exhibits $\mathcal{O}(mnr)$ time overhead comparable to ScaLoRA, it suffers from high memory footprint of $\mathcal{O}(mn)$ due to the backpropagation of Hadamard product.

Similar to other high-rank update approaches, the escalated computational cost is the major limitation of ScaLoRA, which confines its scalability to increasingly large models. We next introduce a variant to mitigate this limitation. Since $\eta$ is typically tiny, the optimal scaling is close to 1 after one update; cf. Appendix D.1. Thus, a natural remedy is to perform ScaLoRA every $I$ iterations, so that the per-step time complexity is amortized to $\mathcal{O}((mnr + (m+n+r)r^2)/I)$ without noticeably exacerbating the performance. We term this intermittent variant as ScaLoRA-I. It is worth stressing that MoRA and HiRA both rely on a fixed structure to impel a high-rank update, which is imposed per optimization step, and cannot be amortized. A summary of the costs is provided in Appendix B, and numerical comparisons using LLMs are presented in Section 4.3.

Another notable limitation of ScaLoRA is its storage. While LoRA and other high-rank variants require saving only the low-dimensional adapters $\mathbf{A}_t$ and $\mathbf{B}_t$, ScaLoRA stores the entire merged matrix $\mathbf{W}_t = \tilde{\mathbf{W}}_t^{\text{pt}} + \tilde{\mathbf{A}}_t\tilde{\mathbf{B}}_t^\top$ due to the modification of $\tilde{\mathbf{W}}_t^{\text{pt}}$. Fortunately, disk space is typically abundant relative to memory, and thereby it does not pose a bottleneck for LLM fine-tuning.

## 4 NUMERICAL TESTS

This section presents numerical tests to validate the effectiveness of the proposed ScaLoRA approach. All setups including datasets, models, and hyperparameters are deferred to Appendix C.

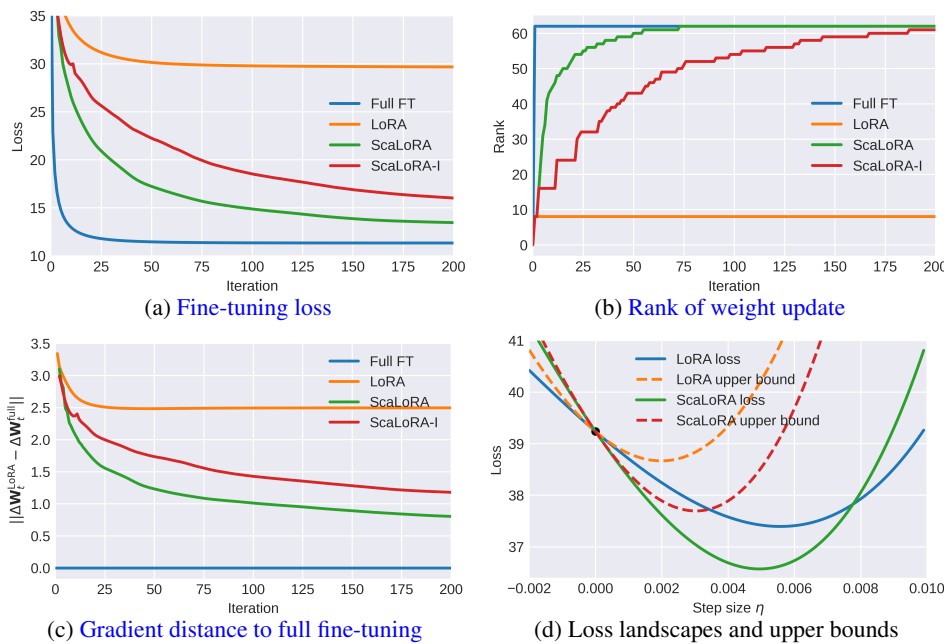

(a) Fine-tuning loss

(b) Rank of weight update

(c) Gradient distance to full fine-tuning

(d) Loss landscapes and upper bounds

Figure 1: Visualization of linear regression on synthetic data.

Table 1: Comparison using DeBERTaV3-base on the GLUE benchmark. The top two results are marked with solid lines and underlines. The results for LoRA approaches are obtained by averaging 3 random runs with $r = 4$, and the full fine-tuning results are from (Zhang et al., 2023).

| Method | CoLA | SST-2 | MRPC | STS-B | QQP | MNLI | QNLI | RTE | All |
|---|---|---|---|---|---|---|---|---|---|
| | Mcc | Acc | Acc | Corr | Acc | Matched | Acc | Acc | Avg |
| Full FT | 69.19 | 95.63 | 89.46 | 91.60 | 92.40 | 89.90 | 94.03 | 83.75 | 88.25 |
| LoRA | $68.10_{\pm1.73}$ | $95.49_{\pm0.05}$ | $89.46_{\pm0.20}$ | $91.09_{\pm0.14}$ | $91.86_{\pm0.03}$ | $90.25_{\pm0.13}$ | $94.30_{\pm0.05}$ | $84.48_{\pm2.04}$ | 88.13 |
| MoRA | $69.67_{\pm0.90}$ | $95.45_{\pm0.44}$ | $89.62_{\pm0.76}$ | $90.90_{\pm0.19}$ | $91.83_{\pm0.12}$ | $90.05_{\pm0.04}$ | $93.81_{\pm0.20}$ | $85.44_{\pm1.19}$ | 88.35 |
| HiRA | $68.82_{\pm1.01}$ | $95.53_{\pm0.19}$ | $89.95_{\pm0.53}$ | $91.15_{\pm0.09}$ | $92.19_{\pm0.06}$ | $90.24_{\pm0.10}$ | $94.15_{\pm0.13}$ | $85.68_{\pm0.17}$ | 88.46 |
| ScaLoRA | $69.86_{\pm0.37}$ | $95.83_{\pm0.29}$ | $90.28_{\pm0.31}$ | $91.47_{\pm0.15}$ | $92.10_{\pm0.07}$ | $90.36_{\pm0.03}$ | $94.34_{\pm0.28}$ | $87.61_{\pm0.34}$ | 88.98 |

## 4.1 LINEAR REGRESSION WITH SYNTHETIC DATA

The first experiment performs linear regression on toy data. The loss function is $\ell(\mathbf{W}) = \frac{1}{2}\|\mathbf{Y} - \mathbf{W}\mathbf{X}\|_F^2$, where $\mathbf{X}$ and $\mathbf{Y}$ are given matrices. LoRA substitutes $\mathbf{W} \in \mathbb{R}^{64 \times 64}$ with $\mathbf{A}\mathbf{B}^\top$. Figure 1 sketches the behavior of LoRA, ScaLoRA(-I), and full fine-tuning. It is seen that ScaLoRA(-I) converges remarkably faster than vanilla LoRA, thanks to the progressively increasing rank of cumulative weight updates, and better alignment to full fine-tuning. In addition, Figure 1d depicts the loss function, and its quadratic upper bound (6). By selecting the optimal per-step LoRA adapters, ScaLoRA minimizes the loss upper bound and the associated loss landscape, leading to accelerated convergence. These observations corroborate our theoretical results in Section 3.

## 4.2 NATURAL LANGUAGE UNDERSTANDING

The next test deals with ScaLoRA's performance on General Language Understanding Evaluation (GLUE) benchmark (Wang et al., 2019), which contains 8 different tasks in the field of natural language understanding (NLU). The model is DeBERTaV3-base (He et al., 2023), a masked language model specialized in NLU with 184M parameters. The rank in LoRA is fixed to $r = 4$ with scaling coefficient 8 for all approaches, and other setups follow from (Zhang et al., 2023). Table 1 compares ScaLoRA to LoRA (Hu et al., 2022), and SOTA high-rank variants MoRA (Jiang et al., 2024) and HiRA (Huang et al., 2025), where the top two results are marked in bold and underlined. Notably, ScaLoRA not only presents $0.5\%+$ average performance gain, but also achieves the best perfor-

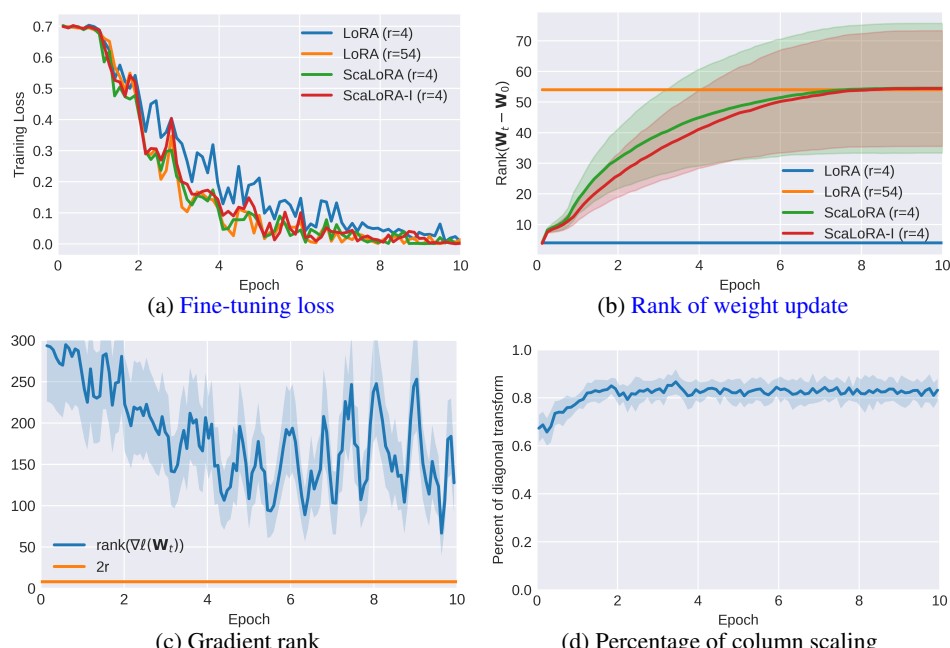

Figure 2: Visualization on the RTE dataset with DeBERTaV3-base.

mance in 7 out of 8 datasets, and exhibits comparable performance ($0.09\%$ less than the highest) on the remaining one. We remark that the GLUE datasets are relatively small, so that full fine-tuning can readily lead to overfitting, and hence inferior performance.

To further investigate the rationale behind ScaLoRA's performance gain, Figures 2a and 2b outline the fine-tuning loss and rank of cumulative weight update for LoRA and ScaLoRA(-I) on the RTE dataset of the GLUE benchmark. MoRA and HiRA are excluded since they rely on different learning rates. Clearly, ScaLoRA gradually accumulates the low-rank update during the fine-tuning epochs, rendering weight updates of average rank $54$. Due to this high-rank update, ScaLoRA's convergence is markedly faster than LoRA with $r = 4$, and aligns with LoRA for $r = 54$ especially in the last 5 epochs. This highlights the high-rank update and fast convergence incurred by ScaLoRA. Interestingly, the increase of rank in Figure 2b becomes slower with epochs. This is because ScaLoRA's direct objective is to minimize the loss, which allows each layer to adaptively adjust the singular values in the most effective directions. When the previous weight updates span a sufficiently large subspace that the new weight increment falls into, the rank stops growing. This in turn confirms LoRA's premise that the optimal weight update lives on a low-rank manifold. In addition, Figures 2c and 2d respectively justify the assumption $\text{rank}(\nabla \ell(\mathbf{W}_t)) \geq 2r$, $\forall t$ in Theorem 1, and the condition $\mathbf{v}_t \in \mathbb{R}_+^{2r}$ in Theorem 5. As the NLU tasks in GLUE are relatively simple and the RTE dataset is small, a low rank of $54$ suffices to fit well the datasets. Next, experiments are conducted on a suite of more challenging tasks with larger LLMs, where higher ranks become necessary.

## 4.3 COMMONSENSE REASONING

Beyond the NLU tasks, further tests are conducted on commonsense reasoning tasks with LLMs including LLaMA2-7B (Touvron et al., 2023) and LLaMA3-8B (Grattafiori et al., 2024). With the LLM size growing, computational cost becomes a bottleneck for fine-tuning. Thus, the intermittent variant ScaLoRA-I with $I = 10$ is also included in the test. The experimental setups follow from (Lion et al., 2025), where LLMs are fine-tuned separately on each dataset, and subsequently evaluated for multiple-choice log-likelihood under the widely-adopted lm-evaluation-harness framework (Gao et al., 2024a). To underscore the importance of high-rank updates for challenging tasks, we restrict the fitting capacity of LoRA and its variants by setting $r = 8$ throughout the test. This setup is intended to emulate more challenging scenarios where higher ranks are necessitated to capture the underlying task structure. Compared to the common choice $r = 32$, this low-rank configuration leads to consistently degraded performance across

Table 2: Commonsense reasoning using LLaMA2-7B and LLaMA3-8B with $r = 8$. The top two results are marked with solid lines and underlines.

| | Method | BoolQ | PIQA | SIQA | HS | WG | ARC-e | ARC-c | OBQA | Avg |
|---|---|---|---|---|---|---|---|---|---|---|
| **LLaMA2-7B** | LoRA | $87.40_{\pm0.58}$ | $81.66_{\pm0.90}$ | $59.16_{\pm1.11}$ | $82.45_{\pm0.38}$ | $79.48_{\pm1.14}$ | $82.91_{\pm0.77}$ | $57.59_{\pm1.44}$ | $58.40_{\pm2.21}$ | 73.63 |
| | ReLoRA | $\underline{87.80}_{\pm0.57}$ | $82.48_{\pm0.89}$ | $60.08_{\pm1.11}$ | $83.23_{\pm0.37}$ | $\underline{82.56}_{\pm1.07}$ | $82.95_{\pm0.77}$ | $\underline{58.11}_{\pm1.44}$ | $58.00_{\pm2.20}$ | 74.40 |
| | LoRA-GA | $\mathbf{87.92}_{\pm0.58}$ | $\mathbf{83.03}_{\pm0.88}$ | $\underline{60.13}_{\pm1.11}$ | $83.30_{\pm0.38}$ | $\mathbf{82.87}_{\pm1.09}$ | $83.25_{\pm0.77}$ | $56.83_{\pm1.44}$ | $58.40_{\pm2.21}$ | 74.34 |
| | MoRA | $87.49_{\pm0.58}$ | $82.54_{\pm0.89}$ | $59.88_{\pm1.11}$ | $82.56_{\pm0.38}$ | $79.08_{\pm1.14}$ | $\underline{83.59}_{\pm0.76}$ | $58.02_{\pm1.44}$ | $57.40_{\pm2.21}$ | 73.82 |
| | HiRA | $87.71_{\pm0.57}$ | $\underline{82.97}_{\pm0.88}$ | $59.83_{\pm1.11}$ | $83.38_{\pm0.37}$ | $81.69_{\pm1.09}$ | $82.83_{\pm0.77}$ | $55.55_{\pm1.45}$ | $57.60_{\pm2.21}$ | 73.95 |
| | ScaLoRA | $87.77_{\pm0.57}$ | $82.43_{\pm0.88}$ | $60.08_{\pm1.11}$ | $\underline{83.43}_{\pm0.37}$ | $82.08_{\pm1.08}$ | $83.54_{\pm0.76}$ | $\underline{58.11}_{\pm1.44}$ | $\underline{58.60}_{\pm2.20}$ | $\underline{74.51}$ |
| | ScaLoRA-I | $87.58_{\pm0.76}$ | $82.26_{\pm0.89}$ | $\mathbf{60.49}_{\pm1.11}$ | $\mathbf{83.52}_{\pm0.37}$ | $81.69_{\pm1.09}$ | $\mathbf{83.75}_{\pm0.76}$ | $\mathbf{58.53}_{\pm1.44}$ | $\mathbf{60.20}_{\pm1.19}$ | $\mathbf{74.75}$ |
| | LoRA$_{r=32}$ | $88.29_{\pm0.56}$ | $82.70_{\pm0.90}$ | $60.54_{\pm1.11}$ | $83.15_{\pm0.37}$ | $82.00_{\pm1.08}$ | $82.79_{\pm0.77}$ | $57.68_{\pm1.44}$ | $59.00_{\pm2.20}$ | 74.52 |
| **LLaMA3-8B** | LoRA | $88.99_{\pm0.55}$ | $85.09_{\pm0.83}$ | $60.95_{\pm1.10}$ | $86.09_{\pm0.35}$ | $82.64_{\pm1.06}$ | $86.62_{\pm0.70}$ | $62.29_{\pm1.42}$ | $62.00_{\pm2.17}$ | 76.83 |
| | ReLoRA | $\underline{89.20}_{\pm0.54}$ | $85.64_{\pm0.82}$ | $60.13_{\pm1.11}$ | $85.99_{\pm0.35}$ | $\underline{85.24}_{\pm1.00}$ | $86.95_{\pm0.69}$ | $63.14_{\pm1.39}$ | $61.80_{\pm2.19}$ | 77.26 |
| | LoRA-GA | $\mathbf{89.69}_{\pm0.53}$ | $84.98_{\pm0.83}$ | $61.00_{\pm0.96}$ | $\underline{86.58}_{\pm0.96}$ | $\mathbf{85.32}_{\pm0.99}$ | $86.11_{\pm0.71}$ | $62.29_{\pm1.42}$ | $61.80_{\pm2.18}$ | 77.22 |
| | MoRA | $88.56_{\pm0.56}$ | $\mathbf{86.18}_{\pm0.81}$ | $60.29_{\pm1.11}$ | $\mathbf{86.69}_{\pm0.34}$ | $82.40_{\pm1.07}$ | $\mathbf{87.79}_{\pm0.67}$ | $64.08_{\pm1.40}$ | $62.20_{\pm2.17}$ | 77.27 |
| | HiRA | $88.87_{\pm0.55}$ | $86.07_{\pm0.81}$ | $60.64_{\pm1.11}$ | $86.11_{\pm0.35}$ | $84.53_{\pm1.02}$ | $\underline{87.12}_{\pm0.69}$ | $63.91_{\pm1.40}$ | $\mathbf{62.40}_{\pm2.17}$ | 77.46 |
| | ScaLoRA | $\underline{89.20}_{\pm0.54}$ | $\mathbf{86.18}_{\pm0.81}$ | $\underline{61.82}_{\pm1.10}$ | $86.51_{\pm0.34}$ | $84.53_{\pm1.02}$ | $86.57_{\pm0.70}$ | $\mathbf{65.61}_{\pm1.39}$ | $\mathbf{62.40}_{\pm2.17}$ | $\mathbf{77.85}$ |
| | ScaLoRA-I | $89.14_{\pm0.54}$ | $86.07_{\pm0.81}$ | $\mathbf{62.33}_{\pm1.10}$ | $86.48_{\pm0.34}$ | $83.35_{\pm1.05}$ | $86.53_{\pm0.70}$ | $\underline{64.68}_{\pm0.70}$ | $62.00_{\pm0.70}$ | $\underline{77.57}$ |
| | LoRA$_{r=32}$ | $89.69_{\pm0.53}$ | $85.47_{\pm0.82}$ | $61.72_{\pm1.10}$ | $86.76_{\pm0.34}$ | $83.35_{\pm1.05}$ | $87.08_{\pm0.69}$ | $64.08_{\pm1.40}$ | $62.20_{\pm2.17}$ | 77.54 |

Table 3: Rank (number of singular values with magnitudes $\geq 0.005$) and effective rank (erank) of weight updates in LLaMA2-7B with $r = 8$. Both Euclidean and intrinsic ranks are shown for HiRA.

| | Method | BoolQ | PIQA | SIQA | HS | WG | ARC-e | ARC-c | OBQA |
|---|---|---|---|---|---|---|---|---|---|
| **Rank** | LoRA | $8_{\pm0}$ | $8_{\pm0}$ | $8_{\pm0}$ | $8_{\pm0}$ | $8_{\pm0}$ | $8_{\pm0}$ | $8_{\pm0}$ | $8_{\pm0}$ |
| | ReLoRA | $16_{\pm0}$ | $16_{\pm0.1}$ | $24_{\pm0.08}$ | $32_{\pm0.33}$ | $32_{\pm1.07}$ | $16_{\pm0.6}$ | $15_{\pm0.3}$ | $36_{\pm2.32}$ |
| | HiRA (Eucl.) | $4004_{\pm217}$ | $3925_{\pm319}$ | $3971_{\pm291}$ | $3889_{\pm344}$ | $3670_{\pm497}$ | $3074_{\pm875}$ | $3315_{\pm721}$ | $3729_{\pm462}$ |
| | HiRA (intr.) | $8_{\pm0}$ | $8_{\pm0}$ | $8_{\pm0}$ | $8_{\pm0}$ | $8_{\pm0}$ | $8_{\pm0}$ | $8_{\pm0}$ | $8_{\pm0}$ |
| | ScaLoRA | $3326_{\pm671}$ | $3482_{\pm544}$ | $3661_{\pm392}$ | $3703_{\pm351}$ | $3695_{\pm363}$ | $2254_{\pm917}$ | $1347_{\pm706}$ | $3015_{\pm891}$ |
| | ScaLoRA-I | $1402_{\pm656}$ | $1990_{\pm843}$ | $2757_{\pm910}$ | $2937_{\pm880}$ | $2891_{\pm912}$ | $20_{\pm11}$ | $20_{\pm3}$ | $453_{\pm265}$ |
| **Erank** | LoRA | $2.7_{\pm0.6}$ | $1.9_{\pm0.4}$ | $1.8_{\pm0.4}$ | $2.3_{\pm0.6}$ | $1.2_{\pm0.2}$ | $1.6_{\pm0.4}$ | $1.7_{\pm0.4}$ | $1.3_{\pm0.3}$ |
| | ReLoRA | $2.6_{\pm0.6}$ | $1.9_{\pm0.5}$ | $1.9_{\pm0.4}$ | $1.6_{\pm0.4}$ | $2.0_{\pm0.6}$ | $1.7_{\pm0.4}$ | $1.7_{\pm0.5}$ | $2.0_{\pm0.6}$ |
| | HiRA (Eucl.) | $358.2_{\pm259.9}$ | $313.8_{\pm228.8}$ | $312.3_{\pm218.3}$ | $219.5_{\pm154.6}$ | $128.4_{\pm72.4}$ | $167.6_{\pm160.3}$ | $203.8_{\pm197.2}$ | $164.5_{\pm120.7}$ |
| | HiRA (intr.) | $2.9_{\pm1.5}$ | $2.4_{\pm1.4}$ | $2.5_{\pm1.3}$ | $1.9_{\pm0.9}$ | $1.5_{\pm0.6}$ | $2.5_{\pm1.4}$ | $2.0_{\pm1.5}$ | $1.7_{\pm0.7}$ |
| | ScaLoRA | $4.8_{\pm1.7}$ | $3.1_{\pm0.8}$ | $3.4_{\pm0.6}$ | $4.2_{\pm1.0}$ | $2.6_{\pm0.7}$ | $2.7_{\pm0.7}$ | $1.9_{\pm0.5}$ | $2.0_{\pm0.5}$ |
| | ScaLoRA-I | $4.6_{\pm1.5}$ | $3.0_{\pm0.8}$ | $2.6_{\pm0.7}$ | $4.2_{\pm1.0}$ | $2.3_{\pm0.6}$ | $2.6_{\pm0.6}$ | $1.9_{\pm0.5}$ | $1.9_{\pm0.5}$ |

all eight tasks. Table 2 compares ScaLoRA with LoRA (Hu et al., 2022), ReLoRA (Lialin et al., 2024), LoRA-GA (Wang et al., 2024), MoRA (Jiang et al., 2024), and HiRA (Huang et al., 2025). It is observed that ScaLoRA and ScaLoRA-I demonstrate similar performance, both outperforming all other competitors by a significant margin. This verifies our claim that ScaLoRA-I does not distinctly affect the effectiveness when $I$ is small. Further, the performance of ScaLoRA(-I) even surpasses LoRA with a higher rank of 32, yet incurring less computational overhead.

Moreover, we further investigate the rank of weight update $\mathbf{W}_T - \mathbf{W}_0$ in LLaMA2-7B under different high-rank adaptation approaches. Following (Lialin et al., 2024; Huang et al., 2025), only the singular values whose magnitudes exceed 0.005 are counted. MoRA has been excluded because of its nonlinearity. For HiRA, as its rank update pertains to the low-dimensional manifold $\{\mathbf{W}^{\text{ft}} \mid \mathbf{W}^{\text{ft}} = (\mathbf{A}\mathbf{B}^{\top}) \odot \mathbf{W}^{\text{pt}}\}$, we report both its Euclidean rank and its intrinsic (latent) rank, where the latter better reflects the geometry induced by its parameterization. The average rank and efficient rank $\text{erank}(\cdot) := \|\cdot\|_{\text{F}}^2 / \|\cdot\|_2^2$ along with their standard deviations across LoRA layers are reported in Table 3. ScaLoRA(-I) yields (e)rank proportional to the size and difficulty of the task. For small datasets such as ARC-e and ARC-c, the limited fine-tuning iterations renders a moderate-rank update, which is nevertheless sufficient to fit the task. In contrast, ReLoRA exhibits markedly lower (e)rank due to its infrequent merging operations. While HiRA consistently produces high Euclidean rank regardless of the dataset size and task difficulty, its intrinsic (e)rank remains low owing to its underlying low-dimensional manifold. Moreover, the erank of ScaLoRA(-I) is significantly higher than other baselines, suggesting that the weight update captures a richer and more diverse subspace of singular directions for task-specific adaptation.

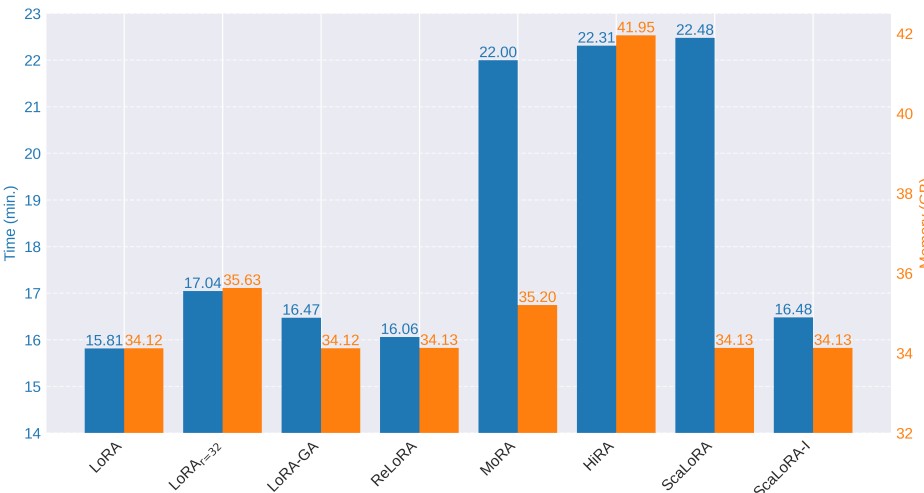

Figure 3: Overhead comparison using LLaMA3-8B on the BoolQ dataset.

Next, Figure 3 depicts the fine-tuning time (minutes) and memory cost (GB) of ScaLoRA(-I) with other alternatives, where the vertical axes start from nonzero values for better visual comparison. It is clear that MoRA, HiRA and ScaLoRA necessitate $50\%+$ time compared to LoRA, on par with our analysis in Section 3.4. Moreover, MoRA and HiRA require 1.08 and 7.83 GB extra memory in comparison to LoRA, while ScaLoRA(-I) merely leads to a negligible growth of 0.01 GB. Additionally, ScaLoRA-I showcases superior scalability in both time and space comparable to LoRA-GA and ReLoRA, which add marginally to LoRA with $r = 4$, and outperforms LoRA with $r = 32$. In practice, an appropriate choice of $I$ can provide a favorable balance between efficiency and convergence. An ablation test on the effect of varying $I$ is presented in Appendix D.2.

### 4.4 MATHEMATICAL PROBLEM SOLVING

The next numerical test assesses ScaLoRA on mathematical problem solving tasks, and scales to the larger Gemma-3-12B (Team et al., 2025) model. The model is fine-tuned on MetaMath (Yu et al., 2024), a mathematical question answering dataset for LLMs, and evaluated on GSM8K (Cobbe et al., 2021) and MATH (Hendrycks et al., 2021) datasets. MoRA and HiRA are omitted due to their limited scalability shown in Figure 3. Additionally, an ablation study is

Table 4: Mathematical problem solving using Gemma-3-12B.

| Method | GSM8K | MATH |
|---|---|---|
| LoRA | $81.20_{\pm 1.08}$ | $37.20_{\pm 0.63}$ |
| ScaLoRA-I | $\mathbf{82.11}_{\pm 1.06}$ | $\mathbf{37.96}_{\pm 0.64}$ |
| Scalar-only | $\underline{81.27}_{\pm 1.07}$ | $\underline{37.90}_{\pm 0.64}$ |

also included to show the enhanced fitting capacity of column scaling as opposed to scalar scaling. A variant of ScaLoRA-I with scalar scaling only is considered. The results are displayed in Table 4, where ScaLoRA-I again outperforms LoRA on both datasets. Moreover, it is also seen that ScaLoRA-I with scalar scaling improves upon LoRA yet underperforms ScaLoRA-I, illustrating the effectiveness of column-wise scaling. Extended ablation study on the scalar-only variant using commonsense reasoning datasets is provided in Appendix D.3.

## 5 CONCLUDING REMARKS

This paper investigated high-rank updates by gradually accumulating the optimal low-rank increments that minimize the per-step loss. It was argued that this idea faces two challenges, namely prohibitive computation and inefficient optimization. To address them, a novel approach termed ScaLoRA was introduced. By restricting the optimal adapters to the family of matrices whose columns are scaled from the original ones, ScaLoRA allowed for efficient optimization without resetting the gradient moment estimators. Performance guarantees were established respectively for scalar and column-wise scaling to pick out the optimal adapters in analytical form. Numerical tests covering natural language understanding, commonsense reasoning, and mathematical problem solving validated the consistent performance gain and scalability of ScaLoRA(-I).

ETHICS STATEMENTS

This work does not involve human subjects, personal data, or sensitive information. All experiments are conducted on publicly available LLMs and benchmark datasets, with details, links, and licenses provided in the Appendix. The proposed method aims to improve computational efficiency and convergence in fine-tuning, which abides by ICLR's code of ethic. Nevertheless, caution is advised when applying the method to generative tasks. The outputs of LLMs should be carefully reviewed, and safeguards such as gating mechanisms should be considered to ensure safety, reliability, and trustworthiness.

REPRODUCIBILITY STATEMENT

We have taken multiple steps to ensure reproducibility. The paper provides full algorithmic details, including theoretical proofs, pesudocodes, implementation details, and hyperparameter settings. We have also uploaded the complete source code and scripts used to reproduce our main results as the supplementary material. All LLMs and datasets used are publicly available, with links provided in the Appendix. These resources collectively enable other researchers to replicate our findings.

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

## A   MISSING PROOFS

This section provides the proofs omitted in the main paper.

### A.1   PROOF OF THEOREM 1

*Proof.* For notational simplicity, we will omit the subscript $t$ in the proof, and write $\tilde{\mathbf{A}}_t^*, \tilde{\mathbf{B}}_t^*$ as $\mathbf{A}, \mathbf{B}$.
We first verify the *sufficiency*. For $\mathbf{A}, \mathbf{B}$ satisfying (8), it follows that

$$
\begin{aligned}
-\eta \nabla\ell(\mathbf{W})\mathbf{B}\mathbf{B}^\top - \eta \mathbf{A}\mathbf{A}^\top \nabla\ell(\mathbf{W}) &= -\frac{1}{L}\nabla\ell(\mathbf{W})\mathbf{V}_{\mathcal{B}}\mathbf{Q}\mathbf{Q}^\top\mathbf{V}_{\mathcal{B}}^\top - \frac{1}{L}\mathbf{U}_{\mathcal{A}}\mathbf{P}\mathbf{P}^\top\mathbf{U}_{\mathcal{A}}^\top\nabla\ell(\mathbf{W}) \\
&= -\frac{1}{L}\nabla\ell(\mathbf{W})\mathbf{V}_{\mathcal{B}}\mathbf{V}_{\mathcal{B}}^\top - \frac{1}{L}\mathbf{U}_{\mathcal{A}}\mathbf{U}_{\mathcal{A}}^\top\nabla\ell(\mathbf{W}) \\
&\stackrel{(a)}{=} -\frac{1}{L}\mathbf{U}\boldsymbol{\Sigma}_{\mathcal{B}}\mathbf{V}_{\mathcal{B}}^\top - \frac{1}{L}\mathbf{U}_{\mathcal{A}}\boldsymbol{\Sigma}_{\mathcal{A},:}\mathbf{V}^\top \\
&= -\frac{1}{L}\sum_{i\in\mathcal{B}}\sigma_i\mathbf{u}_i\mathbf{v}_i^\top - \frac{1}{L}\sum_{i\in\mathcal{A}}\sigma_i\mathbf{u}_i\mathbf{v}_i^\top \\
&= -\frac{1}{L}\sum_{i=1}^{2r}\sigma_i\mathbf{u}_i\mathbf{v}_i^\top
\end{aligned}
\tag{12}
$$

where $(a)$ relies on the SVD $\nabla\ell(\mathbf{W}) = \mathbf{U}\boldsymbol{\Sigma}\mathbf{V}^\top$, and $\mathbf{u}_i, \mathbf{v}_i$ are the $i$-th columns of $\mathbf{U}, \mathbf{V}$.

Using the fact that $\mathrm{rank}(\nabla\ell(\mathbf{W})\mathbf{B}\mathbf{B}^\top) \le r$ and $\mathrm{rank}(\mathbf{A}\mathbf{A}^\top\nabla\ell(\mathbf{W})) \le r$, it holds

$$
\mathrm{rank}(\eta\nabla\ell(\mathbf{W})\mathbf{B}\mathbf{B}^\top + \eta\mathbf{A}\mathbf{A}^\top\nabla\ell(\mathbf{W})) \le r + r = 2r.
\tag{13}
$$

By Eckart–Young–Mirsky theorem (Eckart & Young, 1936), it turns out that (12) is the optimal rank-$2r$ approximation to $\frac{1}{L}\nabla\ell(\mathbf{W})$ that minimizes (7).

Next we show the *necessity*. For notational compactness, define $\mathcal{I} := \{1, \ldots, 2r\}$. Again by Eckart–Young–Mirsky theorem (Eckart & Young, 1936), the optimal rank-$2r$ approximation to $\frac{1}{L}\nabla\ell(\mathbf{W})$ should satisfy

$$
\nabla\ell(\mathbf{W})\mathbf{B}\mathbf{B}^\top + \mathbf{A}\mathbf{A}^\top\nabla\ell(\mathbf{W}) = \frac{1}{L\eta}\mathbf{U}_{\mathcal{I}}\boldsymbol{\Sigma}_{\mathcal{I},\mathcal{I}}\mathbf{V}_{\mathcal{I}}^\top.
\tag{14}
$$

To achieve this rank-$2r$ approximation, (13) suggests that we must have

$$
\mathrm{rank}(\nabla\ell(\mathbf{W})\mathbf{B}\mathbf{B}^\top) = \mathrm{rank}(\mathbf{A}\mathbf{A}^\top\nabla\ell(\mathbf{W})) = r.
$$

Additionally, since

$$
\begin{aligned}
\mathrm{rank}(\nabla\ell(\mathbf{W})\mathbf{B}\mathbf{B}^\top + \mathbf{A}\mathbf{A}^\top\nabla\ell(\mathbf{W})) &= \mathrm{rank}(\frac{1}{L\eta}\mathbf{U}_{\mathcal{I}}[\boldsymbol{\Sigma}]_{\mathcal{I},\mathcal{I}}\mathbf{V}_{\mathcal{I}}^\top) \\
&= \mathrm{rank}(\nabla\ell(\mathbf{W})\mathbf{B}\mathbf{B}^\top) + \mathrm{rank}(\mathbf{A}\mathbf{A}^\top\nabla\ell(\mathbf{W})),
\end{aligned}
$$

it must hold

$$
\mathrm{Col}(\nabla\ell(\mathbf{W})\mathbf{B}\mathbf{B}^\top) \cap \mathrm{Col}(\mathbf{A}\mathbf{A}^\top\nabla\ell(\mathbf{W})) = \{0\}
\tag{15a}
$$

$$
\mathrm{Col}(\nabla\ell(\mathbf{W})\mathbf{B}\mathbf{B}^\top) \oplus \mathrm{Col}(\mathbf{A}\mathbf{A}^\top\nabla\ell(\mathbf{W})) = \mathrm{Col}(\mathbf{U}_{\mathcal{I}}[\boldsymbol{\Sigma}]_{\mathcal{I},\mathcal{I}}\mathbf{V}_{\mathcal{I}}^\top) = \mathrm{Col}(\mathbf{U}_{\mathcal{I}})
\tag{15b}
$$

$$
\mathrm{Row}(\nabla\ell(\mathbf{W})\mathbf{B}\mathbf{B}^\top) \cap \mathrm{Row}(\mathbf{A}\mathbf{A}^\top\nabla\ell(\mathbf{W})) = \{0\}
\tag{15c}
$$

$$
\mathrm{Row}(\nabla\ell(\mathbf{W})\mathbf{B}\mathbf{B}^\top) \oplus \mathrm{Row}(\mathbf{A}\mathbf{A}^\top\nabla\ell(\mathbf{W})) = \mathrm{Row}(\mathbf{U}_{\mathcal{I}}[\boldsymbol{\Sigma}]_{\mathcal{I},\mathcal{I}}\mathbf{V}_{\mathcal{I}}^\top) = \mathrm{Row}(\mathbf{V}_{\mathcal{I}}^\top).
\tag{15d}
$$

In other words, the two terms $\nabla\ell(\mathbf{W})\mathbf{B}\mathbf{B}^\top$ and $\mathbf{A}\mathbf{A}^\top\nabla\ell(\mathbf{W})$ splits the $2r$-dimensional column and row spaces of $\mathbf{U}_{\mathcal{I}}[\boldsymbol{\Sigma}]_{\mathcal{I},\mathcal{I}}\mathbf{V}_{\mathcal{I}}^\top$ into two $r$-dimensional subspaces.

Moreover, because $r = \mathrm{rank}(\mathbf{A}\mathbf{A}^\top\nabla\ell(\mathbf{W})) \le \mathrm{rank}(\mathbf{A}) \le r$, it follows that $\mathrm{rank}(\mathbf{A}) = r$. Thus we obtain $\mathrm{Col}(\mathbf{A}\mathbf{A}^\top\nabla\ell(\mathbf{W})) = \mathrm{Col}(\mathbf{A})$. Then, (14), (15a) and (15b) imply that, the two terms

$\nabla \ell(\mathbf{W})\mathbf{B}\mathbf{B}^\top$ and $\mathbf{A}\mathbf{A}^\top \nabla \ell(\mathbf{W})$ are respectively the orthogonal projections of $\frac{1}{L\eta}\mathbf{U}_\mathcal{I}[\mathbf{\Sigma}]_{\mathcal{I},\mathcal{I}}\mathbf{V}_\mathcal{I}^\top$ onto the disjoint subspaces $\mathrm{Col}(\nabla \ell(\mathbf{W})\mathbf{B}\mathbf{B}^\top)$ and $\mathrm{Col}(\mathbf{A}\mathbf{A}^\top\nabla \ell(\mathbf{W})) = \mathrm{Col}(\mathbf{A})$. To be specific, defining projection matrix $\mathbf{P_A} := \mathbf{A}(\mathbf{A}^\top\mathbf{A})^{-1}\mathbf{A}^\top$, we have

$$\mathbf{A}\mathbf{A}^\top\nabla \ell(\mathbf{W}) = \mathbf{P_A}\frac{1}{L\eta}\mathbf{U}_\mathcal{I}\mathbf{\Sigma}_{\mathcal{I},\mathcal{I}}\mathbf{V}_\mathcal{I}^\top \overset{(a)}{=} \frac{1}{L\eta}\mathbf{P_A}\mathbf{P}_{\mathbf{U}_\mathcal{I}}\nabla \ell(\mathbf{W}) \overset{(b)}{=} \frac{1}{L\eta}\mathbf{P_A}\nabla \ell(\mathbf{W})$$

where $(a)$ utilizes $\mathbf{U}_\mathcal{I}\mathbf{\Sigma}_{\mathcal{I},\mathcal{I}}\mathbf{V}_\mathcal{I}^\top = \mathbf{U}_\mathcal{I}\mathbf{U}_\mathcal{I}^\top\nabla \ell(\mathbf{W}) = \mathbf{P}_{\mathbf{U}_\mathcal{I}}\nabla \ell(\mathbf{W})$, and $(b)$ leverages $\mathrm{Col}(\mathbf{A}) \subset \mathrm{Col}(\mathbf{U}_\mathcal{I})$ so that $\mathbf{P_A}\mathbf{P}_{\mathbf{U}_\mathcal{I}} = \mathbf{P_A}$.

Left-multiplying both sides by $\mathbf{A}^\top$ leads to

$$0 = \mathbf{A}^\top\mathbf{A}\mathbf{A}^\top\nabla \ell(\mathbf{W}) - \frac{1}{L\eta}\mathbf{A}^\top\nabla \ell(\mathbf{W}) = (\mathbf{A}^\top\mathbf{A} - \frac{1}{L\eta}\mathbf{I}_r)\mathbf{A}^\top\nabla \ell(\mathbf{W}).$$

Given that $\mathbf{A}^\top\nabla \ell(\mathbf{W})$ has full row rank $r$, we must have $\mathbf{A}^\top\mathbf{A} - \frac{1}{L\eta}\mathbf{I}_r = 0$. That says, $\sqrt{L\eta}\mathbf{A}$ has orthonormal columns, and hence $\mathbf{P_A} = L\eta\mathbf{A}\mathbf{A}^\top$. Similarly, using (15c) and (15d), we acquire that $\mathbf{B}$ also has orthonormal columns, and $\mathbf{P_B} = L\eta\mathbf{B}\mathbf{B}^\top$.

Now left-multiplying $\mathbf{U}_\mathcal{I}^\top$ and right-multiplying $\mathbf{V}_\mathcal{I}$ on both sides of (14) result in

$$\mathbf{\Sigma}_\mathcal{I}\mathbf{V}_\mathcal{I}^\top\mathbf{B}\mathbf{B}^\top\mathbf{V}_\mathcal{I} + \mathbf{U}_\mathcal{I}^\top\mathbf{A}\mathbf{A}^\top\mathbf{U}_\mathcal{I}\mathbf{\Sigma}_\mathcal{I} = \frac{1}{L\eta}\mathbf{\Sigma}_\mathcal{I}. \tag{16}$$

We next prove that $\mathbf{V}_\mathcal{I}^\top\mathbf{B}\mathbf{B}^\top\mathbf{V}_\mathcal{I}$ and $\mathbf{U}_\mathcal{I}^\top\mathbf{A}\mathbf{A}^\top\mathbf{U}_\mathcal{I}$ are both diagonal. Without loss of generality, assume the $\sigma_i \neq \sigma_j, i \neq j, \ \forall i, j \in \mathcal{I}$. Otherwise, the rank-$2r$ SVD is not unique, and one can always rotate the axes of $\mathbf{U}_\mathcal{I}$ and $\mathbf{V}_\mathcal{I}$ to align with $\mathbf{A}$ and $\mathbf{B}$. By the relationship, the non-diagonal elements satisfy for $\forall i, j \in \mathcal{I}$ and $i \neq j$

$$\sigma_i[\mathbf{V}_\mathcal{I}^\top\mathbf{B}\mathbf{B}^\top\mathbf{V}_\mathcal{I}]_{ij} + [\mathbf{U}_\mathcal{I}^\top\mathbf{A}\mathbf{A}^\top\mathbf{U}_\mathcal{I}]_{ij}\sigma_j = \frac{1}{L\eta}[\mathbf{\Sigma}_\mathcal{I}]_{ij} = 0$$

$$\sigma_j[\mathbf{V}_\mathcal{I}^\top\mathbf{B}\mathbf{B}^\top\mathbf{V}_\mathcal{I}]_{ij} + [\mathbf{U}_\mathcal{I}^\top\mathbf{A}\mathbf{A}^\top\mathbf{U}_\mathcal{I}]_{ij}\sigma_i = \frac{1}{L\eta}[\mathbf{\Sigma}_\mathcal{I}]_{ji} = 0$$

Solving for $[\mathbf{V}_\mathcal{I}^\top\mathbf{B}\mathbf{B}^\top\mathbf{V}_\mathcal{I}]_{ij}$ and $[\mathbf{U}_\mathcal{I}^\top\mathbf{A}\mathbf{A}^\top\mathbf{U}_\mathcal{I}]_{ij}$, we obtain

$$(\sigma_i^2 - \sigma_j^2)[\mathbf{V}_\mathcal{I}^\top\mathbf{B}\mathbf{B}^\top\mathbf{V}_\mathcal{I}]_{ij} = 0, \quad (\sigma_j^2 - \sigma_i^2)[\mathbf{U}_\mathcal{I}^\top\mathbf{A}\mathbf{A}^\top\mathbf{U}_\mathcal{I}]_{ij} = 0.$$

This demonstrates $[\mathbf{V}_\mathcal{I}^\top\mathbf{B}\mathbf{B}^\top\mathbf{V}_\mathcal{I}]_{ij} = [\mathbf{U}_\mathcal{I}^\top\mathbf{A}\mathbf{A}^\top\mathbf{U}_\mathcal{I}]_{ij} = 0$, so $\mathbf{V}_\mathcal{I}^\top\mathbf{B}\mathbf{B}^\top\mathbf{V}_\mathcal{I}$ and $\mathbf{U}_\mathcal{I}^\top\mathbf{A}\mathbf{A}^\top\mathbf{U}_\mathcal{I}$ are diagonal.

Then, recall that $\sqrt{L\eta}\mathbf{A}$ has orthonormal columns, so

$$(L\eta\mathbf{U}_\mathcal{I}^\top\mathbf{A}\mathbf{A}^\top\mathbf{U}_\mathcal{I})^2 = L\eta\mathbf{U}_\mathcal{I}^\top\mathbf{A}\mathbf{A}^\top\mathbf{U}_\mathcal{I}.$$

As the diagonal matrix $\mathbf{U}_\mathcal{I}^\top\mathbf{A}\mathbf{A}^\top\mathbf{U}_\mathcal{I}$ is symmetric positive semi-definite, its diagnoal elements satisfy

$$[L\eta\mathbf{U}_\mathcal{I}^\top\mathbf{A}\mathbf{A}^\top\mathbf{U}_\mathcal{I}]_{ii}^2 = [L\eta\mathbf{U}_\mathcal{I}^\top\mathbf{A}\mathbf{A}^\top\mathbf{U}_\mathcal{I}]_{ii} \geq 0 \ \Rightarrow \ [\mathbf{U}_\mathcal{I}^\top\mathbf{A}\mathbf{A}^\top\mathbf{U}_\mathcal{I}]_{ii} = 0 \text{ or } \frac{1}{L\eta}.$$

Likewise we also have $[\mathbf{V}_\mathcal{I}^\top\mathbf{B}\mathbf{B}^\top\mathbf{V}_\mathcal{I}]_{ii} = 0$ or $\frac{1}{L\eta}$.

Defining $\mathcal{A} := \{i \mid [\mathbf{U}_\mathcal{I}^\top\mathbf{A}\mathbf{A}^\top\mathbf{U}_\mathcal{I}]_{ii} = 1/(L\eta)\}$ and $\mathcal{B} := \{i \mid [\mathbf{V}_\mathcal{I}^\top\mathbf{B}\mathbf{B}^\top\mathbf{V}_\mathcal{I}]_{ii} = 1/(L\eta)\}$, it follows from (16) that

$$|\mathcal{A}| = |\mathcal{B}| = r, \ \mathcal{A} \cup \mathcal{B} = \mathcal{I}.$$

As a result, it holds

$$\mathbf{U}_\mathcal{I}^\top\mathbf{A}\mathbf{A}^\top\mathbf{U}_\mathcal{I} = \frac{1}{L\eta}\sum_{i\in\mathcal{A}}\mathbf{e}_i\mathbf{e}_i^\top \ \Rightarrow \ (\mathbf{U}_\mathcal{I}\mathbf{U}_\mathcal{I}^\top)\mathbf{A}\mathbf{A}^\top(\mathbf{U}_\mathcal{I}\mathbf{U}_\mathcal{I}^\top) = \frac{1}{L\eta}\sum_{i\in\mathcal{A}}\mathbf{u}_i\mathbf{u}_i^\top = \frac{1}{L\eta}\mathbf{U}_\mathcal{A}\mathbf{U}_\mathcal{A}^\top$$

where $\mathbf{e}_i$ is the $i$-th column of the identity matrix $\mathbf{I}_{2r}$.

Notice that $\mathbf{U}_{\mathcal{I}}\mathbf{U}_{\mathcal{I}}^\top = \mathbf{P}_{\mathbf{U}_{\mathcal{I}}}$, and $\mathrm{Col}(\mathbf{A}) \subset \mathrm{Col}(\mathbf{U}_{\mathcal{I}})$. It follows

$$(\mathbf{U}_{\mathcal{I}}\mathbf{U}_{\mathcal{I}}^\top)\mathbf{A}\mathbf{A}^\top(\mathbf{U}_{\mathcal{I}}\mathbf{U}_{\mathcal{I}}^\top) = \mathbf{A}\mathbf{A}^\top = \frac{1}{L\eta}\mathbf{U}_{\mathcal{A}}\mathbf{U}_{\mathcal{A}}^\top.$$

Using the fact that $L\eta\mathbf{A}^\top\mathbf{A} = \mathbf{I}_r$ and $\mathrm{Col}(\mathbf{A}) = \mathrm{Col}(\mathbf{U}_{\mathcal{A}})$, we acquire

$$\mathbf{A} = \frac{1}{\sqrt{L\eta}}\mathbf{U}_{\mathcal{A}}\mathbf{P}, \ \mathbf{P} \in \mathrm{O}(r)$$

and similarly

$$\mathbf{B} = \frac{1}{\sqrt{L\eta}}\mathbf{V}_{\mathcal{B}}\mathbf{Q}, \ \mathbf{Q} \in \mathrm{O}(r)$$

which concludes the proof. $\qquad\square$

### A.2  PROOF OF THEOREM 3

*Proof.* As before, the subscript $t$ will be omitted in the proof for simplicity. First notice that when $\mathbf{A}\mathbf{A}^\top\nabla\ell(\mathbf{W}) \neq 0$ and $\alpha \to \infty$, or $\nabla\ell(\mathbf{W})\mathbf{B}\mathbf{B}^\top \neq 0$ and $\beta \to \infty$, the objective value (9) goes unbounded to $+\infty$. Additionally, if $\mathbf{A}\mathbf{A}^\top\nabla\ell(\mathbf{W}) = 0$ (or $\nabla\ell(\mathbf{W})\mathbf{B}\mathbf{B}^\top = 0$), changing $\alpha$ (or $\beta$) has no impact on the objective value. By Assumption 2 and Lemma 6, at least one of $\mathbf{A}\mathbf{A}^\top\nabla\ell(\mathbf{W})$ and $\nabla\ell(\mathbf{W})\mathbf{B}\mathbf{B}^\top$ is nonzero. As a the objective (9) is a continuous function of $\alpha$ and $\beta$ in $\mathbb{R}^2$, there must be some global minimum achieved in the interior of $\mathbb{R}^2$. Therefore, we can examine the stationary points of the objective.

The first-order stationary point condition yields

$$\alpha^*\Big(\alpha^{*2}\|\mathbf{A}\mathbf{A}^\top\nabla\ell(\mathbf{W})\|_{\mathrm{F}}^2 - \langle\mathbf{A}\mathbf{A}^\top\nabla\ell(\mathbf{W}), \frac{1}{L\eta}\nabla\ell(\mathbf{W}) - \beta^{*2}\nabla\ell(\mathbf{W})\mathbf{B}\mathbf{B}^\top\rangle_{\mathrm{F}}\Big) = 0, \quad (17a)$$

$$\beta^*\Big(\beta^{*2}\|\nabla\ell(\mathbf{W})\mathbf{B}\mathbf{B}^\top\|_{\mathrm{F}}^2 - \langle\nabla\ell(\mathbf{W})\mathbf{B}\mathbf{B}^\top, \frac{1}{L\eta}\nabla\ell(\mathbf{W}) - \alpha^{*2}\mathbf{A}\mathbf{A}^\top\nabla\ell(\mathbf{W})\rangle_{\mathrm{F}}\Big) = 0. \quad (17b)$$

These two equations offers nine stationary points, which are investigated in the following.

We next show that the trivial stationary point $(\alpha, \beta) = (0, 0)$ must not be a local minimum. Plugging $\alpha = 0$ and $\beta = 0$ into (9) leads to objective value of $\|\nabla\ell(\mathbf{W})\|_{\mathrm{F}}^2/2L$. By assumption 2, at lease one of $\|\mathbf{A}^\top\nabla\ell(\mathbf{W})\|_{\mathrm{F}}$ and $\|\nabla\ell(\mathbf{W})\mathbf{B}\|_{\mathrm{F}}$ should be nonzero. Without loss of generality, assume $\|\mathbf{A}^\top\nabla\ell(\mathbf{W})\|_{\mathrm{F}} > 0$. Taking $\beta = 0$ and $0 < \alpha < 2/(\sqrt{L\eta}\|\mathbf{A}\|_2)$, the objective (9) is upper bounded by

$$\frac{L}{2}\Big\|\frac{1}{L}\nabla\ell(\mathbf{W}) - \eta\beta^2\nabla\ell(\mathbf{W})\mathbf{B}\mathbf{B}^\top - \eta\alpha^2\mathbf{A}\mathbf{A}^\top\nabla\ell(\mathbf{W})\Big\|_{\mathrm{F}}^2 \leq \frac{L}{2}\|\nabla\ell(\mathbf{W})\|_{\mathrm{F}}^2\Big\|\frac{1}{L}\mathbf{I}_m - \eta\alpha^2\mathbf{A}\mathbf{A}^\top\Big\|_2^2$$

$$< \frac{L}{2}\|\nabla\ell(\mathbf{W})\|_{\mathrm{F}}^2.$$

This demonstrates $(\alpha, \beta) = (0, 0)$ must not be a local minimum. Therefore, at lease one of $|\alpha^*|$ and $|\beta^*|$ should be strictly positive.

To determine whether $|\alpha^*|$ and $|\beta^*|$ are strictly positive or zeros, we consider the following four cases.

**Case 1:** $C^A > 0$ and $C^B \leq 0$.

We first rewrite the objective (9) as a quadratic function of $a^2 \geq 0$ via

$$\Big\|\frac{1}{L}\nabla\ell(\mathbf{W}) - \eta\beta^2\nabla\ell(\mathbf{W})\mathbf{B}\mathbf{B}^\top - \eta\alpha^2\mathbf{A}\mathbf{A}^\top\nabla\ell(\mathbf{W})\Big\|_{\mathrm{F}}^2$$

$$= \eta^2\|\mathbf{A}\mathbf{A}^\top\nabla\ell(\mathbf{W})\|_{\mathrm{F}}^2\alpha^4 - 2\eta\langle\mathbf{A}\mathbf{A}^\top\nabla\ell(\mathbf{W}), \frac{1}{L}\nabla\ell(\mathbf{W}) - \eta\beta^2\nabla\ell(\mathbf{W})\mathbf{B}\mathbf{B}^\top\rangle_{\mathrm{F}}\alpha^2 + \mathrm{Const.}$$

$$= \eta^2\|\mathbf{A}\mathbf{A}^\top\nabla\ell(\mathbf{W})\|_{\mathrm{F}}^2\alpha^4 - 2\eta\Big(\frac{1}{L}\|\mathbf{A}^\top\nabla\ell(\mathbf{W})\|_{\mathrm{F}}^2 - \eta\beta^2\|\mathbf{A}^\top\nabla\ell(\mathbf{W})\mathbf{B}\|_{\mathrm{F}}^2\Big)\alpha^2 + \mathrm{Const.}$$

which attains its minimal value at

$$\alpha^{*2} = \max\left\{0, \frac{\frac{1}{L\eta}\|\mathbf{A}^\top\nabla\ell(\mathbf{W})\|_\mathrm{F}^2 - \beta^{*2}\|\mathbf{A}^\top\nabla\ell(\mathbf{W})\mathbf{B}\|_\mathrm{F}^2}{\|\mathbf{A}\mathbf{A}^\top\nabla\ell(\mathbf{W})\|_\mathrm{F}^2}\right\} \quad (18)$$

Using $C^A > 0$, we next show that $\alpha^* = 0$ leads to a contradiction, and thus $|\alpha^*|$ must be strictly positive.

Note that $C^A > 0$ indicates $\mathbf{A}^\top\nabla\ell(\mathbf{W}) \neq \mathbf{0}$ and $\nabla\ell(\mathbf{W})\mathbf{B} \neq \mathbf{0}$; otherwise $C^A = 0$ by its definition. By Lemma 6, it follows that $\|\mathbf{A}\mathbf{A}^\top\nabla\ell(\mathbf{W})\|_\mathrm{F} > 0$ and $\|\nabla\ell(\mathbf{W})\mathbf{B}\mathbf{B}^\top\|_\mathrm{F} > 0$. If $\alpha^* = 0$, from the previous discussions we must have $|\beta^*| > 0$. However, applying $\alpha^* = 0$ and $|\beta^*| > 0$ to (17) renders

$$\beta^{*2} = \frac{\langle\nabla\ell(\mathbf{W})\mathbf{B}\mathbf{B}^\top, \frac{1}{L\eta}\nabla\ell(\mathbf{W})\rangle_\mathrm{F}}{\|\nabla\ell(\mathbf{W})\mathbf{B}\mathbf{B}^\top\|_\mathrm{F}^2} = \frac{\|\nabla\ell(\mathbf{W})\mathbf{B}\|_\mathrm{F}^2}{L\eta\|\nabla\ell(\mathbf{W})\mathbf{B}\mathbf{B}^\top\|_\mathrm{F}^2}.$$

As a result, (18) reduces to

$$\alpha^{*2} = \max\left\{0, \frac{\|\mathbf{A}^\top\nabla\ell(\mathbf{W})\|_\mathrm{F}^2 - \frac{\|\nabla\ell(\mathbf{W})\mathbf{B}\|_\mathrm{F}^2}{\|\nabla\ell(\mathbf{W})\mathbf{B}\mathbf{B}^\top\|_\mathrm{F}^2}\|\mathbf{A}^\top\nabla\ell(\mathbf{W})\mathbf{B}\|_\mathrm{F}^2}{L\eta\|\mathbf{A}\mathbf{A}^\top\nabla\ell(\mathbf{W})\|_\mathrm{F}^2}\right\}$$

$$= \max\left\{0, \frac{C^A}{L\eta\|\mathbf{A}\mathbf{A}^\top\nabla\ell(\mathbf{W})\|_\mathrm{F}^2\|\nabla\ell(\mathbf{W})\mathbf{B}\mathbf{B}^\top\|_\mathrm{F}^2}\right\}$$

$$= \frac{C^A}{L\eta\|\mathbf{A}\mathbf{A}^\top\nabla\ell(\mathbf{W})\|_\mathrm{F}^2\|\nabla\ell(\mathbf{W})\mathbf{B}\mathbf{B}^\top\|_\mathrm{F}^2} > 0$$

This contradicts the assumption $\alpha^* = 0$, and thus we must have $|\alpha^*| > 0$.

Next, we show that $C^B \leq 0$ leads to $\beta^* = 0$. Assuming $|\beta^*|$ is also strictly positive, solving (17) results in

$$L\eta C\alpha^{*2} = C^A > 0, \quad L\eta C\beta^{*2} = C^B \leq 0$$

which contradicts $|\alpha^*|, |\beta^*| > 0$.

To this end, it must hold $|\alpha^*| > 0$, $\beta^* = 0$. Combining this with (17) yields the solution

$$\alpha^{*2} = \frac{\|\mathbf{A}^\top\nabla\ell(\mathbf{W})\|_\mathrm{F}^2}{L\eta\|\mathbf{A}\mathbf{A}^\top\nabla\ell(\mathbf{W})\|_\mathrm{F}^2}, \quad \beta^* = 0. \quad (19)$$

**Case 2:** $C^A \leq 0$ and $C^B > 0$.

The analysis is akin to Case 1.

**Case 3:** $C = 0$.

By Assumption 2, at least one of $\mathbf{A}$ and $\mathbf{B}$ should be non-zero. Assume $\mathbf{A} \neq 0$ for simplicity, while similar derivation applies to $\mathbf{B} \neq 0$.

Using Cauchy-Schwarz inequality, it follows

$$C = \|\mathbf{A}\mathbf{A}^\top\nabla\ell(\mathbf{W})\|_\mathrm{F}^2\|\nabla\ell(\mathbf{W})\mathbf{B}\mathbf{B}^\top\|_\mathrm{F}^2 - \|\mathbf{A}^\top\nabla\ell(\mathbf{W})\mathbf{B}\|_\mathrm{F}^4$$

$$= \|\mathbf{A}\mathbf{A}^\top\nabla\ell(\mathbf{W})\|_\mathrm{F}^2\|\nabla\ell(\mathbf{W})\mathbf{B}\mathbf{B}^\top\|_\mathrm{F}^2 - \langle\mathbf{A}\mathbf{A}^\top\nabla\ell(\mathbf{W}), \nabla\ell(\mathbf{W})\mathbf{B}\mathbf{B}^\top\rangle_\mathrm{F}^2 \geq 0$$

where the equality holds if and only if $\nabla\ell(\mathbf{W})\mathbf{B}\mathbf{B}^\top = \xi\mathbf{A}\mathbf{A}^\top\nabla\ell(\mathbf{W})$ for some constant $\xi \in \mathbb{R}$.

If $\xi = 0$, solving (17) with $\nabla\ell(\mathbf{W})\mathbf{B}\mathbf{B}^\top = 0$ gives (19).

If $\xi \neq 0$, substituting $\nabla\ell(\mathbf{W})\mathbf{B}\mathbf{B}^\top = \xi\mathbf{A}\mathbf{A}^\top\nabla\ell(\mathbf{W})$ in (17) leads to

$$\alpha^*\left((\alpha^{*2} + \xi\beta^{*2})\|\mathbf{A}\mathbf{A}^\top\nabla\ell(\mathbf{W})\|_\mathrm{F}^2 - \frac{1}{L\eta}\|\mathbf{A}^\top\nabla\ell(\mathbf{W})\|_\mathrm{F}^2\right) = 0,$$

$$\beta^*\left((\alpha^{*2} + \xi\beta^{*2})\|\mathbf{A}\mathbf{A}^\top\nabla\ell(\mathbf{W})\|_\mathrm{F}^2 - \frac{1}{L\eta}\|\mathbf{A}^\top\nabla\ell(\mathbf{W})\|_\mathrm{F}^2\right) = 0.$$

As $(\alpha, \beta) = (0, 0)$ has been shown non-optimal, it must holds

$$\alpha^{*2} + \xi\beta^{*2} = \frac{\|\mathbf{A}^\top\nabla\ell(\mathbf{W})\|_F^2}{L\eta\|\mathbf{A}\mathbf{A}^\top\nabla\ell(\mathbf{W})\|_F^2}. \tag{20}$$

This relationship and $\nabla\ell(\mathbf{W})\mathbf{B}\mathbf{B}^\top = \xi\mathbf{A}\mathbf{A}^\top\nabla\ell(\mathbf{W})$ renders objective value

$$\frac{L}{2}\left\|\frac{1}{L}\nabla\ell(\mathbf{W}) - \eta\beta^{*2}\nabla\ell(\mathbf{W})\mathbf{B}\mathbf{B}^\top - \eta\alpha^{*2}\mathbf{A}\mathbf{A}^\top\nabla\ell(\mathbf{W})\right\|_F^2$$

$$= \frac{1}{2L}\left\|\nabla\ell(\mathbf{W}) - \frac{\|\mathbf{A}^\top\nabla\ell(\mathbf{W})\|_F^2}{\|\mathbf{A}\mathbf{A}^\top\nabla\ell(\mathbf{W})\|_F^2}\mathbf{A}\mathbf{A}^\top\nabla\ell(\mathbf{W})\right\|_F^2 = \frac{1}{2L}\left(\|\nabla\ell(\mathbf{W})\|_F^2 - \frac{\|\mathbf{A}^\top\nabla\ell(\mathbf{W})\|_F^4}{\|\mathbf{A}\mathbf{A}^\top\nabla\ell(\mathbf{W})\|_F^2}\right)$$

which is a constant independent of $\alpha^{*2}$ and $\beta^{*2}$. In other words, the optimal is achieved as if (20) is satisfied. One of such choices is simply (19).

Likewise, if $\mathbf{B} \neq 0$, a valid choice is

$$\alpha^* = 0, \quad \beta^{*2} = \frac{\|\nabla\ell(\mathbf{W})\mathbf{B}\|_F^2}{L\eta\|\nabla\ell(\mathbf{W})\mathbf{B}\mathbf{B}^\top\|_F^2}.$$

**Case 4:** $C^A \geq 0$, $C^B \geq 0$ and $C > 0$.

We first prove that $C^A = C^B = 0$ is impossible when $C > 0$. Assuming $C^A = C^B = 0$, it follows from their definitions that

$$\|\mathbf{A}^\top\nabla\ell(\mathbf{W})\|_F^2\|\nabla\ell(\mathbf{W})\mathbf{B}\mathbf{B}^\top\|_F^2 = \|\nabla\ell(\mathbf{W})\mathbf{B}\|_F^2\|\mathbf{A}^\top\nabla\ell(\mathbf{W})\mathbf{B}\|_F^2,$$

$$\|\nabla\ell(\mathbf{W})\mathbf{B}\|_F^2\|\mathbf{A}\mathbf{A}^\top\nabla\ell(\mathbf{W})\|_F^2 = \|\mathbf{A}^\top\nabla\ell(\mathbf{W})\|_F^2\|\mathbf{A}^\top\nabla\ell(\mathbf{W})\mathbf{B}\|_F^2$$

Multiplying the two equations on both sides and rearranging the terms yield

$$\|\mathbf{A}^\top\nabla\ell(\mathbf{W})\|_F^2\|\nabla\ell(\mathbf{W})\mathbf{B}\|_F^2\left(\|\mathbf{A}\mathbf{A}^\top\nabla\ell(\mathbf{W})\|_F^2\|\nabla\ell(\mathbf{W})\mathbf{B}\mathbf{B}^\top\|_F^2 - \|\mathbf{A}^\top\nabla\ell(\mathbf{W})\mathbf{B}\|_F^4\right) = 0.$$

As $C = \|\mathbf{A}\mathbf{A}^\top\nabla\ell(\mathbf{W})\|_F^2\|\nabla\ell(\mathbf{W})\mathbf{B}\mathbf{B}^\top\|_F^2 - \|\mathbf{A}^\top\nabla\ell(\mathbf{W})\mathbf{B}\|_F^4 > 0$, we must have either $\|\mathbf{A}^\top\nabla\ell(\mathbf{W})\|_F^2 = 0$ or $\|\nabla\ell(\mathbf{W})\mathbf{B}\|_F^2 = 0$. However, both cases lead to $C = 0$, thus deriving a contradiction.

Now assume $C^A > 0$ without loss of generality, which leads to $|\alpha^*| > 0$ as proved in Case 1. Next, applying (18) into the objective (9) and reformulating it as a quadratic function of $\beta^{*2}$ causes

$$\left\|\frac{1}{L}\nabla\ell(\mathbf{W}) - \eta\beta^2\nabla\ell(\mathbf{W})\mathbf{B}\mathbf{B}^\top - \eta\alpha^{*2}\mathbf{A}\mathbf{A}^\top\nabla\ell(\mathbf{W})\right\|_F^2$$

$$= \left\|\frac{1}{L}\nabla\ell(\mathbf{W}) - \eta\beta^2\nabla\ell(\mathbf{W})\mathbf{B}\mathbf{B}^\top - \eta\frac{\frac{1}{L\eta}\|\mathbf{A}^\top\nabla\ell(\mathbf{W})\|_F^2 - \beta^2\|\mathbf{A}^\top\nabla\ell(\mathbf{W})\mathbf{B}\|_F^2}{\|\mathbf{A}\mathbf{A}^\top\nabla\ell(\mathbf{W})\|_F^2}\mathbf{A}\mathbf{A}^\top\nabla\ell(\mathbf{W})\right\|_F^2$$

$$= \eta^2\left(\|\ell(\mathbf{W})\mathbf{B}\mathbf{B}^\top\|_F^2 - \frac{\|\mathbf{A}^\top\nabla\ell(\mathbf{W})\mathbf{B}\|_F^4}{\|\mathbf{A}\mathbf{A}^\top\nabla\ell(\mathbf{W})\|_F^2}\right)\beta^4 -$$

$$\frac{2\eta}{L}\left(\|\nabla\ell(\mathbf{W})\mathbf{B}\|_F^2 - \frac{\|\mathbf{A}^\top\nabla\ell(\mathbf{W})\|_F^2\|\mathbf{A}^\top\nabla\ell(\mathbf{W})\mathbf{B}\|_F^2}{\|\mathbf{A}\mathbf{A}^\top\nabla\ell(\mathbf{W})\|_F^2}\right)\beta^2 + \text{Const.}$$

$$= \frac{\eta^2 C}{\|\mathbf{A}\mathbf{A}^\top\nabla\ell(\mathbf{W})\|_F^2}\beta^4 - \frac{2\eta C^B}{L\|\mathbf{A}\mathbf{A}^\top\nabla\ell(\mathbf{W})\|_F^2}\beta^2 + \text{Const.}.$$

As $C > 0$, it follows that $\beta^{*2} = C^B/(L\eta C)$. Plugging this back to (18) gives $\alpha^{*2} = C^A/(L\eta C)$. $\square$

### A.3 PROOF OF THEOREM 5

*Proof.* The high-level idea of the proof is similar to the proof of Case 4 of Theorem 3. First, for the same rationale, there must be stationary point(s) in the interior of $\mathbb{R}^{2r}$ achieving the global minimum.

Denoting by $\phi := \alpha^{\circ 2}$ and $\psi := \beta^{\circ 2}$, the objective (10) can be equivalently written as a constrained optimization problem

$$\min_{\phi, \psi \in \mathbb{R}_+^r} \frac{L}{2}\left\|\frac{1}{L}\nabla\ell(\mathbf{W}) - \eta\nabla\ell(\mathbf{W})\mathbf{B}\,\mathrm{diag}^2(\psi)\mathbf{B}^\top - \eta\mathbf{A}\,\mathrm{diag}^2(\phi)\mathbf{A}^\top\nabla\ell(\mathbf{W})\right\|_F^2. \tag{21}$$

The optimal value of (21) is lower bounded by the optimal value of its unconstrained counterpart

$$\min_{\boldsymbol{\phi},\boldsymbol{\psi}} \frac{L}{2}\left\|\frac{1}{L}\nabla\ell(\mathbf{W}) - \eta\nabla\ell(\mathbf{W})\mathbf{B}\operatorname{diag}^2(\boldsymbol{\psi})\mathbf{B}^\top - \eta\mathbf{A}\operatorname{diag}^2(\boldsymbol{\phi})\mathbf{A}^\top\nabla\ell(\mathbf{W})\right\|_{\mathrm{F}}^2. \quad (22)$$

Next, we show that under the conditions of Theorem 5, the optimum points of (22) is inside the constraint $\mathbb{R}^r_+$, which is thus also the optimum of (21).

The optimality condition for (22) is

$$\mathbf{A}^\top\mathbf{A}\operatorname{diag}(\boldsymbol{\phi})\mathbf{A}^\top\nabla\ell(\mathbf{W})\ell(\mathbf{W})^\top\mathbf{A} - \frac{1}{L\eta}\mathbf{A}^\top\nabla\ell(\mathbf{W})\ell(\mathbf{W})^\top\mathbf{A}+$$

$$\mathbf{A}^\top\nabla\ell(\mathbf{W})\mathbf{B}\operatorname{diag}(\boldsymbol{\psi})\mathbf{B}^\top\nabla\ell(\mathbf{W})^\top\mathbf{A} = 0,$$

$$\mathbf{B}^\top\mathbf{B}\operatorname{diag}(\boldsymbol{\psi})\mathbf{B}^\top\nabla\ell(\mathbf{W})^\top\ell(\mathbf{W})\mathbf{B} - \frac{1}{L\eta}\mathbf{B}^\top\nabla\ell(\mathbf{W})^\top\ell(\mathbf{W})\mathbf{B}+$$

$$\mathbf{B}^\top\nabla\ell(\mathbf{W})^\top\mathbf{A}\operatorname{diag}(\boldsymbol{\phi})\mathbf{A}^\top\nabla\ell(\mathbf{W})\mathbf{B} = 0.$$

Notice that these two equations can be expressed using matrices as

$$\operatorname{diag}\left(\mathbf{A}^\top\mathbf{A}\operatorname{diag}(\boldsymbol{\phi})\mathbf{A}^\top\nabla\ell(\mathbf{W})\ell(\mathbf{W})^\top\mathbf{A}\right) - \frac{1}{L\eta}\operatorname{diag}\left(\|\mathbf{A}^\top\nabla\ell(\mathbf{W})\|_{\mathrm{row}}^2\right)+$$

$$\operatorname{diag}\left(\mathbf{A}^\top\nabla\ell(\mathbf{W})\mathbf{B}\operatorname{diag}(\boldsymbol{\psi})\mathbf{B}^\top\nabla\ell(\mathbf{W})^\top\mathbf{A}\right) = \mathbf{0},$$

$$\operatorname{diag}\left(\mathbf{B}^\top\mathbf{B}\operatorname{diag}(\boldsymbol{\psi})\mathbf{B}^\top\nabla\ell(\mathbf{W})^\top\ell(\mathbf{W})\mathbf{B}\right) - \frac{1}{L\eta}\operatorname{diag}\left(\|\mathbf{B}^\top\nabla\ell(\mathbf{W})^\top\|_{\mathrm{row}}^2\right)+$$

$$\operatorname{diag}\left(\mathbf{B}^\top\nabla\ell(\mathbf{W})^\top\mathbf{A}\operatorname{diag}(\boldsymbol{\phi})\mathbf{A}^\top\nabla\ell(\mathbf{W})\mathbf{B}\right)\Big] = \mathbf{0}.$$

By Lemma 7, we obtain

$$\left((\mathbf{A}^\top\mathbf{A})\odot(\mathbf{A}^\top\nabla\ell(\mathbf{W})\ell(\mathbf{W})^\top\mathbf{A})\right)\boldsymbol{\phi} - \frac{1}{L\eta}\|\mathbf{A}^\top\nabla\ell(\mathbf{W})\|_{\mathrm{row}}^2 + \left(\mathbf{A}^\top\nabla\ell(\mathbf{W})\mathbf{B}\right)^{\circ 2}\boldsymbol{\psi} = \mathbf{0},$$

$$\left((\mathbf{B}^\top\mathbf{B})\odot(\mathbf{B}^\top\nabla\ell(\mathbf{W})^\top\ell(\mathbf{W})\mathbf{B})\right)\boldsymbol{\psi} - \frac{1}{L\eta}\|\mathbf{B}^\top\nabla\ell(\mathbf{W})^\top\|_{\mathrm{row}}^2 + \left(\mathbf{B}^\top\nabla\ell(\mathbf{W})^\top\mathbf{A}\right)^{\circ 2}\boldsymbol{\phi} = \mathbf{0}.$$

Then, we can rewrite these using block matrices as

$$\begin{bmatrix} (\mathbf{A}^\top\mathbf{A})\odot(\mathbf{A}^\top\nabla\ell(\mathbf{W})\nabla\ell(\mathbf{W})^\top\mathbf{A}) & (\mathbf{A}^\top\nabla\ell(\mathbf{W})\mathbf{B})^{\circ 2} \\ (\mathbf{B}^\top\nabla\ell(\mathbf{W})^\top\mathbf{A})^{\circ 2} & (\mathbf{B}^\top\mathbf{B})\odot(\mathbf{B}^\top\nabla\ell(\mathbf{W})^\top\nabla\ell(\mathbf{W})\mathbf{B}) \end{bmatrix}\begin{bmatrix}\boldsymbol{\phi}\\\boldsymbol{\psi}\end{bmatrix}-$$

$$\frac{1}{L\eta}\begin{bmatrix}\|\mathbf{A}^\top\nabla\ell(\mathbf{W})\|_{\mathrm{row}}^2\\\|\mathbf{B}^\top\nabla\ell(\mathbf{W})^\top\|_{\mathrm{row}}^2\end{bmatrix} = \mathbf{0}$$

$$\implies \left[(\mathbf{S}^{A\top}\mathbf{S}^A)\odot(\mathbf{S}^{B\top}\mathbf{S}^B)\right]\begin{bmatrix}\boldsymbol{\phi}\\\boldsymbol{\psi}\end{bmatrix} - \frac{1}{L\eta}\boldsymbol{\lambda} = \mathbf{0}$$

Therefore, the stationary points of (22) are

$$\begin{bmatrix}\boldsymbol{\phi}\\\boldsymbol{\psi}\end{bmatrix} \in \left\{\frac{1}{L\eta}\left[(\mathbf{S}^{A\top}\mathbf{S}^A)\odot(\mathbf{S}^{B\top}\mathbf{S}^B)\right]^\dagger\boldsymbol{\lambda} + \mathbf{v} \;\middle|\; \mathbf{v}\in\operatorname{Null}\left((\mathbf{S}^{A\top}\mathbf{S}^A)\odot(\mathbf{S}^{B\top}\mathbf{S}^B)\right)\right\} := \mathcal{S}$$

It is easy to verify that the null space vector $\mathbf{v}$ will not affect the objective value, and thus one can take any $\mathbf{v}$ to reach the global minimum.

By the conditions in Theorem 5, we have $\mathbf{v}_t \in \mathcal{S}\cap\mathbb{R}^{2r}_+ \subseteq \mathbb{R}^{2r}_+$. As a consequence, $\mathbf{v}_t$ is also the global optimum of the contrained optimization (21). Taking Hadamard square root results in (11), which concludes the proof. $\qquad\square$

### A.4 Moment estimators in adaptive optimizers

Optimizers such as Adam(W) leverages the first and entry-wise second moment estimators of the stochastic gradient to adaptively update the parameters. For LoRA, the parameters are $\mathbf{A}$ and $\mathbf{B}$ (viewed as stochastic matrices), whose corresponding gradient moments are

$$\mathbb{E}[\nabla_{\mathbf{A}}\ell(\mathbf{W}^{\mathrm{pt}} + \mathbf{A}\mathbf{B}^\top)] = \mathbb{E}[\nabla\ell(\mathbf{W})\mathbf{B}], \;\; \mathbb{E}[(\nabla_{\mathbf{A}}\ell(\mathbf{W}^{\mathrm{pt}} + \mathbf{A}\mathbf{B}^\top))^{\circ 2}] = \mathbb{E}[(\nabla\ell(\mathbf{W})\mathbf{B})^{\circ 2}],$$

$$\mathbb{E}[\nabla_{\mathbf{B}}\ell(\mathbf{W}^{\mathrm{pt}} + \mathbf{A}\mathbf{B}^\top)] = \mathbb{E}[\nabla\ell(\mathbf{W})^\top\mathbf{A}], \;\; \mathbb{E}[(\nabla_{\mathbf{B}}\ell(\mathbf{W}^{\mathrm{pt}} + \mathbf{A}\mathbf{B}^\top))^{\circ 2}] = \mathbb{E}[(\nabla\ell(\mathbf{W})^\top\mathbf{A})^{\circ 2}].$$

Given dampening parameters $\beta_1, \beta_2 \in (0, 1)$, the first and second moment estimators $m_t(\cdot)$ and $v_t(\cdot)$ are defined as the exponential moving averages

$$m_t(\nabla\ell(\mathbf{W})\mathbf{B}) = (1 - \beta_1)\nabla\ell(\mathbf{W}_t)\mathbf{B}_t + \beta_1 m_{t-1}(\nabla\ell(\mathbf{W})\mathbf{B})$$

$$= (1 - \beta_1)\sum_{\tau=0}^{t}\beta_1^{t-\tau}\nabla\ell(\mathbf{W}_\tau)\mathbf{B}_\tau, \tag{23a}$$

$$v_t(\nabla\ell(\mathbf{W})\mathbf{B}) = (1 - \beta_2)\big[\nabla\ell(\mathbf{W}_t)\mathbf{B}_t\big]^{\circ 2} + \beta_2 v_{t-1}\nabla\ell(\mathbf{W})\mathbf{B}$$

$$= (1 - \beta_2)\sum_{\tau=0}^{t}\beta_2^{t-\tau}\big[\nabla\ell(\mathbf{W}_\tau)\mathbf{B}_\tau\big]^{\circ 2}, \tag{23b}$$

$$m_t(\nabla\ell(\mathbf{W})^\top\mathbf{A}) = (1 - \beta_1)\sum_{\tau=0}^{t}\beta_1^{t-\tau}\nabla\ell(\mathbf{W}_\tau)^\top\mathbf{A}_\tau, \tag{23c}$$

$$v_t(\nabla\ell(\mathbf{W})^\top\mathbf{A}) = (1 - \beta_2)\sum_{\tau=0}^{t}\beta_2^{t-\tau}\big[\nabla\ell(\mathbf{W}_\tau)^\top\mathbf{A}_\tau\big]^{\circ 2}. \tag{23d}$$

Moreover, these optimizers rely on the following standard assumption characterizing the gradient stochasticity.

**Assumption 3.** *Stochastic gradient samples $\nabla\ell(\mathbf{W}_t)\mathbf{A}_t$ and $\nabla\ell(\mathbf{W}_t)^\top\mathbf{B}_t$ are unbiased and have bounded variance for $\forall t$.*

Under this assumption, it can be readily verified that the moment estimators in (23) are also unbiased and variance-bounded.

Next, we prove the two lemmas in Section 3.2.

**Proof of Lemma 2.**

*Proof.* The proof directly follows from the definition (23). Specifically, it holds

$$m_t(\nabla_{\tilde{\mathbf{A}}}\ell(\mathbf{W})) = m_t(\nabla\ell(\mathbf{W})\tilde{\mathbf{B}}) = m_t(\beta\nabla\ell(\mathbf{W})\mathbf{B})$$

$$= \beta(1 - \beta_1)\sum_{\tau=0}^{t}\beta_1^{t-\tau}\nabla\ell(\mathbf{W}_\tau)\mathbf{B}_\tau$$

$$= \beta m_t(\nabla\ell(\mathbf{W})\mathbf{B}) = m_t(\nabla_{\mathbf{A}}\ell(\mathbf{W})).$$

Similar derivations can be shown for other three moment estimators. $\qquad\square$

**Proof of Lemma 4.**

*Proof.* For the column-wise scaling, its first moment estimator of $\nabla_{\tilde{\mathbf{A}}}\ell(\mathbf{W})$ follows as

$$m_t(\nabla_{\tilde{\mathbf{A}}}\ell(\mathbf{W})) = m_t(\nabla\ell(\mathbf{W})\tilde{\mathbf{B}}) = m_t(\nabla\ell(\mathbf{W})\mathbf{B}\,\mathrm{diag}(\boldsymbol{\beta}))$$

$$= (1 - \beta_1)\sum_{\tau=0}^{t}\beta_1^{t-\tau}\nabla\ell(\mathbf{W}_\tau)\mathbf{B}_\tau\,\mathrm{diag}(\boldsymbol{\beta})$$

$$= m_t(\nabla\ell(\mathbf{W})\mathbf{B})\,\mathrm{diag}(\boldsymbol{\beta}) = m_t(\nabla_{\mathbf{A}}\ell(\mathbf{W}))\,\mathrm{diag}(\boldsymbol{\beta}).$$

And the second moment estimator turns out to be

$$v_t(\nabla_{\tilde{\mathbf{A}}}\ell(\mathbf{W})) = v_t(\nabla\ell(\mathbf{W})\mathbf{B}\,\mathrm{diag}(\boldsymbol{\beta}))$$

$$= (1 - \beta_2)\sum_{\tau=0}^{t}\beta_2^{t-\tau}\big[\nabla\ell(\mathbf{W}_\tau)\mathbf{B}_\tau\,\mathrm{diag}(\boldsymbol{\beta})\big]^{\circ 2}$$

$$= (1 - \beta_2)\sum_{\tau=0}^{t}\beta_2^{t-\tau}\big[\nabla\ell(\mathbf{W}_\tau)\mathbf{B}_\tau\big]^{\circ 2}\,\mathrm{diag}^2(\boldsymbol{\beta})$$

$$= m_t(\nabla\ell(\mathbf{W})\mathbf{B})\,\mathrm{diag}^2(\boldsymbol{\beta}) = m_t(\nabla_{\mathbf{A}}\ell(\mathbf{W}))\,\mathrm{diag}^2(\boldsymbol{\beta}).$$

The same derivations apply to the gradient moment estimators of $\tilde{\mathbf{B}}$. $\qquad\square$

## A.5 USEFUL FACTS

**Lemma 6.** *If* $\|\mathbf{A}^\top \mathbf{G}\|_F > 0$*, then* $\|\mathbf{A}\mathbf{A}^\top \mathbf{G}\|_F > 0$*.*

*Proof.* We prove by contradiction. Suppose $\|\mathbf{A}^\top \mathbf{G}\|_F > 0$ but $\|\mathbf{A}\mathbf{A}^\top \mathbf{G}\|_F = 0$. Then we have $\mathbf{A}^\top \mathbf{G} \neq \mathbf{0}$ and $\mathbf{A}\mathbf{A}^\top \mathbf{G} = \mathbf{0}$. The latter suggests $\mathrm{Col}(\mathbf{A}^\top \mathbf{G}) \subseteq \mathrm{Null}(\mathbf{A})$. Given that $\mathrm{Col}(\mathbf{A}^\top \mathbf{G}) \subseteq \mathrm{Col}(\mathbf{A}^\top)$, we have $\mathrm{Col}(\mathbf{A}^\top \mathbf{G}) \subseteq \mathrm{Null}(\mathbf{A}) \cap \mathrm{Col}(\mathbf{A}^\top) = \{\mathbf{0}\}$, which contradicts $\mathbf{A}^\top \mathbf{G} \neq \mathbf{0}$. This prove is thus completed. □

**Lemma 7** ((Horn & Johnson, 2012))**.** *For matrices* $\mathbf{M}_1, \mathbf{M}_2 \in \mathbb{R}^{m \times n}$*, and vector* $\mathbf{v} \in \mathbb{R}^n$*,*

$$(\mathbf{M}_1 \odot \mathbf{M}_2)\mathbf{v} = \mathrm{diag}(\mathbf{M}_1 \, \mathrm{diag}(\mathbf{v})\mathbf{M}_2^\top).$$

**Theorem 8** ((Schur, 1911); Schur product theorem)**.** *If matrices* $\mathbf{M}_1, \mathbf{M}_2 \succeq 0$*, then* $\mathbf{M}_1 \odot \mathbf{M}_2 \succeq 0$*.*

## B PSEUDOCODES AND COMPLEXITY COMPARISON

Algorithm 1 provides the pseudocodes for our ScaLoRA approach, where AdaOpt refers to one adaptive optimizer step.

---

**Algorithm 1:** Scaled low-rank adaptation (ScaLoRA)

**Input:** Loss $\ell$, pre-trained weight $\mathbf{W}^{\mathrm{pt}}$, maximum iterations $T$, and learning rate $\eta$.
**Initialize:** $\mathbf{A}_0$ and $\mathbf{B}_0$.
1 **for** $t = 0, \ldots, T-1$ **do**
2     Solve $\mathbf{v}_t$ from $\left[(\mathbf{S}_t^{A\top}\mathbf{S}_t^A) \odot (\mathbf{S}_t^{B\top}\mathbf{S}_t^B)\right]\mathbf{v}_t = \boldsymbol{\lambda}_t$;
3     **if** $\mathbf{v}_t \in \mathbb{R}_+^{2r}$ **then**
4         Compute $\boldsymbol{\alpha}_t^*$ and $\boldsymbol{\beta}_t^*$ using Theorem 5;
5         Scale $\tilde{\mathbf{A}}_t = \mathbf{A}_t \, \mathrm{diag}(\boldsymbol{\alpha}_t^*)$, $\tilde{\mathbf{B}}_t = \mathbf{B}_t \, \mathrm{diag}(\boldsymbol{\beta}_t^*)$;
6         Alter moment estimators $m_t$ and $v_t$ using Lemma 4;
7     **else**
8         Compute $\alpha_t^*$ and $\beta_t^*$ using Theorem 3;
9         Scale $\tilde{\mathbf{A}}_t = \alpha_t^* \mathbf{A}_t$, $\tilde{\mathbf{B}}_t = \beta_t^* \mathbf{B}_t$;
10         Alter moment estimators $m_t$ and $v_t$ using Lemma 2;
11     **end**
12     Merge $\mathbf{A}_t \mathbf{B}_t^\top$ and factor out $\tilde{\mathbf{A}}_t \tilde{\mathbf{B}}_t^\top$ using (3);
13     Update $\mathbf{A}_{t+1} = \mathrm{AdaOpt}(\tilde{\mathbf{A}}_t, \eta, m_t, v_t)$, $\mathbf{B}_{t+1} = \mathrm{AdaOpt}(\tilde{\mathbf{B}}_t, \eta, m_t, v_t)$;
14 **end**
**Output:** $\mathbf{A}_T$ and $\mathbf{B}_T$.

---

Table 5 summarizes the theoretical overhead comparison, where $k$ represents for the batch size. Note that the low-rank matrices' Frobenius norms $\|\mathbf{A}_t\mathbf{A}_t^\top \nabla\ell(\mathbf{W}_t)\|_F^2$ and $\|\nabla\ell(\mathbf{W}_t)\mathbf{B}_t\mathbf{B}_t^\top\|_F^2$ in Theorem 3 can be calculated through the trick

$$\|\mathbf{A}_t\mathbf{A}_t^\top \nabla\ell(\mathbf{W}_t)\|_F^2 = \mathrm{tr}\left(\nabla\ell(\mathbf{W}_t)^\top \mathbf{A}_t\mathbf{A}_t^\top \mathbf{A}_t\mathbf{A}_t^\top \nabla\ell(\mathbf{W}_t)\right)$$

$$= \sum_{i=1}^n \sum_{j=1}^r \left[\left((\nabla\ell(\mathbf{W}_t)^\top \mathbf{A}_t)(\mathbf{A}_t^\top \mathbf{A}_t)\right) \odot \left(\nabla\ell(\mathbf{W}_t)^\top \mathbf{A}_t\right)\right]_{ij}$$

which reduces the computational overhead from $\mathcal{O}(m^2 r)$ to $\mathcal{O}((m+n)r^2)$.

Further, ScaLoRA-I guarantees a constant percentage of additional time overhead upon choosing $I = \Omega(r)$, which does not grow with the model hidden size $m$ and $n$. Using the complexity analysis in Table 5, the extra cost of ScaLoRA-I relative to LoRA is $\frac{\mathcal{O}(mnr/I)}{\Omega(kmn)} = \mathcal{O}(1/k)$, where high-order terms are dropped under $r \ll m, n$. This ensures the scalability of ScaLoRA-I to larger models and higher $r$.

Table 5: Additional complexities introduced by LoRA variants

| Method | Time | Space |
|---|---|---|
| LoRA forward/backward | $\Omega(kmn)$ | $\Omega(kmn)$ |
| MoRA | Depends on $f_{\text{compress}}$ and $f_{\text{decompress}}$ | |
| HiRA | $\mathcal{O}(mnr)$ | $\mathcal{O}(mn)$ |
| ScaLoRA | $\mathcal{O}(mnr + (m+n+r)r^2)$ | $\mathcal{O}((m+n+r)r)$ |
| ScaLoRA-I | $\mathcal{O}((mnr + (m+n+r)r^2)/I)$ | $\mathcal{O}((m+n+r)r)$ |

## C  EXPERIMENTAL SETUPS

This section lists the detailed datasets, models, and hyperparameters.

### C.1  PLATFORMS

All the numerical tests are conducted on a server equipped with four Nvidia A100 GPUs. All codes are written in PyTorch (Paszke et al., 2019), and partially built on (Hu et al., 2023; Lion et al., 2025).

### C.2  SETUPS FOR LINEAR REGRESSION

The numerical test considers optimization objective

$$\min_{\mathbf{W}} \frac{1}{2}\|\mathbf{Y} - \mathbf{W}\mathbf{X}\|_{\text{F}}^2$$

where the entries of $\mathbf{X} \in \mathbb{R}^{n \times k}$ and $\mathbf{Y} \in \mathbb{R}^{m \times k}$ are both randomly generated from standard Gaussian $\mathcal{N}(0, 1)$. For LoRA, the objective function is

$$\min_{\mathbf{A},\mathbf{B}} \frac{1}{2}\|\mathbf{Y} - \mathbf{A}\mathbf{B}^\top\mathbf{X}\|_{\text{F}}^2.$$

The test utilizes $m = n = 64$, $k = 100$, and $r = 8$. The optimizer is standard GD.

### C.3  SETUPS FOR NATURAL LANGUAGE UNDERSTANDING

**General Language Understanding Evaluation (GLUE) benchmark** (Wang et al., 2019) is a widely used suite of datasets designed to evaluate the general-purpose natural language understanding (NLU) capabilities of models. In this work, we adopt the following 8 subsets of GLUE:

- **MNLI** (Williams et al., 2018) (Multi-Genre Natural Language Inference) evaluates a model's ability to perform natural language *inference* across multiple genres of text.
- **SST-2** (Socher et al., 2013) (Stanford Sentiment Treebank) is a *sentiment classification* dataset with binary labels.
- **MRPC** (Dolan & Brockett, 2005) (Microsoft Research Paraphrase Corpus) focuses on *paraphrase detection*, i.e., determining whether two sentences are semantically equivalent.
- **CoLA** (Warstadt et al., 2019) (Corpus of Linguistic Acceptability) requires models to determine whether a sentence is *grammatically acceptable*.
- **QNLI** (Rajpurkar et al., 2018) (Question Natural Language Inference) is a question-answering dataset reformulated as a binary *inference* task.
- **QQP**[1] (Quora Question Pairs) consists of pairs of questions, and the task is to predict whether they are semantically equivalent.
- **RTE**[2] (Recognizing Textual Entailment) contains sentence pairs for *textual entailment* classification.

---

[1]https://quoradata.quora.com/First-Quora-Dataset-Release-Question-Pairs
[2]https://paperswithcode.com/dataset/rte

- **STS-B** (Cer et al., 2017) (Semantic Textual Similarity Benchmark) evaluates the degree of *semantic similarity* between two sentences on a continuous scale.

Together, these datasets provide a comprehensive benchmark for testing general-purpose language models under diverse NLU tasks. All datasets are distributed under permissive licenses. A summary of the datasets is provided in Table 6.

Table 6: Summary of GLUE benchmark datasets.

| Name | Task | #train | #test | Metrics |
|------|------|--------|-------|---------|
| MNLI | Natural language inference | 393k | 20k | Matched & mismatched accuracy |
| SST-2 | Sentiment classification | 67k | 1.8k | Accuracy |
| MRPC | Paraphrase detection | 3.7k | 1.7k | Accuracy, F1 |
| CoLA | Acceptability judgment | 8.5k | 1k | Matthews correlation |
| QNLI | QA/NLI | 105k | 5.4k | Accuracy |
| QQP | Paraphrase detection | 364k | 391k | Accuracy, F1 |
| RTE | Textual entailment | 2.5k | 3k | Accuracy |
| STS-B | Semantic similarity | 7k | 1.4k | Pearson & Spearman correlations |

**DeBERTaV3-base** (He et al., 2023) is a transformer-based encoder model with approximately 184M parameters. It builds on the DeBERTa architecture by incorporating disentangled attention and an enhanced masked language modeling objective, leading to improved efficiency and performance across a range of tasks. The publicly available model checkpoint[3] is released under the MIT license.

**Hyperparameters** and general setups for natural language understanding tests follow from the protocols in (Hu et al., 2022; Zhang et al., 2023). Specifically, the LoRA adapters are inserted to all linear layers including `query_proj`, `key_proj`, `value_proj`, `output.dense`, and `intermediate.dense` modules, reducing the number of parameters from 184M to 0.67M. The LoRA rank is set to $r = 4$ with scaling factor $8$ throughout the test. Learning rates are selected via grid search from $\{0.8, 1, 2, 3, 4, 5, 6, 8, 10, 20\} \times 10^{-4}$ for each approach, with finer resolution allocated to the lower end of the range to better capture the region where many methods are more sensitive. For HiRA, the learning-rates are scaled by an additional factor of 10 to offset the magnitude change due to Hadamard product. The the number epochs are reduced due to the fast convergence of ScaLoRA, while other hyperparameters follow the defaults in (Hu et al., 2022); see Table 7.

Table 7: Hyperparameter for natural language understanding tests.

| Hyperparam | CoLA | SST-2 | MRPC | STS-B | QQP | MNLI | QNLI | RTE |
|------------|------|-------|------|-------|-----|------|------|-----|
| LR (LoRA) | 8e-4 | 6e-4 | 8e-4 | 8e-4 | 5e-4 | 2e-4 | 4e-4 | 6e-4 |
| LR (MoRA) | 6e-4 | 6e-4 | 1e-3 | 8e-4 | 5e-4 | 2e-4 | 4e-4 | 6e-4 |
| LR (HiRA) | 6e-3 | 6e-3 | 8e-3 | 8e-3 | 5e-3 | 2e-3 | 4e-3 | 8e-3 |
| LR (ScaLoRA) | 6e-4 | 6e-4 | 1e-3 | 8e-4 | 5e-4 | 2e-4 | 4e-4 | 6e-4 |
| LR scheduler | | | | Linear | | | | |
| Epochs | 10 | 2 | 10 | 10 | 5 | 5 | 3 | 10 |
| Batch size | | | | 32 | | | | |
| Cutoff length | 64 | 128 | 128 | 128 | 320 | 256 | 512 | 320 |
| Warmup steps | 100 | 500 | 10% | 100 | 1000 | 1000 | 500 | 50 |
| Class dropout | 0.1 | 0 | 0 | 0.2 | 0.2 | 0.15 | 0.1 | 0.2 |
| Weight decay | 0 | 0.01 | 0.01 | 0.1 | 0.01 | 0 | 0.01 | 0.01 |

## C.4 SETUPS FOR COMMONSENSE REASONING

**Commonsense reasoning datasets** (Hu et al., 2023) evaluate a model's ability to apply everyday knowledge and make inferences beyond explicitly provided textual information. Such benchmarks

---

[3] https://huggingface.co/microsoft/deberta-v3-base

are essential for assessing reasoning over both physical and social contexts, which remain challenging for language models despite strong performance on surface-level tasks. The datasets considered in this work cover a wide range of commonsense reasoning scenarios:

- **BoolQ** (Clark et al., 2019) (Boolean Questions) is a reading comprehension dataset consisting of yes/no questions. Each question is paired with a passage from Wikipedia, requiring the model to extract and reason over information in the passage to provide the correct binary answer.
- **WG**(Sakaguchi et al., 2021) (WinoGrande) is a large-scale coreference resolution benchmark that mitigates annotation artifacts found in traditional Winograd schemas.
- **PIQA**(Bisk et al., 2020) (Physical Interaction QA) measures knowledge of physical commonsense, particularly intuitive reasoning about how objects interact.
- **SIQA**(Sap et al., 2019) (Social-IQ-A) targets social commonsense reasoning, requiring models to infer motivations, emotions, and social interactions.
- **HS**(Zellers et al., 2019) (HellaSwag) evaluates grounded commonsense inference through multiple-choice sentence completion, designed to be adversarially difficult.
- **ARC**(Chollet, 2019) (AI2 Reasoning Challenge) consists of grade-school science questions, split into **ARC-e** (easy) and **ARC-c** (challenge), based on difficulty levels.
- **OpenbookQA**(Mihaylov et al., 2018) contains multiple-choice science questions that require integrating commonsense with elementary scientific facts, simulating open-book reasoning.

Together, these datasets span multiple domains (physical, social, and scientific reasoning) and provide diverse evaluation challenges. All datasets are publicly available under open or research-friendly licenses. Table 8 provides a detailed summary.

Table 8: Summary of commonsense reasoning datasets.

| Name | Task | #train | #test |
|---|---|---|---|
| WinoGrande | Coreference resolution | 40k | 1.3k |
| PIQA | Physical reasoning | 16k | 3k |
| SIQA | Social reasoning | 33k | 2k |
| HellaSwag | Sentence completion | 70k | 10k |
| ARC-easy | Multiple-choice QA | 2.3k | 1.2k |
| ARC-challenge | Multiple-choice QA | 2.6k | 1.2k |
| OpenbookQA | Open-book QA | 5.0k | 500 |

**LLaMA2-7B** (Touvron et al., 2023) is the second-generation model in the LLaMA family, offering improvements in training stability, data curation, and overall performance compared to its predecessor. The released checkpoint[4] is distributed under a permissive license that supports both academic research and commercial applications.

**LLaMA3-8B** (Grattafiori et al., 2024) pertains to the third-generation LLaMA models, trained with larger and more diverse datasets and incorporating architectural refinements for improved reasoning and instruction-following ability. Its checkpoint[5] is available under Meta's permissive license, likewise allowing both research use and commercial deployment.

**Hyperparameters** for this test are adapted from (Lion et al., 2025). LoRA modules are applied to all projection matrices, including `q_proj`, `k_proj`, `v_proj`, `up_proj`, and `down_proj`. The LoRA rank, scaling factor, and dropout rate are set to 8, 16, and 5%, respectively. We use a batch size of 16 and finetune for 3 epochs across all tasks. The sequence cutoff length is fixed at 256 tokens. Learning rates are reported in Table 9, with a cosine scheduler and 3% warmup steps. Learning rates are selected via a grid search over $\{0.8, 1, 2, 3, 4, 5, 6, 8, 10, 20\} \times 10^{-4}$ using finer resolution in the lower range. The learning-rates of HiRA and LoRA-GA are respectively scaled by an additional factor of 10 and $1/10$ to compensate the magnitude change due to Hadamard product and large initialization. ReLoRA uses a re-initialization frequency of 200 steps with 10 re-warmup

---

[4]https://huggingface.co/meta-llama/Llama-2-7b
[5]https://huggingface.co/meta-llama/Meta-Llama-3-8B

Table 9: Learning rates for commonsense reasoning tasks.

| | Method | BoolQ | PIQA | SIQA | HS | WG | ARC-e | ARC-c | OBQA |
|---|---|---|---|---|---|---|---|---|---|
| **LLaMA2-7B** | LoRA | 8e-4 | 4e-4 | 4e-4 | 4e-4 | 1e-4 | 1e-4 | 4e-4 | 4e-4 |
| | ReLoRA | 8e-4 | 4e-4 | 4e-4 | 4e-4 | 1e-4 | 2e-4 | 4e-4 | 4e-4 |
| | LoRA-GA | 2e-4 | 1e-4 | 1e-4 | 1e-4 | 2e-5 | 8e-5 | 1e-4 | 2e-4 |
| | MoRA | 8e-4 | 4e-4 | 4e-4 | 4e-4 | 1e-4 | 2e-4 | 2e-4 | 4e-4 |
| | HiRA | 8e-3 | 4e-3 | 4e-3 | 2e-3 | 1e-3 | 2e-3 | 4e-3 | 4e-3 |
| | ScaLoRA(-I) | 8e-4 | 2e-4 | 2e-4 | 4e-4 | 1e-4 | 1e-4 | 4e-4 | 4e-4 |
| | LoRA$_{r=32}$ | 8e-4 | 2e-4 | 2e-4 | 2e-4 | 1e-4 | 1e-4 | 2e-4 | 2e-4 |
| **LLaMA3-8B** | LoRA | 4e-4 | 1e-4 | 1e-4 | 1e-4 | 8e-5 | 2e-4 | 2e-4 | 5e-4 |
| | ReLoRA | 4e-4 | 1e-4 | 1e-4 | 1e-4 | 8e-5 | 2e-4 | 2e-4 | 4e-4 |
| | LoRA-GA | 1e-4 | 8e-5 | 6e-5 | 3e-5 | 4e-5 | 5e-5 | 8e-5 | 2e-4 |
| | MoRA | 4e-4 | 1e-4 | 1e-4 | 1e-4 | 8e-5 | 1e-4 | 2e-4 | 2e-4 |
| | HiRA | 8e-3 | 2e-3 | 2e-3 | 1e-3 | 1e-3 | 4e-3 | 8e-3 | 4e-3 |
| | ScaLoRA(-I) | 4e-4 | 1e-4 | 1e-4 | 1e-4 | 8e-5 | 8e-5 | 4e-4 | 5e-4 |
| | LoRA$_{r=32}$ | 4e-4 | 8e-5 | 1e-4 | 1e-4 | 8e-5 | 1e-4 | 2e-4 | 2e-4 |

steps for the three smaller datasets ARC-e, ARC-c, and OBQA, and a frequency of 2000 steps with 100 re-warmup steps for the remaining larger datasets. LoRA-GA employs a scaling factor $\gamma = 128$ for stability, and a sample batch size of 32 for gradient estimation.

## C.5 SETUPS FOR MATHEMATICAL PROBLEM SOLVING

This experiment is conducted by fine-tuning the Gemma-3-12B model on the MetaMathQA dataset and subsequently testing its performance on GSM8K and MATH datasets. Below are brief introductions to the datasets and the model involved.

**MetaMathQA** (Yu et al., 2024) is a synthetic math reasoning dataset released under the Apache-2.0 license and created via question bootstrapping. By rewriting problems through forward, backward, and rephrased perspectives, it augments diversity and improves generalization of mathematical problem-solving models.

**GSM8K** (Grade-School Math 8K) (Cobbe et al., 2021) is released under the MIT license and consists of roughly 8.5K high-quality, linguistically varied word problems from middle-school curricula, each requiring multiple reasoning steps. It is designed to be solvable by bright students and serves as a standard benchmark for evaluating multi-step mathematical reasoning.

**MATH** (Hendrycks et al., 2021) is also released under the MIT license and includes about 12.5K high-school competition–style math problems across topics such as algebra, number theory, geometry, and probability. Each problem is paired with a detailed step-by-step solution, challenging language models with complex mathematical reasoning tasks.

**Gemma-3-12B-pt** (Team et al., 2025) is a 12-billion parameter multimodal language model developed by Google DeepMind. It is part of the Gemma-3 family, which includes models from 1B to 27B parameters, optimized for tasks such as question answering, summarization, and reasoning. The model checkpoint[6] is released under Google's Gemma Term of Use[7], permitting both research and commercial applications.

**Hyperparameters** are similar to the previous commonsense reasoning test. Specifically, LoRA modules are applied to all projection matrices; i.e., q_proj, k_proj, v_proj, up_proj, and down_proj. The LoRA rank, scaling factor, and dropout rate are set to 8, 16, and 5%, respectively. Given the large dataset size, the batch size is increased to 64, while the number of training epochs is reduced to 2. The sequence length is capped at 256 tokens, and the learning rate is fixed at $10^{-4}$.

---

[6] https://huggingface.co/google/gemma-3-12b-pt
[7] https://ai.google.dev/gemma/terms

# D    ADDITIONAL NUMERICAL RESULTS

## D.1    MOTIVATION FOR SCALORA-I

Figure 4 visualizes the deviation of the scaling factors $\alpha_t$ and $\beta_t$ from 1 when applying optimal scaling at each iteration, with DeBERTaV3-base model and CoLA dataset. It is seen that these deviations are below 0.1 and 0.2 for most iterations, which is expected given the relatively small learning rate $\eta$. Since $\mathbf{A}_t$ and $\mathbf{B}_t$ thereby change only slightly, it is natural to consider a lazy update strategy that performs the scaling after sufficient changes have accumulated. Moreover, the figure also shows that $\mathbf{B}_t$ requires noticeably larger adjustments than $\mathbf{A}_t$, consistent with the empirical findings and theoretical analyses in (Zhu et al., 2024).

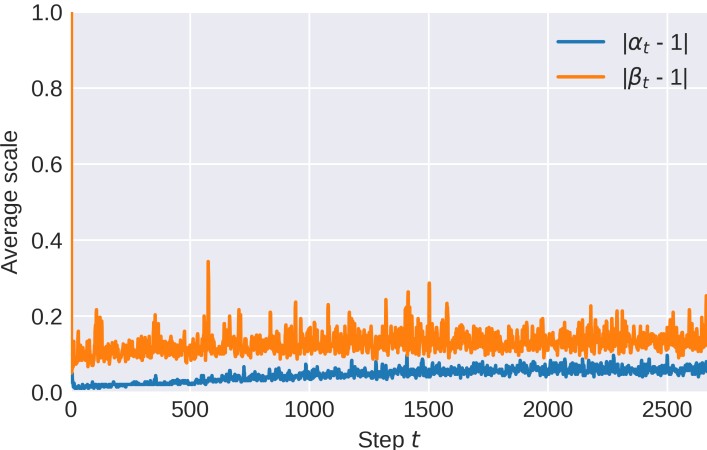

Figure 4: Visualization of scaling factor change during fine-tuning.

## D.2    ABLATION STUDY ON CHOICE OF $I$

Next, ablation experiment on the choice of $I$ is conducted using LLaMA3-8B on the ARC-c dataset, where increasing the rank to 32 yields a remarkable improvement in LoRA. To evaluate the impact of $I$ on the effectiveness and convergence, we report the test accuracy, the running average of fine-tuning loss, and the elapsed time relative to LoRA for $I \in \{1, 3, 10, 30, 100\}$. The results are summarized in Table 10. As $I$ increases, accuracy and time complexity both decrease, while the fine-tuning loss tends to grow. Notably, $I = 10$ provides a good trade-off between loss reduction and computational cost. In particular, it achieves convergence comparable to $I = 1$ yet introducing only a $4\%$ additional overhead relative to LoRA.

Table 10: Ablation study on the choice of $I$ using LLaMA3-8B on ARC-c task.

| Method | Acc | FT loss | Time |
|---|---|---|---|
| ScaLoRA $I = 1$ | 65.61 | 0.8693 | 1.42× |
| ScaLoRA $I = 3$ | 65.14 | 0.8734 | 1.15× |
| ScaLoRA $I = 10$ | 64.68 | 0.8705 | 1.04× |
| ScaLoRA $I = 30$ | 63.57 | 0.8960 | 1.02× |
| ScaLoRA $I = 100$ | 63.33 | 0.9851 | 1.01× |
| LoRA $r = 8$ | 62.29 | 1.2013 | 1× |
| LoRA $r = 32$ | 64.08 | 0.866 | 1.08× |

## D.3    EXTENDED ABLATION STUDY ON EFFECTIVENESS OF COLUMN SCALING

This subsection investigates the effectiveness of Theorem 5 through a more detailed comparison. Following the setup in Section 4.4, we analyze the ScaLoRA-I variant that uses only scalar scaling.

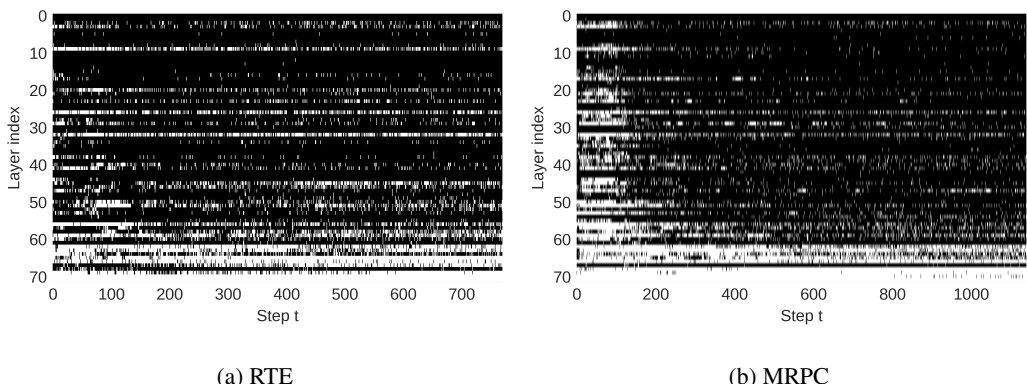

(a) RTE            (b) MRPC

Figure 5: Patterns of column scaling across layers.

Table 11 compares this scalar-only variant against ScaLoRA-I on commonsense reasoning benchmarks using LLaMA2-7B with $r = 8$. We observe that the scalar-only variant suffers a notable performance degradation on the SIQA, WG, ARC-c, and OBQA datasets, while performing comparably to ScaLoRA-I on the remaining four. Overall, this results in an average performance drop of $0.72\%$, though it still exceeds LoRA by $0.60\%$. This underscores the significance and effectiveness of column-wise scaling.

Table 11: Ablation study on column scaling using LLaMA2-7B on commonsense reasoning tasks.

| Method | BoolQ | PIQA | SIQA | HS | WG | ARC-e | ARC-c | OBQA | Avg |
|---|---|---|---|---|---|---|---|---|---|
| LoRA | 87.40 | 81.66 | 59.16 | 82.45 | 79.48 | 82.91 | 57.59 | 58.40 | 73.63 |
| ScaLoRA-I | **87.58** | 82.26 | **60.49** | 83.52 | **81.69** | **83.75** | **58.53** | **60.20** | **74.75** |
| Scalar-only | 87.31 | **82.32** | 59.37 | **83.60** | 80.93 | 83.38 | 56.11 | 59.20 | 74.03 |

### D.4 PATTERNS OF LAYERS WITH COLUMN SCALING

Interestingly, the layers satisfying $\mathbf{v}_t \succeq 0$ exhibit discernible patterns, with certain layers more prone than others to violating this condition. Figure 5 depicts these patterns for DeBERTaV3-base on two GLUE tasks, where column and scalar scaling are marked in black and white, respectively. We observe that some layers are consistently transformed using column scaling, while others predominantly undergo scalar scaling. This pattern also varies across tasks. In practice, when such patterns are known a priori, one may fix the scaling scheme accordingly to eliminate the computational overhead for solving the $2r \times 2r$ linear system.

