# OpenReview forum: "ScaLoRA: Optimally Scaled Low-Rank Adaptation for Efficient High-Rank Fine-Tuning"
_ICLR.cc/2026/Conference — Submitted to ICLR 2026_

### Official Review · Reviewer_yyfD · 2025-10-26

**Soundness:** 3
**Presentation:** 3
**Contribution:** 2
**Rating:** 6
**Confidence:** 3

**Summary:**

The paper studies cumulative LoRA for fine-tuning models, that is the low rank updates are frequently merged into the base model in order to increase the overall rank of the adapters. This problem poses the question of how to initialize the low rank matrices after each merge. This paper provides a theoretical analysis of the initialization. Assuming the smoothness of the loss function, and minimizing the upper bound for the function progress via smoothness as a proxy problem, the paper shows that the optimal initialization is obtained by truncating the SVD of the gradient matrix. The paper then focuses on the initialization by scaling the previous matrices in two cases: scalar and column scaling. In both cases, the authors derive the optimal scaling. In experiment, the paper show the performance gain of ScaLoRA in various tasks as well as the runtime and memory requirements in comparison with existing methods.

**Strengths:**

- The strength of the paper lies in a rigorous analysis of the methods for initializing A and B matrices when restarting. In all settings the paper studies, the authors are able to derive the optimal strategies.
- In experiment, the proposed methods show a notable performance gain. Specifically, the variant with I=10 performs better than the other high-rank methods (HiRA, MoRA), on the Commonsense reasoning baseline with much better overhead.
- The presentation of the paper is overall easy to follow.

**Weaknesses:**

- The main weakness of the paper is that the idea of initialization of the A, B matrices using gradient approximation is known from LoRA-GA (Wang et al. 2024). While Wang et al. 2024 doesn't show both the sufficient and necessary conditions, they have started the main idea. That said, to my knowledge, column scaling is novel.
- The vanilla ScaLoRA has a very high time overhead
- Another limitation is the algorithm is tested with small r (the high rank baseline is LoRA with r=32).
- One missing baseline: I think the authors should add the baseline which uses Theorem 1 and does the merging after I iterations, for a suitable I (ie, cumulative LoRA + LoRA-GA for initialization)
- A few minor points: Line 160: what do the authors mean by "directly minimizing the loss function is generally infeasible"; don't we usually optimize the loss function directly? Line192: L is used for smoothness but also the time, which might be confusing.

**Questions:**

- Please see the weaknesses. I suggest adding experiments with higher rank baselines and the baseline with LoRA-GA for initialization.

---

> ### Author Response · Authors · 2025-11-21
>
> We thank the reviewer for the thoughtful comments provided. The weaknesses are addressed point-by-point in the following.
>
> ### Response to W1
>
> While LoRA-GA aligns the weight update with full fine-tuning gradient at initialization $t=0$, our ScaLoRA focuses on aligning the low-rank update with the evolving gradient in the subsequent iterations $t \ge 1$. As suggested by the reviewer, LoRA-GA has been added to Table 2 and Figure 3 of the revised manuscript for both performance and overhead comparisons. A snippet of the updated table is provided below, where LoRA-GA demonstrates 0.71% (LLaMA2-7B) and 0.39% (LLaMA3-8B) average performance gains over LoRA with $r=8$, though it remains slightly below LoRA with $r=32$. In addition, ScaLoRA(-I) further showcases 0.41% and 0.63% average improvements over LoRA-GA, which underscores the benefits of adjusting the low-rank adapters every (a few) iteration.
>
> **LLaMA2-7B**
>
> | Method      | BoolQ | PIQA  | SIQA  |  HS   |  WG   | ARC-e | ARC-c | OBQA  |    Avg    |
> | ----------- | :---: | :---: | :---: | :---: | :---: | :---: | :---: | :---: | :-------: |
> | LoRA        | 87.40 | 81.66 | 59.16 | 82.45 | 79.48 | 82.91 | 57.59 | 58.40 |   73.63   |
> | *LoRA-GA*   | 87.92 | 83.03 | 60.13 | 82.30 | 82.87 | 83.25 | 56.83 | 58.40 |   74.34   |
> | ScaLoRA     | 87.77 | 82.43 | 60.08 | 83.43 | 82.08 | 83.54 | 58.11 | 58.60 |   74.51   |
> | ScaLoRA-I   | 87.58 | 82.26 | 60.49 | 83.52 | 81.69 | 83.75 | 58.53 | 60.20 | **74.75** |
> | LoRA (r=32) | 88.29 | 82.70 | 60.54 | 83.15 | 82.00 | 82.79 | 57.68 | 59.00 |   74.52   |
>
> **LLaMA3-8B**
>
> | Method      | BoolQ | PIQA  | SIQA  |  HS   |  WG   | ARC-e | ARC-c | OBQA  |    Avg    |
> | ----------- | :---: | :---: | :---: | :---: | :---: | :---: | :---: | :---: | :-------: |
> | LoRA        | 88.99 | 85.09 | 60.95 | 86.09 | 82.64 | 86.62 | 62.29 | 62.00 |   76.83   |
> | *LoRA-GA*   | 89.69 | 84.98 | 61.00 | 86.58 | 85.32 | 86.11 | 62.29 | 61.80 |   77.22   |
> | ScaLoRA     | 89.20 | 86.18 | 61.82 | 86.51 | 84.53 | 86.57 | 65.61 | 62.40 | **77.85** |
> | ScaLoRA-I   | 89.14 | 86.07 | 62.33 | 86.48 | 83.35 | 86.53 | 64.68 | 62.00 |   77.57   |
> | LoRA (r=32) | 89.69 | 85.47 | 61.72 | 86.76 | 83.35 | 87.08 | 64.08 | 62.20 |   77.54   |
>
> ### Response to W2
>
> In practice, it is recommended to use ScaLoRA-I for enhanced scalability. It guarantees a constant percentage of additional time overhead upon choosing $I = \Omega (r)$. According to the complexity analysis in Table 5 of Appendix B, the extra cost of ScaLoRA-I relative to LoRA is $\frac{\mathcal{O}(mnr/I)}{\Omega(kmn)} = \mathcal{O}(1/k)$, where high-order terms are omitted given $r \ll m,n$, and $k$ denotes the input batch size. Notably, this overhead does not grow with the model hidden dimension $m$ and $n$, thus ensuring the scalability and efficiency to larger models. To validate this claim, we scaled our approach to Gemma-3-27B-pt with $r=8$. On a single H100 GPU, LoRA achieved a throughput of $0.58$ it/s with $64.28$ GB memory usage, whereas ScaLoRA-I with $I=10$ showed $0.56$ it/s ($0.97\times$ LoRA) efficiency and the same memory footprint of $64.28$ GB. Extended numerical tests have been added to Appendix D to better motivate ScaLoRA-I, and to justify the choice of $I$ through ablation studies.
>
> ### Response to W3
>
> We intentionally limit the fitting capacity of LoRA and its variants by setting a small $r$. This setup emulates more challenging scenarios where higher ranks are necessitated to adequately capture the underlying task structure. We follow the reviewer’s recommendation to evaluate ScaLoRA-I using LLaMA2-7B with $r = 32$, and Reviewer hHtD’s suggestion to compare it against the $r=32$ results reported by PoLAR [1]. To ensure $I = \Omega (r)$, we have scaled it from 10 to 40 accordingly. As shown in the table, ScaLoRA-I achieves consistently strong performance across all eight tasks and surpasses PoLAR by 1.07% on average, highlighting its effectiveness in higher-rank configurations. Moreover, the time overhead of ScaLoRA-I remains $1.047\times$ of LoRA, which aligns with our analysis above.
>
> | Method    | BoolQ | PIQA  | SIQA  |  HS   |  WG   | ARC-e | ARC-c | OBQA  |    Avg    |
> | --------- | :---: | :---: | :---: | :---: | :---: | :---: | :---: | :---: | :-------: |
> | LoRA      | 87.89 | 81.56 | 59.06 | 82.51 | 72.61 | 82.37 | 56.83 | 54.60 |   72.18   |
> | DoRA      | 87.61 | 81.45 | 58.70 | 82.50 | 74.43 | 82.28 | 57.17 | 55.60 |   72.47   |
> | PoLAR     | 88.13 | 82.64 | 60.03 | 83.12 | 82.00 | 81.99 | 56.14 | 55.60 |   73.71   |
> | ScaLoRA-I | 87.80 | 82.48 | 60.18 | 83.28 | 82.72 | 83.12 | 58.45 | 60.20 | **74.78** |
>
> [1] K. Lion, L. Zhang, B. Li, and N. He, “PoLAR: Polar-Decomposed Low-Rank Adapter Representation,” in *NeurIPS*, 2025.

---

> ### Author Response · Authors · 2025-11-21
>
> ### Response to W4
>
> Unfortunately, adopting LoRA-GA for every re-initialization step of ReLoRA would result in prohibitively high computational cost, due to the substantial SVD overhead. In our attempts, it takes over 1 minute to perform LoRA-GA on LLaMA2-7B with an A100 GPU; see also Table 5 of the LoRA-GA paper. By contrast, ScaLoRA(-I) relies on lightweight transformations (i.e., column and scalar scaling) that can be efficiently calculated within 0.2 second. For this reason, LoRA-GA is more suitable for initialization, rather than repeated use within ReLoRA.
>
> ### Response to W5
>
> To improve clarity, we have revised the sentence at Line 160 to: “Due to the nonlinearity of LLMs, the global optimum of the loss function is analytically infeasible.” In addition, we have replaced the iteration symbol $L$ with $S$ in Line 192 to avoid confusion with the Lipschitz smoothness constant. Thank you for pointing these issues out.

---

### Official Review · Reviewer_skhi · 2025-10-26

**Soundness:** 2
**Presentation:** 3
**Contribution:** 2
**Rating:** 4
**Confidence:** 4

**Summary:**

This paper introduces ScaLoRA, a theoretically grounded extension of LoRA that aims to recover high-rank fine-tuning behavior while preserving parameter efficiency. The key idea is to progressively accumulate high-rank updates by optimally scaling LoRA’s low-rank adapters at each iteration (or periodically, in ScaLoRA-I). The authors derive closed-form optimal scaling rules for scalar and column-wise transformations, ensuring analytical tractability and compatibility with adaptive optimizers like AdamW, avoiding the need for restarts. Experiments across model families and tasks show consistent but modest gains over LoRA, HiRA, and MoRA, with ScaLoRA-I maintaining LoRA level runtime and compute efficiency. The work is mathematically neat and clearly written, though the empirical impact remains relatively limited.

**Strengths:**

`Strong theoretical foundation`

The derivations are rigorous and provide analytical insight into optimal scaling for low-rank adapters, bridging LoRA’s empirical intuition with optimization theory.


`Practical implementation design`

The method integrates well with adaptive optimizers without restarting gradient statistics.


`Consistent performance and fast convergence`

The approach improves over LoRA and some recent high-rank variants (HiRA, MoRA) and achieves similar performance to higher-rank LoRA models with fewer parameters.


`Clear complexity and efficiency analysis`
The paper transparently reports computational and memory tradeoffs, and ScaLoRA-I effectively achieves LoRA-level runtime and compute efficiency.

**Weaknesses:**

`W1: Marginal empirical gains relative to complexity`

While ScaLoRA improves over LoRA and its recent high-rank variants (HiRA, MoRA), the differences are consistently minor.In many cases (e.g., Table 2, LLaMA-3-8B results), the rank-8 ScaLoRA matches rank-32 LoRA, but the additional complexity and theoretical machinery do not clearly justify the incremental benefit in using rank-8 vs rank-32 adapters (since the memory usage would be very similar in this case). This weakens the practical motivation: practitioners may prefer slightly higher LoRA ranks rather than adopting a more complicated method.


`W2: Lack of systematic analysis for iteration frequency (I)`

The intermittent scaling variant, ScaLoRA-I, is an important part of the paper’s efficiency argument, yet its behavior is not thoroughly studied. Without understanding how performance varies with I, it is unclear how sensitive the method is or how one should set this hyperparameter in practice.


`W3: Evaluation focuses on tasks with small LoRA–FT gap`

The chosen benchmarks are domains where LoRA already performs very close to full fine-tuning, leaving limited room to showcase ScaLoRA’s benefits. On these tasks, even basic LoRA performs more or less like full FT. The authors’ claim that ScaLoRA better approximates full fine-tuning would be more convincing if tested on harder domains, where LoRA’s performance drop is more substantial.

**Questions:**

- Q1: Can you please provide experiments comparing ScaLoRA with ReLoRA and ABBA, which also aim to recover high-rank or high-expressive updates under a PEFT setting?


- Q2: Can you please expand the discussion comparing ScaLoRA with LoRA-Pro and LoRA-SB? I would like to understand how the approaches differ conceptually - the 2 mentioned papers aim to minimize deviation between LoRA and full fine-tuning updates.


- Q3: Can you please add ablations on the iteration number I to show how scaling frequency affects accuracy and convergence speed?


- Q4: Can you please evaluate on a task where LoRA performs much worse than full fine-tuning to better demonstrate ScaLoRA’s effectiveness in approximating full FT?

---

I would be happy to increase my score if the authors are able to clarify my questions.

---

> ### Author Response · Authors · 2025-11-21
>
> We appreciate the reviewer’s insightful suggestions. In the following, we respond to each weaknesses and questions in turn.
>
> ### R1. Empirical gains and complexity (response to W1)
>
> ScaLoRA(-I) demonstrates a significantly larger empirical gain than both MoRA and HiRA over vanilla LoRA. For instance, while HiRA yields performance improvements of 0.32% (LLaMA2-7B) and 0.63% (LLaMA3-8B) over LoRA, ScaLoRA(-I) achieves 1.12% and 1.02% gains, respectively. As pointed out in W3, the advantages of ScaLoRA over high-rank LoRA become more pronounced on complex tasks, for which a sufficiently high rank is required but incurs prohibitive time and memory overhead. Results are provided in R3 to support this statement.
>
> ### R2. Ablation study for frequency $I$ (response to W2 and Q3)
>
> Following the reviewer’s suggestion, ablation tests for the impact of $I$ on the effectiveness and convergence of ScaLoRA-I has been incorporated into Appendix D.2. A copy of the key results are attached here, where the test is performed with LLaMA3-8B on the ARC-c dataset. We report the test accuracy, the running average of fine-tuning loss, and the elapsed time relative to LoRA for $I \in \{ 1,3,10,30,100 \}$. As $I$ increases, both accuracy and time overhead decrease, while the fine-tuning loss tends to grow, implying slower convergence. Notably, $I=10$ provides a favorable trade-off between loss reduction and computational cost. In particular, it achieves convergence comparable to $I = 1$ yet introducing only a $4\%$ additional overhead relative to LoRA. In practice, a recommended choice is $I = \Omega (r)$, which ensures a constant percentage of additional time overhead and thus the desired scalability; see analysis in Appendix B.
>
> | Method        |  Acc  | FT loss | Time  |
> | ------------- | :---: | :-----: | :---: |
> | ScaLoRA I=1   | 65.61 | 0.8693  | 1.42x |
> | ScaLoRA I=3   | 65.14 | 0.8734  | 1.15x |
> | ScaLoRA I=10  | 64.68 | 0.8705  | 1.04x |
> | ScaLoRA I=30  | 63.57 | 0.8960  | 1.02x |
> | ScaLoRA I=100 | 63.33 | 0.9851  | 1.01x |
> | LoRA r=8      | 62.29 | 1.2013  |  1x   |
> | LoRA r=32     | 64.08 |  0.866  | 1.08x |
>
> ### R3. More challenging tasks (response to W3 and Q4)
>
> While we are unable to perform a comprehensive test on more challenging tasks due to the limited rebuttal time, a workaround is to further reduce $r$ to extremely small values (e.g., 1) in the commonsense reasoning tasks, forcing LoRA to operate under severe capacity constraints. This setup reliably induces large performance degradation in LoRA, making it a good proxy for task difficulty.
>
> Below we report the rank-one results on a subset of tasks. ScaLoRA is implemented with $I = \Omega (r) = 1$, adding marginal (<5%) extra time complexity over LoRA. From the table, LoRA exhibits a large performance drop (2.78% average loss compared to $r=32$), whereas ScaLoRA recovers accuracy close to the high-rank baseline.
>
> | Method  | BoolQ | ARC-e | ARC-c | OBQA  |  Avg  |
> | ------- | :---: | :---: | :---: | :---: | :---: |
> | LoRA    | 87.12 | 85.06 | 61.13 | 58.60 | 72.98 |
> | ScaLoRA | 88.87 | 86.20 | 63.15 | 61.60 | 74.96 |

---

> ### Author Response · Authors · 2025-11-21
>
> ### R4. Additional baselines (response to Q1)
>
> In response to the requests from multiple reviewers, we have augmented our evaluation with LoRA-GA and ReLoRA in Table 2 and Figure 3 of the revised paper. ABBA is temporarily omitted due to its conceptual similarity to HiRA and the time constraints. As the updated table (attached below) shows, while both ReLoRA and LoRA-GA offer a strong performance boost over vanilla LoRA, their results remain slightly below the high-rank LoRA with $r=32$. In comparison, ScaLoRA(-I) consistently outperforms all of these variants, demonstrating its effectiveness across different settings.
>
> **LLaMA2-7B**
>
> | Method      | BoolQ | PIQA  | SIQA  |  HS   |  WG   | ARC-e | ARC-c | OBQA  |    Avg    |
> | ----------- | :---: | :---: | :---: | :---: | :---: | :---: | :---: | :---: | :-------: |
> | LoRA        | 87.40 | 81.66 | 59.16 | 82.45 | 79.48 | 82.91 | 57.59 | 58.40 |   73.63   |
> | ReLoRA      | 87.80 | 82.48 | 60.08 | 83.23 | 82.56 | 82.95 | 58.11 | 58.00 |   74.40   |
> | LoRA-GA     | 87.92 | 83.03 | 60.13 | 82.30 | 82.87 | 83.25 | 56.83 | 58.40 |   74.34   |
> | MoRA        | 87.49 | 82.54 | 59.88 | 82.56 | 79.08 | 83.59 | 58.02 | 57.40 |   73.82   |
> | HiRA        | 87.71 | 82.97 | 59.83 | 83.38 | 81.69 | 82.83 | 55.55 | 57.60 |   73.95   |
> | ScaLoRA     | 87.77 | 82.43 | 60.08 | 83.43 | 82.08 | 83.54 | 58.11 | 58.60 |   74.51   |
> | ScaLoRA-I   | 87.58 | 82.26 | 60.49 | 83.52 | 81.69 | 83.75 | 58.53 | 60.20 | **74.75** |
> | LoRA (r=32) | 88.29 | 82.70 | 60.54 | 83.15 | 82.00 | 82.79 | 57.68 | 59.00 |   74.52   |
>
> **LLaMA3-8B**
>
> | Method      | BoolQ | PIQA  | SIQA  |  HS   |  WG   | ARC-e | ARC-c | OBQA  |    Avg    |
> | ----------- | :---: | :---: | :---: | :---: | :---: | :---: | :---: | :---: | :-------: |
> | LoRA        | 88.99 | 85.09 | 60.95 | 86.09 | 82.64 | 86.62 | 62.29 | 62.00 |   76.83   |
> | ReLoRA      | 89.20 | 85.64 | 60.13 | 85.99 | 85.24 | 86.95 | 63.14 | 61.80 |   77.26   |
> | LoRA-GA     | 89.69 | 84.98 | 61.00 | 86.58 | 85.32 | 86.11 | 62.29 | 61.80 |   77.22   |
> | MoRA        | 88.56 | 86.18 | 60.29 | 86.69 | 82.40 | 87.79 | 64.08 | 62.20 |   77.27   |
> | HiRA        | 88.87 | 86.07 | 60.64 | 86.11 | 84.53 | 87.12 | 63.91 | 62.40 |   77.46   |
> | ScaLoRA     | 89.20 | 86.18 | 61.82 | 86.51 | 84.53 | 86.57 | 65.61 | 62.40 | **77.85** |
> | ScaLoRA-I   | 89.14 | 86.07 | 62.33 | 86.48 | 83.35 | 86.53 | 64.68 | 62.00 |   77.57   |
> | LoRA (r=32) | 89.69 | 85.47 | 61.72 | 86.76 | 83.35 | 87.08 | 64.08 | 62.20 |   77.54   |
>
> ### R5. Comparison to LoRA-Pro and LoRA-SB (response to Q2)
>
> Although all the three methods reduce the gradient discrepancy between LoRA and full fine-tuning, their goals and mechanisms are markedly distinct. Specifically, LoRA-SB is an initialization-based approach tailored to LoRA-XS, which freezes $\mathbf{A},\mathbf{B}$ and fine-tunes only a $r \times r$ matrix between them. LoRA-Pro aims to narrow the gap to full fine-tuning by replacing the gradients induced by backpropagation with optimizable matrices, but the resultant weight update remains low-rank. By contrast, ScaLoRA’s primary goal is to increase the rank of weight update by accumulating adapters across varying directions, while retaining standard backpropagation for optimization.

---

> > ### Comment · Reviewer_skhi · 2025-11-25
> >
> > After carefully reviewing the authors’ responses and the subsequent discussions, I have decided to maintain my original score of 4. My primary concern is that the real empirical advantages of ScaLoRA over existing methods have not been convincingly demonstrated. In addition, the reliance on the frequency parameter I introduces an extra tuning burden, affecting the trade-off between performance and wall-clock time and potentially limiting practical applicability.
> >
> > Although ScaLoRA shows improvements over LoRA at rank = 1, I am not fully convinced that these gains will generalize to higher ranks on more challenging tasks (which is the more practical setting). If the scoring scale allowed finer granularity, I would place my score closer to a 5, as I remain quite borderline regarding this paper.

---

> > > ### Author Response · Authors · 2025-11-26
> > >
> > > Thank you for the prompt reply. We would like to further clarify the choice of frequency parameter I. In fact, ScaLoRA-I guarantees a constant percentage of additional time overhead upon choosing $I = \Omega (r)$. Using the complexity analysis in Table 5 of Appendix B, the extra cost of ScaLoRA-I relative to LoRA is $\frac{\mathcal{O}(mnr/I)}{\Omega(kmn)} = \mathcal{O}(1/k)$, where high-order terms are omitted given $r \ll m,n$, and $k$ denotes the input batch size. Notably, this overhead does not grow with the model hidden dimension $m$ and $n$, thus ensuring the scalability and efficiency to larger models. That said, $I$ can be readily selected without additional tuning burden or impact on runtime efficiency.
> > >
> > > To validate this empirically, we scaled to Gemma-3-27B-pt with $r=8$. On a single H100 GPU, LoRA achieved a throughput of $0.58$ it/s with $64.28$ GB memory usage, while ScaLoRA-I with $I=10$ reached $0.56$ it/s ($0.97\times$ LoRA) efficiency and the same memory footprint.
> > >
> > > We kindly ask the reviewer to consider raising the score if you are currently borderline but **leaning toward acceptance**.

---

> ### Comment · Reviewer_skhi · 2025-11-27
>
> Thank you for the clarification. After revisiting your response and results, I am largely convinced that the choice of I is not as problematic as initially thought. I encourage the authors to clarify this point in the manuscript as well, since this choice is important for practical implementation. The empirical results are solid (although not exceptional).
>
> Since the paper is **currently at the borderline** and my **major concerns have been addressed**, I am changing my recommendation to accept.

---

> > ### Author Response · Authors · 2025-11-27
> >
> > Thank you for raising the score and the insightful suggestions! The discussion on the choice of $I$ has been incorporated into Appendix B of the revised manuscript.

---

### Official Review · Reviewer_2F1U · 2025-10-29

**Soundness:** 2
**Presentation:** 2
**Contribution:** 2
**Rating:** 2
**Confidence:** 3

**Summary:**

This paper proposes a method for fine-tuning LLMs through iterative merging of low-rank matrices. At each step, an alternative low-rank matrix is computed by scaling the standard LoRA components with either a scalar or a diagonal matrix. The scaling coefficients are determined by minimizing a quadratic upper bound using the standard smoothness assumption. Numerical experiments have been reported on several standard fine-tuning benchmarks, demonstrating its effectiveness.

**Strengths:**

The paper aims to address the limitations of LoRA and has a clear motivation.

**Weaknesses:**

We typically have three standard classes of memory-efficient optimization approaches:

1. LoRA style methods: These treat A and B as two fixed learnable parameters and directly optimize them using an optimizer. This approach is memory-efficient during both training and model storage, particularly when fine-tuning multiple task-specific models.

2. Memory-efficient optimizers. Methods that modify the optimization procedure itself to reduce memory usage. A closely related work is Chain of LoRA [1]: At each iteration $t \ge 0$, we aim to approximately solve the subproblem: $\min_{A_t,B_t}\{f(W_t+A_t B_t^T)\}$ where $W_{t+1} = W_t + A_t B_t^T$.

This paper seems to follow the second style. However, the whole logic and the proposed methodology appear to be somewhat confusing and unclear.

1. Unlike eqn (1), it is unclear why $A_t$ and $B_t$ are updated using eqn (4).
2. In the equation in the middle between (6) and (7), it is unclear why consider minimizing the upper bound of $\ell (W_t + \Delta \tilde{W}_t)$ instead of the one for $\ell (W_t +  \Delta W_t)$.
3. Following 2, it seems the paper wants to update $\Delta W_t \approx - \frac{1}{L} \nabla \ell (W_t)$, then the motivation of introducing $\tilde{A}$ and $\tilde{B}$ is unclear to me. Moreover, ideally, one might expect an update of the form $\Delta W_t \approx -\eta_t m_t$ where $m_t$ is the momentum as if we do the full fine-tuning using, e.g., AdamW.
4. Unlike LoRA, in this scenario, $\tilde{A}$ and $\tilde{B}$ are not fixed parameters to be minimized. The meaning of momentum here is unclear to me.
5. The setting is different from LoRA since we have to store the full final model instead of $\Delta W$, which is a low-rank matrix as in LoRA. Indeed, we should treat the proposed algorithm as a memory-efficient optimization method. Then, numerically, it appears that some important baseline methods seem missing for comparisons, e.g., Chain of LoRA, GaLoRA[2], etc.






[1] Chain of LoRA: Efficient Fine-tuning of Language Models via Residual Learning, ICML 2024.
[2] GaLore: Memory-Efficient LLM Training by Gradient Low-Rank Projection, ICML 2024.

**Questions:**

Please see the weakness above.

---

> ### Author Response · Authors · 2025-11-21
>
> We thank the reviewer’s constructive comments. Next, the concerns are addressed one by one.
>
> ### Regarding W1
>
> Eq. (4) follows directly from Eq. (3), which first merges $\mathbf{A}_ t, \mathbf{B}_ t$ and then performs standard GD on the alternative $\tilde{\mathbf{A}}_ t, \tilde{\mathbf{B}}_ t$. The updated matrices become $\mathbf{A}_ {t+1}$ and $\mathbf{B}_ {t+1}$ without tilde because, in step $t+1$ we can again identify a new alternative low-rank matrix $\tilde{\mathbf{A}}_ {t+1} \tilde{\mathbf{B}}_ {t+1}^\top$ to replace $\mathbf{A}_ {t+1} \mathbf{B}_ {t+1}^\top$. Throughout the paper the tilde symbol is used exclusively for the optimized replacement constructed within the iteration. A simple way to help interpret this is to substitutes $\tilde{\mathbf{A}}_ t, \tilde{\mathbf{B}}_ t$ with $\mathbf{A}_ {t+\frac{1}{2}}, \mathbf{B}_{t+\frac{1}{2}}$ that emphasize their role as intermediate matrices within the iterative process.
>
> ### Regarding W2
>
> While Eq. (6) provides a quadratic upper bound of update (1), Eq. (7) is instead the upper bound corresponding to (4), which depends on the choice of $\tilde{\mathbf{A}}_ {t+1} \tilde{\mathbf{B}}_ {t+1}^\top$. Recall that $\Delta \mathbf{W}_ t$ and $\Delta \tilde{\mathbf{W}}_ t$ denote the weight increment under (1) and (4), respectively. As our goal is to identify $\tilde{\mathbf{A}}_ {t+1} \tilde{\mathbf{B}}_ {t+1}^\top$, which renders weight change $\Delta \tilde{\mathbf{W}}_t$, we should analyze the weight $\mathbf{W}_t + \Delta \tilde{\mathbf{W}}_t$ after update (4) rather than $\mathbf{W}_t + \Delta \mathbf{W}_t$ from (1).
>
> ### Regarding W3
>
> The motivation for introducing $\tilde{\mathbf{A}}$ and $\tilde{\mathbf{B}}$ is outlined in the first paragraph of Section 3. To be specific, we aim to acquire a high-rank update that spans a sufficiently rich subspace by accumulating low-rank adapters along diverse directions. To identify the most effective update direction per step, we seek the low-rank matrix that best aligns with full fine-tuning. Moreover, we focus on the gradient rather than momentum because the former affords cleaner analytical formulation. The same practice is also used in LoRA-GA [1] and LoRA-Pro [2]. That says, we agree that aligning the momentum can be an interesting and promising direction for future work.
> [1] S. Wang, L. Yu, and J. Li, “LoRA-GA: Low-Rank Adaptation with Gradient Approximation,” in *NeurIPS*, 2024.
> [2] Z. Wang, J. Liang, R. He, Z. Wang, and T. Tan, “LoRA-Pro: Are Low-Rank Adapters Properly Optimized?”, in *ICLR*, 2025.
> ### Regarding W4
>
> Momentum is essentially the first moment $m(\nabla_{\mathbf{A}} \ell) := \mathbb{E}[\nabla_{\mathbf{A}} \ell]$ estimator of the stochastic gradient, where $\nabla_{\mathbf{A}} \ell$ is viewed as a random matrix, and $\\{ \nabla_{\mathbf{A}_ t} \ell(\mathbf{W}_ t) \\}_ t$ are its samples. In AdamW, it is approximated via the exponential moving average (EMA) $m(\nabla_{\mathbf{A}} \ell) \approx (1-\beta_1) \sum_ {\tau = 0}^t \beta_ 1^{t-\tau} \nabla_{\mathbf{A}_ \tau} \ell(\mathbf{W}_ \tau) := m_t(\nabla_{\mathbf{A}} \ell)$ of gradient samples. Likewise, the momentum of $\tilde{\mathbf{A}}$ is defined as $m(\nabla_{\tilde{\mathbf{A}}} \ell) := \mathbb{E}[\nabla_{\tilde{\mathbf{A}}} \ell]$, and is estimated via its EMA. The key point of Section 3.2 and 3.3 is that, when $\tilde{\mathbf{A}}$ is restricted to be structured transformations of $\mathbf{A}$, the former’s momentum can be analytically computed from the momentum of $\mathbf{A}$. More details are elaborated in Appendix A.4. In each iteration of ScaLoRA, we are scaling the underlying random matrix from $\mathbf{A}, \mathbf{B}$ to $\tilde{\mathbf{A}}, \tilde{\mathbf{B}}$, thus necessitating an update in the momentum.

---

> ### Author Response · Authors · 2025-11-21
>
> ### Regarding W5
>
> Chain of LoRA (CoLA) shares the same high-level idea with ReLoRA, that is, learning a cascade of low-rank adapters. These adapters are sequentially merged into the pretrained weight, with optimizer reset after each merge. As a conceptual alternative to CoLA and in response to multiple reviewers, we have included ReLoRA to Table 2 and Figure 3 of the revised manuscript for both performance and overhead comparisons. ReLoRA improves over LoRA with $r=8$ by 0.77% (LLaMA2-7B) and 0.43% (LLaMA3-8B) on average, though it remains slightly below LoRA with $r=32$. In addition, ScaLoRA(-I) further showcases 0.35% and 0.59% average gains on top of ReLoRA, highlighting the benefits of more frequent direction adjustments and restart-free optimization.
>
> Regarding GaLore, it focuses more on memory-efficient pre-training compared to fine-tuning. Moreover, it relies on low-rank projection as opposed to adapter matrices. For these reasons, we do not classify it as a LoRA variant for our comparisons.
>
> ---
>
> Given most of the reviewer’s questions pertain to understanding the core idea of the method, we hope the above clarifications make the mechanism and motivation of the paper clearer. We welcome any further suggestions for improving the manuscript’s clarity.

---

### Official Review · Reviewer_Zz6k · 2025-10-30

**Soundness:** 3
**Presentation:** 2
**Contribution:** 3
**Rating:** 8
**Confidence:** 2

**Summary:**

The paper tries to allow high-rank updates in LoRA by summing a series of low-rank updates at each step within different subspaces, which is achieved by factoring out a "new LoRA" $\tilde{A}_t \tilde{B}^T_t$ from the LoRA update $A_t B_t^T$ at each step, and merging the remnant into the original weights.

The paper then points out that optimization on LoRA is equivalent to aligning LoRA update with full fine-tuning's weight update. The paper then try to find the optimal $\tilde{A}_t \tilde{B}^T_t$ such that it best matches full fine-tuning updates, and derives the condition of the minimizer $\tilde{A}_t \tilde{B}^T_t$ need to meet. The minimizer is nevertheless non-unique, and the condition requires calculating the SVD of the gradient. Hence the paper consider some minimizers that are specific transformation of the original $A_t B_t^T$, by scalar or column-wise scaling (e.g. $\tilde{A}_t = A_t diag (\alpha), \tilde{B}_t = B_t diag (\beta)$), and derive the optimal scale $\alpha, \beta$ that can be obtained efficiently. In both cases, the new Adam states can be derived. The actual performance is empirically demonstrated, and the efficiency can be further improved by making the updates intermittently.

**Strengths:**

A novel approach to optimally expand the rank of LoRA updates to best match full fine-tuning is presented. The theoretical analysis is interesting.

**Weaknesses:**

The presentation is a bit difficult to follow; the empirical study is limited, and the improvement is small; details below.

**Questions:**

L139: It is a bit confusing when an $\tilde{A}_t \tilde{B}^T_t$ suddenly appears. It will be much easier to understand if the authors mention that this term is something we are going to in the following section, and point out that $\tilde{A}_t$ and $\tilde{B}_t$ are something we create from $A_t$ and $B_t$ at each iteration t.

L148: What is $\Delta \tilde{W}_t$? It appears to be the update to the merged W at step t, i.e., the "actual" update to the weight matrix. It is better to explicitly point this out for easier understanding.

The basic idea looks similar to ReLoRA but use a different formulation of the "new LoRA" at each step to allow much more efficient optimization. The paper will be easier to follow if the authors first introduce ReLoRA in Sec 3 and explicitly discuss how the paper improves over it.

Is the sign of terms in Eq. (7) correct?

L311-314: If it is for the final storage, why can't we still merge the resultant updates into the original weights?

How are the layers where column-wise scaling doesn't work distributed in a model? Is there any pattern?

How can ScaLoRA combine with other recent LoRA improvements, e.g. PiSSA?

I don't think Scalora's advantage in capacity can be demonstrated on GLUE tasks where capacity is not a problem and full fine-tuning leads to worse results compared to standard LoRA. I would suggest more experiments on tasks where standard LoRA fails to match full fine-tuning; check arXiv:2202.07962 for some examples.

---

> ### Author Response · Authors · 2025-11-21
>
> We thank the reviewer for the interest and the thoughtful questions. Next, we address each point in detail.
>
> ### Regarding Q1 and Q2
>
> In response to the reviewer’s advice, Section 3 of the manuscript has been updated to clarify that the subsequent sections will focus on the optimal choice of the alternative adapter $\tilde{\mathbf{A}}_t \tilde{\mathbf{B}}_t^\top$, to elaborate on the definition of $\Delta \tilde{\mathbf{W}}_t$, and to provide an explicit comparison with ReLoRA. The corresponding revisions are highlighted in blue in the updated manuscript.
>
> ### Regarding Q3
>
> The sign in Eq. (7) is correct. To see the reason, note that the weight change for full fine-tuning is $\Delta \mathbf{W}_ t^* := - \frac{1}{L} \nabla \ell(\mathbf{W}_ t)$, where $1/L$ can be viewed as the learning rate. Also recall that the weight update for ScaLoRA is $\Delta \tilde{\mathbf{W}_ t} := -\eta \nabla \ell\mathbf(W_t) \tilde{\mathbf{B}}_ t \tilde{\mathbf{B}}_ t^\top - \eta \tilde{\mathbf{A}}_ t \tilde{\mathbf{A}}_ t ^\top \nabla \ell\mathbf(W_t)$. Thus, Eq. (7) minimizes the discrepancy between the weight increments of full fine-tuning and ScaLoRA through
> $$
> \min_{\tilde{\mathbf{A}}_t, \tilde{\mathbf{B}}_t} \frac{L}{2} || - \Delta \mathbf{W}_t^* + \Delta \tilde{\mathbf{W}}_t ||_F^2 = \frac{L}{2} || \Delta \mathbf{W}_t^* - \Delta \tilde{\mathbf{W}}_t ||_F^2,
> $$
> This is consistent with the equation between (6) and (7).
>
> ### Regarding Q4
>
> For the final storage, $\tilde{\mathbf{A}}_t \tilde{\mathbf{B}}_t^\top$ is merged into $\tilde{\mathbf{W}}_t^{\mathrm{pt}}$, and the stored matrix is the final $\mathbf{W}_t$ rather than its two separate components. We have revised the corresponding description in Section 3.4 to make this merging-and-storage procedure clearer.
>
> ### Regarding Q5
>
> Interestingly, the layers satisfying $\mathbf{v}_t \succeq 0$ exhibit structured patterns, with certain layers more prone than others to violating this condition. A figure has been added to Appendix D.4 to depict these patterns, where column and scalar scaling are marked in black and white, respectively. From Figure 5, it is observed that some layers are consistently updated using column-wise scaling, while others predominantly use scalar scaling, and these patterns vary across tasks. In practice, when such patterns are known a priori, one may fix the scaling scheme accordingly, thereby eliminating the computational overhead of solving the $2r \times 2r$ linear system. We thank the reviewer for bringing this to our attention.
>
> ### Regarding Q6
>
> Unlike MoRA and HiRA that modify the architecture of LoRA, our ScaLoRA is compatible with initialization-based LoRA advances such as PiSSA and LoRA-GA. This is because ScaLoRA is applied *after* initialization to dynamically align the low-rank factors with the evolving gradient, and hence does not conflict with techniques that refine initialization. In other words, initialization tricks can be applied first, and ScaLoRA can be subsequently employed during fine-tuning to further improve convergence and rank.

---

> ### Author Response · Authors · 2025-11-21
>
> ### Regarding Q7
>
> We agree with the reviewer that the NLU tasks in GLUE are relatively simple and a low rank suffices to fit well the datasets. Therefore, we intentionally set $r$ to a small value of 8 (instead of the default 32) for the commonsense reasoning and mathematical problem solving tasks, so that higher ranks are necessitated to adequately capture the underlying task structure. To demonstrate this, we report below the average rank and efficient rank $\text{erank} (\cdot) := || \cdot ||_\mathrm{F}^2 / || \cdot ||_2^2$ across LoRA layers of LLaMA2-7B. For HiRA, its intrinsic (latent) rank is reported in addition to the Euclidean rank, where the former better reflects the low-rank manifold geometry induced by its parameterization. Full results can be found in Table 3 of the updated manuscript. It is observed that ScaLoRA(-I) yields (e)rank proportional to the size and difficulty of the task. For smaller datasets such as ARC-e and ARC-c, the limited fine-tuning iterations results in a moderate-rank update, which is nevertheless sufficient to fit the dataset. In contrast, ReLoRA exhibits markedly lower (e)rank due to its infrequent merging operations. While HiRA consistently produces high Euclidean rank regardless of the dataset size or task difficulty, the intrinsic (e)rank remains low on its low-dimensional manifold. Moreover, the erank of ScaLoRA(-I) is significantly higher than other baselines, suggesting that the weight update captures a richer and more diverse subspace of singular directions for task-specific adaptation.
>
> **Rank / Effective rank**
>
> | Method       |    BoolQ     |     PIQA     |     SIQA     |      HS      |      WG      |    ARC-e     |    ARC-c     |     OBQA     |
> | ------------ | :----------: | :----------: | :----------: | :----------: | :----------: | :----------: | :----------: | :----------: |
> | LoRA         |   8 / 2.7    |   8 / 1.9    |   8 / 1.8    |   8 / 2.3    |   8 / 1.2    |   8 / 1.6    |   8 / 1.7    |   8 / 1.3    |
> | ReLoRA       |   16 / 2.6   |   16 / 1.9   |   24 / 1.9   |   32 / 1.6   |   32 / 2.0   |   16 / 1.7   |   15 / 1.7   |   36 / 2.0   |
> | HiRA (Eucl.) | 4004 / 358.2 | 3925 / 313.8 | 3971 / 312.3 | 3889 / 219.5 | 3670 / 128.4 | 3074 / 167.6 | 3315 / 203.8 | 3729 / 164.5 |
> | HiRA (intr.) |   8 / 2.9    |   8 / 2.4    |   8 / 2.5    |   8 / 1.9    |   8 / 1.5    |   8 / 2.5    |   8 / 2.0    |   8 / 1.7    |
> | ScaLoRA      |  3326 / 4.8  |  3482 / 3.1  |  3661 / 3.4  |  3703 / 4.2  |  3695 / 2.6  |  2254 / 2.7  |  1347 / 1.9  |  3015 / 2.0  |
> | ScaLoRA-I    |  1402 / 4.6  |  1990 / 3.0  |  2757 / 2.6  |  2937 / 4.2  |  2891 / 2.3  |   20 / 2.6   |   20 / 1.9   |  453 / 1.9   |
>
> Extended results for more challenging rank-one setup can be found in our response R3 to Reviewer skhi.

---

### Official Review · Reviewer_hHtD · 2025-10-30

**Soundness:** 3
**Presentation:** 3
**Contribution:** 3
**Rating:** 4
**Confidence:** 4

**Summary:**

ScaLoRA is proposed as an efficient fine-tuning method for large language models, aiming to overcome the limitations of LoRA. While LoRA is computationally efficient, its restricted representation power can hurt performance. ScaLoRA addresses this by dynamically scaling low-rank adapters in directions that minimize the loss at each iteration, thereby accumulating high-rank updates over time. It analytically computes optimal scalar or column-wise scaling coefficients while keeping optimizer states intact. Experiments show that ScaLoRA consistently outperforms LoRA and other high-rank variants in both accuracy and convergence speed.

**Strengths:**

- The motivation and methodology are theoretically clear and well-established.
- Although it does not always guarantee high-rank updates, the use of diagonal scaling over A and B to accumulate diverse directions is novel.
- While the method adds computational overhead, the proposal of ScaLoRA-I provides a practical trade-off between performance and efficiency.

**Weaknesses:**

- W1. Despite the introduction of ScaLoRA-I as a heuristic workaround, the computational cost of ScaLoRA may hinder scalability to larger models.
- W2. There is a lack of strong baselines, and the performance gains are relatively small. LoRA-GA is an important baseline but is not included. Comparisons to other scaling-based methods like LoRA+ are needed.
- W3. Experimental settings, such as the dimension of W in Section 4.1, should be more clearly stated. For example, the fact that W has a dimension of 64 is only mentioned in the appendix.

---
> [1] Wang, Shaowen, Linxi Yu, and Jian Li. "Lora-ga: Low-rank adaptation with gradient approximation." Advances in Neural Information Processing Systems 37 (2024): 54905-54931.
>
> [2] Hayou, Soufiane, Nikhil Ghosh, and Bin Yu. "LoRA+ efficient low rank adaptation of large models." Proceedings of the 41st International Conference on Machine Learning. 2024.

**Questions:**

- Q1. Why didn't you follow the $r=2$, $r=8$ experimental setup from AdaLoRA, which is known to yield strong performance? Similarly, for LLaMA, the $r=4$ setting seems too small given the 4096 hidden dimension. What about higher ranks like 8 (PoLAR) or 32 (DoRA)? Please compare for larger rank performance and copy the results from baseline in PoLAR or DoRA and also compare the empirical runtime/GPU usage for ScaLoRA and ScaLoRA-I. Please also clarify the hyperparameter search ranges used.
- Q2. Please provide a comparison of effective rank across methods.
- Q3. As in Table 3, how does performance vary when only scalar scaling is used, either fixed or trainable? Like LoRA+, it's possible that scale tuning of adapters or their gradients contributed to the performance improvements.
- Q4. It appears that the method must be applied at every iteration. Is it possible to apply it only once at initialization?
- Q5. Does ScaLoRA-I also maintain high rank? In the setup of Figure 2, what is the final rank achieved by ScaLoRA?
- Q6. Is the method truly producing high-rank updates? For example, in the DeBERTa model with a 768 hidden size, the rank is only around 54. In contrast, in the original results in HiRA paper, it achieves a rank of 2800 on LLaMA-3-8B, covering about 70% of 4096, while ScaLoRA covers only 7% on DeBERTa. Furthermore, Is high-rank achieved on LLaMA as well?

Overall, the motivation and theoretical framework are clear and well-grounded. While scalability remains a concern, the heuristic variant ScaLoRA-I addresses this to some extent. However, the paper would benefit from stronger baselines, clearer experimental setups, and more careful tuning. If these concerns are addressed, I would be happy to raise my score.

---

> ### Author Response · Authors · 2025-11-21
>
> We sincerely thank the reviewer for the insightful questions and suggestions. Below please find a one-by-one response to the weaknesses and questions.
>
> ### R1. Scalability of ScaLoRA-I (response to W1)
>
> ScaLoRA-I guarantees a constant percentage of additional time overhead upon choosing $I = \Omega (r)$. Using the complexity analysis in Table 5 of Appendix B, the extra cost of ScaLoRA-I relative to LoRA is $\frac{\mathcal{O}(mnr/I)}{\Omega(kmn)} = \mathcal{O}(1/k)$, where high-order terms are omitted given $r \ll m,n$, and $k$ denotes the input batch size. This overhead does not grow with the model hidden dimension $m$ and $n$, thus ensuring the scalability and efficiency to larger models.
>
> To validate this empirically, we scaled to Gemma-3-27B-pt with $r=8$. On a single H100 GPU, LoRA achieved a throughput of $0.58$ it/s with $64.28$ GB memory usage, whereas ScaLoRA-I with $I=10$ showed $0.56$ it/s ($0.97\times$ LoRA) efficiency and the same memory footprint. Furthermore, extended numerical tests have been added to Appendix D to better motivate ScaLoRA-I, and to justify the choice of $I$ through ablation studies.
>
> ### R2. Additional baselines (response to W2 and Q3)
>
> Per the reviewer’s suggestion, additional baselines including LoRA-GA and ReLoRA have been added to Table 2 and Figure 3 of the revised paper for both performance and overhead comparisons. A copy of the updated table is reproduced below. Both ReLoRA and LoRA-GA improve considerably over vanilla LoRA, while their performance remains slightly below LoRA with $r=32$. In contrast, ScaLoRA(-I) consistently outperforms these variants across settings, demonstrating its robustness.
>
> **LLaMA2-7B**
>
> | Method      | BoolQ | PIQA  | SIQA  |  HS   |  WG   | ARC-e | ARC-c | OBQA  |    Avg    |
> | ----------- | :---: | :---: | :---: | :---: | :---: | :---: | :---: | :---: | :-------: |
> | LoRA        | 87.40 | 81.66 | 59.16 | 82.45 | 79.48 | 82.91 | 57.59 | 58.40 |   73.63   |
> | *ReLoRA*    | 87.80 | 82.48 | 60.08 | 83.23 | 82.56 | 82.95 | 58.11 | 58.00 |   74.40   |
> | *LoRA-GA*   | 87.92 | 83.03 | 60.13 | 82.30 | 82.87 | 83.25 | 56.83 | 58.40 |   74.34   |
> | MoRA        | 87.49 | 82.54 | 59.88 | 82.56 | 79.08 | 83.59 | 58.02 | 57.40 |   73.82   |
> | HiRA        | 87.71 | 82.97 | 59.83 | 83.38 | 81.69 | 82.83 | 55.55 | 57.60 |   73.95   |
> | ScaLoRA     | 87.77 | 82.43 | 60.08 | 83.43 | 82.08 | 83.54 | 58.11 | 58.60 |   74.51   |
> | ScaLoRA-I   | 87.58 | 82.26 | 60.49 | 83.52 | 81.69 | 83.75 | 58.53 | 60.20 | **74.75** |
> | LoRA (r=32) | 88.29 | 82.70 | 60.54 | 83.15 | 82.00 | 82.79 | 57.68 | 59.00 |   74.52   |
>
> **LLaMA3-8B**
>
> | Method      | BoolQ | PIQA  | SIQA  |  HS   |  WG   | ARC-e | ARC-c | OBQA  |    Avg    |
> | ----------- | :---: | :---: | :---: | :---: | :---: | :---: | :---: | :---: | :-------: |
> | LoRA        | 88.99 | 85.09 | 60.95 | 86.09 | 82.64 | 86.62 | 62.29 | 62.00 |   76.83   |
> | *ReLoRA*    | 89.20 | 85.64 | 60.13 | 85.99 | 85.24 | 86.95 | 63.14 | 61.80 |   77.26   |
> | *LoRA-GA*   | 89.69 | 84.98 | 61.00 | 86.58 | 85.32 | 86.11 | 62.29 | 61.80 |   77.22   |
> | MoRA        | 88.56 | 86.18 | 60.29 | 86.69 | 82.40 | 87.79 | 64.08 | 62.20 |   77.27   |
> | HiRA        | 88.87 | 86.07 | 60.64 | 86.11 | 84.53 | 87.12 | 63.91 | 62.40 |   77.46   |
> | ScaLoRA     | 89.20 | 86.18 | 61.82 | 86.51 | 84.53 | 86.57 | 65.61 | 62.40 | **77.85** |
> | ScaLoRA-I   | 89.14 | 86.07 | 62.33 | 86.48 | 83.35 | 86.53 | 64.68 | 62.00 |   77.57   |
> | LoRA (r=32) | 89.69 | 85.47 | 61.72 | 86.76 | 83.35 | 87.08 | 64.08 | 62.20 |   77.54   |
>
> For LoRA+, we followed the suggestion in Q3 to perform an extended ablation test on the Commonsense reasoning dataset using LLaMA2-7B with $r=8$. As shown, the scalar-only variant analogous to LoRA+ suffers a notable performance degradation on SIQA, WG, ARC-c, and OBQA, while performing comparably to ScaLoRA-I on the remaining four datasets. Overall, this leads to an average performance drop of 0.72%, though it still exceeds LoRA by 0.60%. This highlights the importance and expressiveness of column-wise scaling beyond the scalar scaling. The full results are offered in Appendix D.3 of the updated manuscript.
>
> | Method      | BoolQ | PIQA  | SIQA  | HS    | WG    | ARC-e | ARC-c | OBQA  | Avg       |
> | ----------- | ----- | ----- | ----- | ----- | ----- | ----- | ----- | ----- | --------- |
> | LoRA        | 87.40 | 81.66 | 59.16 | 82.45 | 79.48 | 82.91 | 57.59 | 58.40 | 73.63     |
> | ScaLoRA-I   | 87.58 | 82.26 | 60.49 | 83.52 | 81.69 | 83.75 | 58.53 | 60.20 | **74.75** |
> | Scalar-only | 87.31 | 82.32 | 59.37 | 83.60 | 80.93 | 83.38 | 56.11 | 59.20 | 74.03     |

---

> ### Author Response · Authors · 2025-11-21
>
> ### R3. Experimental setups (response to W3 and Q1)
>
> Detailed experimental settings including dimension of $\mathbf{W}$ in the toy test and learning rate search range have been added to Section 4.1 and Appendix C of the revised paper. In particular, the learning rates were selected from $\{0.8, 1, 2, 3, 4, 5, 6, 8, 10, 20 \} \times 10^{-4}$ with finer resolution in the lower range. The learning-rates for HiRA and LoRA-GA are respectively scaled by factors of $10$ and $1/10$ to compensate the magnitude change due to Hadamard product and large initialization. ReLoRA uses a re-initialization frequency of 200 steps with 10 re-warmup steps for the three smaller datasets ARC-e, ARC-c, and OBQA, and a frequency of 2000 steps with 100 re-warmup steps for the remaining larger datasets. LoRA-GA employs a scaling factor $\gamma = 128$ for stability, and a sample batch size of $32$ for gradient estimation.
>
> For the selection of rank, we intentionally restrict the fitting capacity of LoRA and its variants by setting a small $r$. This setup emulates more challenging scenarios where higher ranks are necessitated to adequately capture the underlying task structure. We follow the reviewer’s suggestion to evaluate ScaLoRA-I using LLaMA2-7B with $r = 32$, and compare it against the reported results of PoLAR. To ensure $I = \Omega (r)$, we have scaled it from 10 to 40 accordingly. As shown in the table, ScaLoRA-I achieves consistently strong performance across all eight tasks and surpasses PoLAR by 1.07% on average, highlighting its effectiveness in higher-rank configurations. Moreover, the time overhead of ScaLoRA-I remains $1.047\times$ of LoRA, which aligns with our analysis in R1.
>
> | Method | BoolQ | PIQA  | SIQA  |  HS   |  WG   | ARC-e | ARC-c | OBQA  | Avg  |
> | --- | :---: | :---: | :---: | :---: | :---: | :---: | :---: | :---: | :---: |
> | LoRA | 87.89 | 81.56 | 59.06 | 82.51 | 72.61 | 82.37 | 56.83 | 54.60 | 72.18 |
> | DoRA | 87.61 | 81.45 | 58.70 | 82.50 | 74.43 | 82.28 | 57.17 | 55.60 | 72.47 |
> | PoLAR | 88.13 | 82.64 | 60.03 | 83.12 | 82.00 | 81.99 | 56.14 | 55.60 | 73.71 |
> | ScaLoRA-I | 87.80 | 82.48 | 60.18 | 83.28 | 82.72 | 83.12 | 58.45 | 60.20 | **74.78** |
>
> ### R4. Comparison of rank and effective rank (response to Q2, Q5 and Q6)
>
> As the NLU tasks in GLUE are relatively simple and the RTE dataset is small, a low rank of 54 suffices to fit well the datasets. To show the high-rank claim, we investigate the rank of weight update $\mathbf{W}_ T - \mathbf{W}_ 0$ in the more challenging commonsense reasoning tasks using the larger model LLaMA2-7B with $r=8$. Following ReLoRA and HiRA, only the singular values whose magnitudes exceed 0.005 are counted. MoRA has been excluded because of its nonlinearity. For HiRA, as its rank update pertains to the low-dimensional manifold $\\{ \mathbf{W}^\mathrm{ft} \mid \mathbf{W}^\mathrm{ft} = (\mathbf{A} \mathbf{B}^\top) \odot \mathbf{W}^\mathrm{pt} \\}$, we report both the Euclidean rank and intrinsic (latent) rank, where the latter better reflects the geometry induced by its parameterization. The average rank and efficient rank $\text{erank} (\cdot) := || \cdot ||_\mathrm{F}^2 / || \cdot ||_2^2$ across LoRA layers are reported in the table attached. Full results with standard deviations are offered in Table 3 of the updated manuscript. It is observed that ScaLoRA(-I) yields (e)rank proportional to the size and difficulty of the task. For small datasets such as ARC-e and ARC-c, the limited fine-tuning iterations renders a moderate-rank update, which is nevertheless sufficient to fit the dataset. In contrast, ReLoRA exhibits markedly lower (e)rank due to its infrequent merging operations. While HiRA consistently produces high Euclidean rank regardless of the dataset size and task difficulty, its intrinsic (e)rank remains low owing to its underlying low-dimensional manifold. Moreover, the erank of ScaLoRA(-I) is significantly higher than other baselines, suggesting that the weight update captures a richer and more diverse subspace of singular directions for task-specific adaptation.
>
> **Rank / Effective rank**
>
> | Method | BoolQ | PIQA | SIQA | HS | WG | ARC-e | ARC-c | OBQA |
> | --- | :---: | :---: | :---: | :---: | :---: | :---: | :---: | :---: |
> | LoRA  |   8 / 2.7  | 8 / 1.9 | 8 / 1.8 |  8 / 2.3  |  8 / 1.2  |  8 / 1.6  |  8 / 1.7 |   8 / 1.3 |
> | ReLoRA  |   16 / 2.6 | 16 / 1.9 | 24 / 1.9  |   32 / 1.6   |   32 / 2.0   |   16 / 1.7   |   15 / 1.7  | 36 / 2.0 |
> | HiRA (Eucl.) | 4004 / 358.2 | 3925 / 313.8 | 3971 / 312.3 | 3889 / 219.5 | 3670 / 128.4 | 3074 / 167.6 | 3315 / 203.8 | 3729 / 164.5 |
> | HiRA (intr.) |   8 / 2.9    |   8 / 2.4    |   8 / 2.5 | 8 / 1.9 | 8 / 1.5 | 8 / 2.5 |   8 / 2.0  |   8 / 1.7  |
> | ScaLoRA |  3326 / 4.8  |  3482 / 3.1  |  3661 / 3.4  |  3703 / 4.2  |  3695 / 2.6  |  2254 / 2.7 | 1347 / 1.9 |  3015 / 2.0 |
> | ScaLoRA-I  | 1402 / 4.6  |  1990 / 3.0  |  2757 / 2.6  |  2937 / 4.2  |  2891 / 2.3  |  20 / 2.6  | 20 / 1.9 | 453 / 1.9  |

---

> ### Author Response · Authors · 2025-11-21
>
> ### R5. Apply ScaLoRA to initialization (response to Q4)
>
> Because the moment estimators are not yet established at initialization, applying ScaLoRA at only $t = 0$ translates to solving the unconstrained problem in Eq. (7), whose closed-form solution is provided in Theorem 1. When taking $\mathbf{P}_0 = \mathbf{Q}_0 = \mathbf{I}_r$, this initialization recovers LoRA-GA. Although ScaLoRA could in principle be applied only once at initialization, this would forfeit its key advantage of dynamically aligning the low-rank update with the evolving gradient. Appendix D.1 visualizes the change of the optimal scaling factor over fine-tuning iterations, where large rotation of directions are sometimes required. A consistent performance degradation of LoRA-GA relative to ScaLoRA(-I) is also observed from the tables in R2. Consequently, it is necessary to apply ScaLoRA every (a few) iteration.
>
> ---
>
> Again, we thank the reviewer for the careful reading and constructive feedback. We hope the additional experiments and clarifications address the concerns; we are happy to provide further details or additional analyses if useful.

---

> ### Comment · Reviewer_hHtD · 2025-11-28
>
> Thank you for providing extensive experiments and detailed results.
> While reading the R4 section, I had one additional question I would like to clarify.
>
> From the reported table in R4, ScaLoRA appears to have very high Euclidean raw rank, but its effective rank remains relatively low (around 2–5). For HiRA, authors explicitly report both Euclidean and intrinsic ranks, and the distinction makes it clear why intrinsic effective rank is small even though the Euclidean rank is large.
>
> However, since ScaLoRA does not seem to involve an intrinsic manifold structure, my understanding is that the ranks reported for ScaLoRA correspond purely to Euclidean rank. If this interpretation is correct, I am a bit confused about how to view ScaLoRA as a “high-rank update,” given that its effective rank is quite similar to LoRA’s and does not appear particularly high.
>
> If I am misunderstanding the interpretation of the ranks—or if there is an additional aspect of ScaLoRA’s update structure that explains this behavior—I would greatly appreciate your clarification.

---

> ### Author Response · Authors · 2025-11-28
>
> Thank you for the reply and constructive follow-up question. In the Table of R4, Euclidean ranks are reported for all other methods except for HiRA. While "rank" measures the **number of effective singular values** (i.e., those magnitude >0.005), "erank" captures the **distribution of singular values**. A higher “erank” indicates that the update is closer to a (scaled) orthogonal matrix with evenly distributed singular values.
>
> From the Table, ScaLoRA's "rank" is significantly higher than LoRA, and is even comparable to the HiRA's Euclidean rank. In addition, ScaLoRA's "erank" is also **70.43% higher** than LoRA on average, whereas HiRA's intrinsic "erank" is comparable to LoRA. This suggests the singular values of ScaLoRA are more evenly distributed relative to LoRA and HiRA. While one could further increase the “erank” by imposing an SVD-based factorization such as PoLAR instead of the vanilla LoRA, our focus in this paper is on **improving the number of effective singular values rather than their distribution**.
>
> We appreciate your thoughtful feedback, and we hope this clarifies your concerns.

---

### Author Response · Authors · 2025-11-30
**Additional Comments to AC**

While the review process is currently frozen due to the data leak incident, we would like to bring to your attention three discussions that were directly affected by the freeze.

1. We engaged in an in-depth discussion with Reviewer skhi on **Nov 25–27** to address the questions. The reviewer commented that the concerns were fully resolved, and subsequently raised the score to 8 **before the data leak**. Nevertheless, this score change was reverted as a consequence of the data leak incident.

2. Reviewer hHtD also explicitly expressed **willingness to increase the score** pending additional experiments and clarifications. We have conducted the requested numerical tests and provided detailed responses. Although the discussion was interrupted by the freeze, we believe the reviewer had sufficient grounds to revise the score upward based on the completed updates.

3. Reviewer 2F1U initially assigned a score of 2, but most concerns are regarding **understanding the core idea and methodology** of the paper. Extensive clarifications, further explanations, and corresponding paper updates have been provided to directly address these issues. We hope these efforts could alleviate the reviewer’s concerns and motivate reconsideration.

We respectfully request that the AC take these factors into account when assessing the paper under the unusual circumstances.

---

### Author Response · Authors · 2025-12-03
**Summary of Rebuttal**

To help the AC evaluate our submission, we summarize the key questions raised by each reviewer and provide a concise summary of our responses. All updates described below have been incorporated into the revised manuscript, with references included in parentheses.

1. Reviewer hHtD (rating 4) raised concerns regarding the scalability of ScaLoRA-I, the need for additional baselines, details of the experimental setup, rank comparison, experiments with higher rank, and the applicability of ScaLoRA to initialization. The reviewer noted that **“if these concerns are addressed, I would be happy to raise my score.”**

   In response, we added scalability analysis with computational complexity measured on larger models (Appendix B), included the requested baselines and higher-rank experiments (Table 2, Figure 3, Table 11), detailed our hyperparameter selection rules (Appendix C.3), clarified the comparison between rank and effective rank (Table 3), and analyzed initialization behavior. In the follow-up discussion, we further elaborated on the interpretation and comparison of rank vs. effective rank (Section 4.3).

2. Reviewer Zz6k (rating 8) suggested improving the presentation of Section 3, strengthening the motivation from ReLoRA, explaining Eq. (7), clarifying the final storage, and illustrating the column-wise scaling pattern.

   We revised Section 3 (including the initial 3 paragraphs, and Subsections 3.1 and 3.4) accordingly, and visualized the column-wise scaling patterns (Appendix D.4).

3. Reviewer 2F1U (rating 2) requested clarification of the equations underlying our methodology, including Eqs. (4), (6), and (7), as well as comparisons with additional baselines.

   The rebuttal offers point-by-point explanations to clarify the core ideas, and the revised manuscript now includes the requested baselines (Table 2 and Figure 3).

4. Reviewer skhi (rating 4) asked about the empirical gains and complexity, systematic analysis for hyperparemeter $I$, evaluation on more challenging tasks, additional baselines, and scalability to larger models. The reviewer also commented that **“I would be happy to increase my score if the authors are able to clarify my questions.”**

   Our rebuttal offered detailed analyses of empirical gains and complexity, ablation study for scaling frequency $I$ (Appendix D.2), comparison relative to requested baselines (Table 2 and Figure 3), and both theoretical and numerical scalability tests. **The reviewer expressed satisfaction with our responses and raised the score to 8, though this change was later reverted due to the data leak.**

5. Reviewer yyfD (rating 6) requested comparison with LoRA-GA, clarification of time overhead, experiments with higher rank, inclusion of one missing baseline, and several minor revisions.

   We incorporated comparisons with LoRA-GA as a baseline (Section 4.3), clarified overhead (Appendix B), added the requested experiments, and implemented the suggested textual revisions (lines 167 and 199).

We believe our efforts during the rebuttal sufficiently addressed the majority of reviewers’ questions and concerns, and we hope this summary provides a clear overview of the improvements made to the revised manuscript.

---

### Meta-Review · Area_Chair_dmP5 · 2025-12-30

**Summary:**

This paper proposes **ScaLoRA**, a method that accumulates progressively higher-rank weight updates by optimally scaling low-rank adapters across training iterations. The authors position the method as an extension of LoRA that better approximates full fine-tuning while retaining parameter efficiency. Several reviewers found the theoretical development rigorous and the presentation clear, and noted consistent (though modest) empirical gains over LoRA-style baselines across NLP benchmarks.

However, a **fundamental concern remains unresolved**, most prominently raised by **Reviewer 2F1U**:
**ScaLoRA is not a low-rank adaptation method in the standard sense**, but rather a form of **memory-efficient fine-tuning that incrementally accumulates high-rank updates**. Because the method explicitly merges updates into the base weights over time, it departs from the defining constraint of LoRA-like methods (i.e., maintaining a fixed low-rank parameterization).

As a result, the **most relevant comparisons are missing**. Instead of focusing primarily on LoRA variants, the paper should be compared against **memory-efficient fine-tuning methods that explicitly aim to approximate full training dynamics**, including:

* **GaLoRe** — Gradient Low-Rank Projection
  [https://arxiv.org/pdf/2403.03507](https://arxiv.org/pdf/2403.03507)
* **BAdam** — Blockwise Adaptive Moments
  [https://proceedings.neurips.cc/paper_files/paper/2024/file/2c570b0f9938c7a58a612e5b00af9cc0-Paper-Conference.pdf](https://proceedings.neurips.cc/paper_files/paper/2024/file/2c570b0f9938c7a58a612e5b00af9cc0-Paper-Conference.pdf)
* **Frugal** — Frugal Fine-Tuning
  [https://openreview.net/pdf?id=B4TyAILcE4](https://openreview.net/pdf?id=B4TyAILcE4)
* **FIRA** — Fine-Tuning via Incremental Rank Adaptation
  [https://arxiv.org/pdf/2410.01623](https://arxiv.org/pdf/2410.01623)

All of these methods perform **full fine-tuning–level updates with strong memory efficiency guarantees**, making them more appropriate baselines than classical LoRA variants.

In addition, while the paper provides detailed approximation analysis, **the theoretical results are not substantially novel**. Similar arguments—showing how low-rank or structured updates approximate full-gradient dynamics—appear in prior work, including:

* **LoRA-Pro**
  [https://openreview.net/pdf?id=gTwRMU3lJ5](https://openreview.net/pdf?id=gTwRMU3lJ5)
* **AltLoRA**
  [https://openreview.net/attachment?id=9YNJ03jYsU&name=pdf](https://openreview.net/attachment?id=9YNJ03jYsU&name=pdf)
* **LOFT**
  [https://arxiv.org/pdf/2505.21289](https://arxiv.org/pdf/2505.21289)

The current manuscript does not clearly articulate what is fundamentally new in its approximation guarantees relative to these works, beyond a specific instantiation via column-wise scaling.

## Recommendation

**Reject**

Despite being technically competent and well written, the paper suffers from **mischaracterization of its contribution**, **incomplete and misaligned baselines**, and **limited theoretical novelty**. These are fundamental issues that outweigh the empirical improvements reported.

**Reviewer Concerns:**

### Addressed

* Improved clarity of equations and optimization dynamics.
* Additional ablation studies and expanded experimental results.
* Inclusion of some additional LoRA-related baselines (e.g., ReLoRA, LoRA-GA).
* Scalability discussion via the ScaLoRA-I variant.

### Outstanding

* **Conceptual misclassification** of the method as low-rank adaptation rather than memory-efficient fine-tuning.
* **Lack of comparison with the most relevant state-of-the-art methods** (GaLoRe, BAdam, Frugal, FIRA).
* **Limited novelty of the theoretical analysis** relative to prior work on approximating full fine-tuning with low-rank or structured updates.
* Insufficient justification that ScaLoRA represents a fundamentally new paradigm rather than an incremental refinement.

**Reviewer Scores:**

* **Reviewer 2F1U** would likely maintain a **low score**, as the core conceptual concern remains unresolved.
* Other reviewers expressed conditional positivity contingent on additional experiments and clarifications, many of which were addressed; however, these improvements do not resolve the central positioning and novelty issues.
* Overall, the reviews remain **polarized**, with at least one strong rejection grounded in foundational concerns.

---

### Decision · Program_Chairs · 2026-01-26

Reject